# Reading between the rational sections:
# Global structures of 4d $\mathcal{N} = 2$ KK theories

Cyril Closset[1] and Horia Magureanu[2]

**1** School of Mathematics, University of Birmingham,
Watson Building, Edgbaston, Birmingham B15 2TT, UK
**2** Mathematical Institute, University of Oxford,
Andrew-Wiles Building, Woodstock Road, Oxford, OX2 6GG, UK

## Abstract

We study how the global structure of rank-one 4d $\mathcal{N} = 2$ supersymmetric field theories is encoded into global aspects of the Seiberg-Witten elliptic fibration. Starting with the prototypical example of the $\mathfrak{su}(2)$ gauge theory, we distinguish between relative and absolute Seiberg-Witten curves. For instance, we discuss in detail the three distinct *absolute curves* for the $SU(2)$ and $SO(3)_\pm$ 4d $\mathcal{N} = 2$ gauge theories. We propose that the 1-form symmetry of an absolute theory is isomorphic to a torsion subgroup of the Mordell-Weil group of sections of the absolute curve, while the full defect group of the theory is encoded in the torsion sections of a so-called *relative curve*. We explicitly show that the relative and absolute curves are related by isogenies (that is, homomorphisms of elliptic curves) generated by torsion sections – hence, gauging a one-form symmetry corresponds to composing isogenies between Seiberg-Witten curves. We apply this approach to Kaluza-Klein (KK) 4d $\mathcal{N} = 2$ theories that arise from toroidal compactifications of 5d and 6d SCFTs to four dimensions, uncovering an intricate pattern of 4d global structures obtained by gauging discrete 0-form and/or 1-form symmetries. Incidentally, we propose a 6d BPS quiver for the 6d M-string theory on $\mathbb{R}^4 \times T^2$.

 Check for updates

# 1    Introduction

Local quantum field theories often contain extended operators or defects. There can exist several consistent quantum field theories with the same local dynamics, which are distinguished by choosing different consistent sets of extended operators. Moreover, extended operators can be charged under generalised symmetries [1]. For instance, any four-dimensional $\mathfrak{su}(2)$ gauge theory with matter fields in odd-dimensional irreducible representations of the gauge algebra admits a set of mutually-local Wilson and/or 't Hooft lines. Choosing a consistent maximal set of lines is famously equivalent to choosing the gauge group and discrete $\theta$-angle, $SU(2)$ or $SO(3)_\pm$ [2, 3].

A given quantum field theory without a choice of consistent extended operators is sometimes called a *relative theory*, while any consistent choice of extended operators upgrades the relative theory to an *absolute theory*, assuming that a such a choice is possible. Given a relative

theory, the choice of an absolute theory is usually called the *global structure* of the quantum field theory. In the gauge-theory example just mentioned, the relative theory is already determined by the Lie algebra $\mathfrak{su}(2)$, while the global structure is essentially the choice of a Lie group with this Lie algebra. The absolute theory then possesses a 1-form symmetry – the electric $\mathbb{Z}_2^{[1]}$ center symmetry for $SU(2)$, or its magnetic version for the $SO(3)$ gauge theory.

Let us consider 4d $\mathcal{N} = 2$ supersymmetric quantum field theories (SQFT), whose low-energy dynamics on the Coulomb branch (CB) is famously described by the Seiberg-Witten (SW) geometry [4,5]. We are interested in the global structure of rank-one 4d $\mathcal{N} = 2$ SQFTs, including Kaluza-Klein (KK) field theories obtained by compactifying certain 5d superconformal field theories (SCFTs) on a circle, as well as from the 6d $\mathcal{N} = (2,0)$ $A_1$ theory (also known as the M-string theory) compactified on a torus. We will also clarify aspects of the global structure of the prototypical 4d $\mathcal{N} = 2$ gauge theory, the pure $SU(2)$ gauge theory first solved by Seiberg and Witten.

In this paper, we limit our investigation to rank-one theories. Then, the SW geometry is a rational elliptic surface (RES), namely a one-parameter family of elliptic curves – simply known as *the* Seiberg-Witten curve. The singularity structure of the elliptic fibration over the Coulomb branch encodes the low-energy dynamics, including the presence of massless states that can appear in the non-perturbative regime. It is natural to ask whether this low-energy description also encodes the global structure of the theory – *e.g.* how does it distinguish between the pure $SU(2)$ and $SO(3)$ gauge theories, whose local dynamics are identical? A simple argument tells us that the $SU(2)$ and $SO(3)$ curves should be related by a particular rescaling of their periods [6]. Surprisingly, however, the pure $SO(3)_\pm$ SW curves have not appeared explicitly in the literature. In this paper, we carefully distinguish between the various SW curves for the relative and absolute versions of all rank-one $\mathcal{N} = 2$ SQFTs with non-trivial 1-form symmetries. In the case of the $\mathfrak{su}(2)$ gauge theories, we have four distinct SW geometries:

$$
\begin{array}{ccccc}
SU(2) & & \mathfrak{su}(2) & & SO(3)_\pm \\
(I_4^*; 2I_1), \quad \Gamma^0(4) & \xleftarrow{\;\;\text{2-iso}\;\;} & (I_2^*; 2I_2), \quad \Gamma(2) & \xleftarrow{\;\;\text{2-iso}\;\;} & (I_1^*; I_1, I_4), \quad \Gamma_0(4). \\
\Phi(\mathcal{S}) = \mathbb{Z}_2 & & \Phi(\mathcal{S}) = \mathbb{Z}_2 \oplus \mathbb{Z}_2 & & \Phi(\mathcal{S}) = \mathbb{Z}_4
\end{array} \tag{1}
$$

Here, the second line denotes the singularity structure on the Coulomb branch, with the notation $(F_\infty; F_\nu)$ for the fiber at infinity and $F_\nu$ the bulk singularity fibers, and we also give the modular group $\Gamma \subset \mathrm{PSL}(2,\mathbb{Z})$ associated to these Coulomb branches (that is, the CB is a genus-0 modular curve for that modular group) – see appendix A for a brief review, and [7,8] for a detailed introduction to the RES formalism. It is important to note that the $F_\nu = I_k$ singularities in (1) are all undeformable in the sense of [9]. The third line in (1) displays the Mordell-Weil (MW) group of rational sections of the SW geometry $\mathcal{S}$, denoted by $\Phi(\mathcal{S})$. As proposed in [7], the MW group encodes the 1-form symmetry of the theory; see also [10] for an earlier, closely related discussion. The main purpose of this paper is to further elucidate this relationship by understanding how to gauge a 1-form symmetry at the level of the SW geometry.

The Seiberg-Witten curve first derived in [4] was the $\Gamma(2)$ curve, while the $\Gamma^0(4)$ curve was proposed in [5] as the $N_f = 0$ curve in the series of SW curves that describe $SU(2)$ SQCD with $N_f$ flavours. It was already pointed out in [5] that the two curves are related by an isogeny. Recall that an $N$-isogeny is an $N$-to-1 homomorphism of elliptic curves – that is, a $N$-to-1 rational map that preserves the zero-section, and hence the abelian group structure on the curve. In the past literature, it was sometimes assumed, more or less explicitly, that curves related by isogenies were physically equivalent. In this work, instead, we insist on the fact that only the $\Gamma^0(4)$ curve in (1) corresponds to the pure $SU(2)$ gauge theory. The $\Gamma(2)$ curve still plays an important role, however. We identify it as the *relative curve* for the $\mathfrak{su}(2)$ gauge theory viewed as a relative theory, before any choice of global form. The hallmark of a

relative curve is that the torsion part of its Mordell-Weil group encodes the *defect group* $\mathbb{D}$ of the theory. Then, one can obtain all possible *absolute curves* – the SW curves for the distinct absolute theories – by performing isogenies along Lagrangian subgroups of $\mathbb{D}$. In particular, for the $\mathfrak{su}(2)$ theory, we obtain in this way the SW curves with modular group $\Gamma_0(4)$ that correspond to the $SO(3)_\pm$ theories. The isogenies are performed 'along torsion sections', as we will explain. Given a rational section of order $N$ in $\Phi_{\text{tor}}(\mathcal{S})$, for either a relative or an absolute curve, one can define the isogenous SW geometry as a quotient of the elliptic fiber. By exploring these $N$-to-1 relations in much detail, we will show that gauging $\mathbb{Z}_N^{[1]}$ 1-form symmetries corresponds to a composition to two $N$-isogenies, as in the $\mathfrak{su}(2)$ example (1). We will thus learn how to 'read between the rational sections' of the Seiberg-Witten geometry, in close parallel with how one can 'read between the defect lines' of 4d gauge theories more generally [3]. Our approach builds on many previous works on this and other closely related subjects [2, 10–15] – in particular, a proposed relation between global structures and isogenies already appeared (somewhat obliquely) in [2]. We will further comment on the wider picture, and on perspectives for future work, in the final section.

We analyse all the rank-1 4d $\mathcal{N} = 2$ supersymmetric field theories with non-trivial 1-form symmetries, since there are very few of them. If we restrict ourselves to theories with a four-dimensional UV completion, these are only the pure $\mathfrak{su}(2)$ gauge theory, which is asymptotically free, and the 4d $\mathcal{N} = 2^*$ $\mathfrak{su}(2)$ theory, which is UV completed by the 4d $\mathcal{N} = 4$ SCFT. There also exists very important theories with a 5d or 6d UV completion, namely the $E_1$ and $E_0$ 5d SCFTs, and the 6d M-string theory. All these theories (except for $E_0$) are related through various limit, with the 6d M-string theory being the "grandparent" theory. We will also see that the 5d (and 6d) theories have a richer structure of global structures in 4d due to the presence of both 1-form and 0-form symmetries that descend from the (electric) 1-form symmetry in 5d. We map out all these global structures. Along the way, we also briefly comment on the 5d BPS quivers for these theories [8, 16], and in particular we identify the 6d BPS quiver that describes the 4d $\mathcal{N} = 2$ BPS states of the M-string theory compactified on $T^2$.

This paper is organised as follows. In section 2, we study the pure $\mathfrak{su}(2)$ gauge theory and we abstract general lessons on how to properly 'read between the rational sections' by formulating three physics conjectures. In section 3, we study the $\mathcal{N} = 2^*$ theory, which enjoys a non-trivial $S$-duality group acting on the UV gauge coupling. We study the SW geometries for the 5d theories $E_1$ and $E_0$ in sections 4 and 5, respectively. We discuss the 6d M-string theory in section 6. Finally, we present some conclusions and challenges for future work in section 7. Various useful background materials are collected in two appendices.

## 2 Global aspects of rank-one Seiberg-Witten geometries

The low-energy effective field theory on the Coulomb branch of any rank-one 4d $\mathcal{N} = 2$ SQFT is written in terms of the low-energy 'photon' $a$ (*i.e.* the scalar in the low-energy $U(1)$ vector multiplet) at any given point on the Coulomb branch, $u \in \mathcal{B}$. The SW solution gives us the exact expression $a(u)$ as a 'physical period' of the SW curve $E$ fibered over the Coulomb branch,

$$E \longrightarrow \mathcal{S} \longrightarrow \mathcal{B}. \tag{2}$$

It is defined as the electric period of the Seiberg-Witten differential $\lambda$:

$$a \equiv \oint_{\gamma_A} \lambda, \qquad a_D \equiv \oint_{\gamma_B} \lambda. \tag{3}$$

More precisely, we have a non-trivial rank-2 $SL(2, \mathbb{Z})$ vector bundle of physical periods of $E$ with sections $(a_D, a)$. The periods determine the exact prepotential $\mathcal{F}(a)$ and the low-energy

effective gauge coupling, $\tau(u)$, of the 4d $\mathcal{N} = 2$ theory:

$$a_D = \frac{\partial \mathcal{F}}{\partial a}, \qquad \tau = \frac{\partial a_D}{\partial a}. \tag{4}$$

The SW geometry (2) is a one-parameter family of elliptic curves with a section, such that the total space $\mathcal{S}$ is a rational elliptic surface (RES). It is best presented in Weierstrass normal form:

$$y^2 = 4x^3 - g_2(u) x - g_3(u), \tag{5}$$

where $g_2$ and $g_3$ are functions $u$ as well as of various masses and marginal gauge couplings that may occur in a given theory. We review some standard facts about elliptic curves in appendix A. Note that the point at infinity ($u \to \infty$) on the Coulomb branch $\mathcal{B} \cong \mathbb{P}^1$ is singled out physically as the UV cusp – for 4d $\mathcal{N} = 2$ asymptotically-free gauge theories, it is a weak-coupling cusp. We refer to [7] for a more thorough introduction to the RES formalism and to appendix A.1 for a brief review of the types of singularities that can appear on the CB of rank-one theories.

## 2.1 The pure $\mathfrak{su}(2)$ theories

Let us start by reviewing the Seiberg-Witten geometry for the 4d $\mathcal{N} = 2$ pure $SU(2)$ gauge theory. We then introduce the Seiberg-Witten geometries from the pure $SO(3)_\pm$ theories and discuss how they differ from the $SU(2)$ curve. The $SU(2)$ and $SO(3)_\pm$ gauge theories are related to each other by gauging one-form symmetries. In subsection 2.2, we will explain how this gauging can be performed directly at the level of the Seiberg-Witten curves.

### 2.1.1 The $SU(2)$ curve

The SW geometry for the pure $SU(2)$ theory is given by [4]:

$$y^2 = x^3 + ux^2 + \frac{1}{4}\Lambda^4 x, \qquad \lambda = \frac{1}{4\pi} \frac{y \, dx}{x^2}, \tag{6}$$

where $\Lambda$ is the dynamically-generated scale of the $SU(2)$ gauge theory, and $\lambda$ is the SW differential. The parameter $u$ is the Coulomb branch VEV, which is related to the low-energy photon $a$ by:

$$u \equiv \left\langle \operatorname{Tr} \Phi^2 \right\rangle \approx -a^2, \qquad \Phi = -\frac{i}{\sqrt{2}} \begin{pmatrix} a & 0 \\ 0 & -a \end{pmatrix}, \tag{7}$$

in our conventions.[1] The approximation $u \approx -a^2$ is only valid in the semi-classical limit $|u| \to \infty$.

**The Seiberg-Witten curve.** By a simple change of coordinate $x \to x - \frac{u}{3}$, $y \to \frac{1}{2}y$, the $SU(2)$ curve (6) can be brought to the Weierstrass normal form (5) with:

$$g_2^{SU(2)} = \frac{4u^2}{3} - \Lambda^4, \qquad g_3^{SU(2)} = -\frac{8u^3}{27} + \frac{u\Lambda^4}{3}, \tag{8}$$

with the discriminant:

$$\Delta^{SU(2)} = \Lambda^8 (u - \Lambda^2)(u + \Lambda^2), \tag{9}$$

and the SW differential:[2]

$$\lambda = \frac{1}{8\pi} \frac{y \, dx}{(x - \frac{u}{3})^2}, \qquad \frac{d\lambda}{du} = \frac{1}{2\pi} \frac{dx}{y}. \tag{10}$$

---

[1]Our parameter $u$ corresponds to $-u$ in [4, 5].

[2]Note that $\lambda$ is defined only up to a differential on the curve, *i.e.* up to a shift $\lambda \to \lambda + dh$.

The exact geometric periods,

$$\omega_a = \oint_{\gamma_A} \frac{dx}{y}, \qquad\qquad \omega_D = \oint_{\gamma_B} \frac{dx}{y}, \qquad (11)$$

are thus related to the physical periods $a$ and $a_D$ according to:

$$\omega_a = 2\pi \frac{da}{du}, \qquad\qquad \omega_D = 2\pi \frac{da_D}{du}. \qquad (12)$$

At large $u$, the periods have a leading divergence:

$$a(u) \approx -i\sqrt{u}, \qquad a_D(u) \approx -\frac{2\sqrt{u}}{\pi} \log \frac{\Lambda^2}{u}, \qquad (13)$$

which gives us the following $I_4^*$ monodromy at infinity:

$$\begin{pmatrix} a_D \\ a \end{pmatrix} \to \mathbb{M}_\infty \begin{pmatrix} a_D \\ a \end{pmatrix}, \quad \text{as} \quad \frac{1}{u} \to e^{2\pi i}\frac{1}{u}, \qquad \mathbb{M}_\infty = -T^4 = \begin{pmatrix} -1 & -4 \\ 0 & -1 \end{pmatrix}, \qquad (14)$$

as expected from the one-loop prepotential [4,5]. In the interior of the Coulomb branch, the exact periods determine the mass of the BPS particles. At a generic point $u \in \mathcal{B}$, there may exist massive half-BPS one-particle excitations with magnetic-electric charges $\gamma \equiv (m, q) \in \Gamma \cong \mathbb{Z}^2$ under the low-energy $U(1)$. The central charge of such a dyon reads:

$$Z_\gamma(u) = m\, a_D(u) + q\, a(u). \qquad (15)$$

Given two dyons $\gamma_1 = (m_1, q_1)$ and $\gamma_2 = (m_2, q_2)$, their Dirac pairing is given by:

$$\langle \gamma_1, \gamma_2 \rangle = m_1 q_2 - q_1 m_2 = [\gamma_1] \cdot [\gamma_2]. \qquad (16)$$

Here, the charge $\gamma = (m, q)$ can be identified with a homology 1-cycle on the SW curve:

$$[\gamma] = m\gamma_B + q\gamma_A, \qquad \Gamma \cong H_1(E, \mathbb{Z}), \qquad (17)$$

and the Dirac pairing is identified with the intersection pairing $[\gamma_1] \cdot [\gamma_2]$. In other words, this SW geometry is given by a family of principally-polarised elliptic curves.[3]

**Coulomb branch singularities.** The SW singularities at strong coupling correspond to the two Kodaira singularities of type $I_1$ that are apparent from (8)-(9). At $u = \pm\Lambda$, the monopole $\gamma_M = (1, 0)$ and the dyon $\gamma_D = (-1, 2)$ become massless, respectively, which gives us the non-trivial monodromies [5] (following (A.5)):

$$\mathbb{M}_{(1,0)} = STS^{-1} = \begin{pmatrix} 1 & 0 \\ -1 & 1 \end{pmatrix}, \qquad \mathbb{M}_{(-1,2)} = (T^2 S)T(T^2 S)^{-1} = \begin{pmatrix} -1 & 4 \\ -1 & 3 \end{pmatrix}. \qquad (18)$$

The pure $SU(2)$ theory has a classical $U(1)_R$ R-symmetry which is reduced to a $\mathbb{Z}_8^{(R)}$ symmetry in the quantum theory (due to the gauge-$U(1)_R$ ABJ anomaly). This discrete $R$-symmetry is spontaneously broken to $\mathbb{Z}_4$ on the Coulomb branch, wherein $\mathbb{Z}_2 \subset \mathbb{Z}_8^{(R)}$ acts on the Coulomb branch as a sign flip, $u \to -u$. In particular, this $\mathbb{Z}_2$ action exchanges the two SW singularities, which are therefore physically equivalent.

---

[3]For our purpose, we could take the fact that the Dirac pairing and homology pairing agree as the *definition* of what it means to have a principal polarisation of $E$. We refer to [13] for a recent discussion of polarisations of SW geometries that generalises to higher rank.

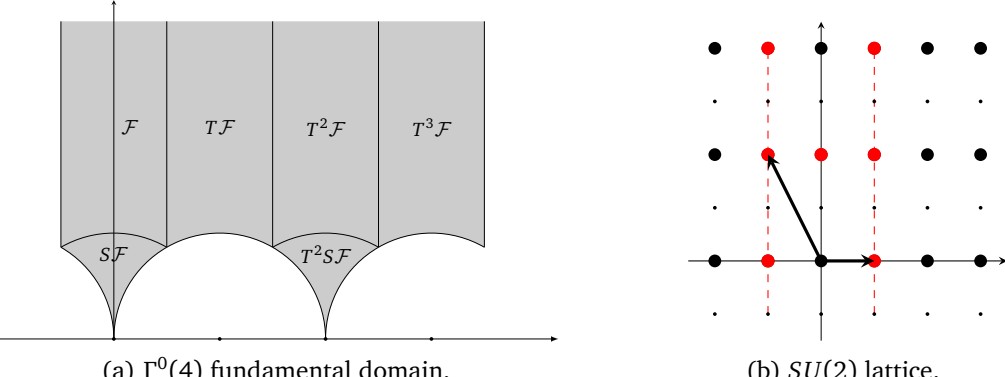

| (a) $\Gamma^0(4)$ fundamental domain. | (b) $SU(2)$ lattice. |
|---|---|

Figure 1: LEFT: Fundamental domain for $\Gamma^0(4)$ on the upper-half-plane $\mathcal{H} \cong \{\tau\}$. The width-1 cusps at $\tau = 0$ and $\tau = 2$ correspond to the monopole and dyon points, respectively. The width-4 cusp at $\tau = i\infty$ is the weak-coupling point. RIGHT: The charge lattice $\Gamma$ of the $SU(2)$ theory (small dots), with the allowed dynamical particle charges $\gamma \in \widehat{\Gamma}_{SU(2)}$ denoted by larger dots, and with the weak-coupling BPS spectrum coloured in red.

The $SU(2)$ Coulomb branch is also a modular curve for the modular group $\Gamma^0(4)$ – that is, $\mathcal{B} \cong \mathcal{H}/\Gamma^0(4)$, where the upper-half-plane $\mathcal{H}$ is spanned by the effective holomorphic gauge coupling $\tau$ (see appendix A.5 for a review). The $\Gamma^0(4)$ modular function is given by [7,17,18]:

$$\frac{u(\tau)}{\Lambda^2} = \frac{\vartheta_2(\tau)^4 + \vartheta_3(\tau)^4}{2\vartheta_2(\tau)^2\vartheta_3(\tau)^2} = 1 + \frac{1}{8}\left(\frac{\eta(\frac{\tau}{4})}{\eta(\tau)}\right)^8. \tag{19}$$

The SW singularities are then mapped to width-1 cusps of $\Gamma^0(4)$ as shown in figure 1. (See also [18,19] for more details on how such maps are explicitly realised.)

**BPS states and BPS lines.** Given any SW geometry, we can also, in principle, compute the spectrum of BPS particles at a given point $u \in \mathcal{B}$ – that is, the set of charges $\gamma$ that correspond to stable BPS states. In the present case, the BPS spectrum is given by [4,20]:

$$\mathscr{S}_S : \quad (1,0), \quad (-1,2), \qquad \mathscr{S}_W : \quad (0,2), \quad (1,2n), \quad n \in \mathbb{Z}, \tag{20}$$

at strong and weak coupling, respectively, together with the antiparticles of opposite charge. This is depicted in figure 1b. In particular, the state $\gamma_W = (0,2)$ is the $W$-boson. All BPS particles can be viewed as bound states of the simple states $\gamma_M = (1,0)$ and $\gamma_D = (-1,2)$ that become light at the SW singularities, as encoded by the BPS quiver [21]:

$$\left(\gamma_M\right)\!\longrightarrow\!\!\!\longrightarrow\!\left(\gamma_D\right). \tag{21}$$

The only allowed BPS states in this theory, anywhere on the Coulomb branch, are of the form $(m, 2n)$ for $m, n \in \mathbb{Z}$. The set of allowed BPS states form a sublattice $\widehat{\Gamma}_{SU(2)}$ of the lattice $\Gamma \cong \mathbb{Z}^2$ of possible magnetic-electric charges of the theory.

We may also consider all the possible BPS lines allowed in this theory, which are best viewed as the worldlines of background (non-dynamical) dyons with charges $\gamma_L$ such that:

$$\langle \gamma_L, \gamma \rangle \in \mathbb{Z}, \qquad \forall \, \gamma \in \widehat{\Gamma}_{SU(2)}. \tag{22}$$

In general, given a lattice of allowed charges, there might exist several distinct consistent sets of genuine line operators [2,3] (see appendix B for a review). Our claim is that, given a

principally-polarised SW geometry, we have already chosen this global structure implicitly. In the present case, the only genuine lines that are allowed are such that:

$$\gamma_L \in \Gamma, \tag{23}$$

since any other choice of lines would not be compatible with the homology lattice for this SW curve. (For instance, the magnetic line $\left(\frac{1}{2}, 0\right)$ satisfies (22) but not (23).) In the Type-IIB geometric engineering picture, this corresponds to D3-branes wrapping the one-cycle $\gamma \subset E$ at $u \in \mathcal{B}$ and stretching all the way to infinity. Most of those lines can be screened by the dynamical particles, except for the ones valued in:

$$\Gamma/\widehat{\Gamma}_{SU(2)} \cong \mathbb{Z}_2, \tag{24}$$

namely the lines $\gamma_L = (0, 0)$ and $(0, 1)$, which are charged under the electric one-form symmetry group:

$$\Gamma^{[1]} \equiv \operatorname{Hom}\left(\Gamma/\widehat{\Gamma}_{SU(2)}, U(1)\right) = \mathbb{Z}_2^{[1]}. \tag{25}$$

Recall that, given the BPS spectrum (20), the one-form symmetry $\Gamma^{[1]}$ can be computed as the subgroup of the accidental $U(1)_m^{[1]} \times U(1)_e^{[1]}$ one-form symmetry acting on the infrared photon that preserves the BPS states [14]. Its generators $g^{\Gamma^{[1]}} = (k_m, k_q)$ act on a BPS state $|\gamma\rangle$ with charges $\gamma = (m, q)$ by a phase:

$$g^{\Gamma^{[1]}}: \quad |\gamma\rangle \rightarrow e^{2\pi i(k_m m + k_q q)}|\gamma\rangle. \tag{26}$$

Here, we have, we have $g^{\mathbb{Z}_2^{[1]}} = \left(0, \frac{1}{2}\right)$ so that, indeed, $\Gamma^{[1]} = \mathbb{Z}_2^{[1]}$ as a subgroup of the electric one-form symmetry $U(1)_e^{[1]}$ [7].

**Torsion section and one-form symmetry.** A very important property of the Seiberg-Witten geometry (8) is that it has a non-trivial Mordell-Weil (MW) group:

$$\Phi(\mathcal{S}) = \Phi_{\text{tor}}(\mathcal{S}) = \mathbb{Z}_2, \qquad P_{\mathbb{Z}_2} = \left(\frac{u}{3}, 0\right), \tag{27}$$

which is generated by the non-trivial rational torsion section, as indicated. (That is, we have $2P_{\mathbb{Z}_2} = 0$. See [7] for an introduction to the MW group in the present context.) In the present case, we identify the MW group with the 1-form symmetry, $\Phi_{\text{tor}}(\mathcal{S}) \cong \Gamma^{[1]}$ [7].

### 2.1.2 The $SO(3)_\pm$ curves

The $SO(3)$ $\mathcal{N} = 2$ gauge theory differs from the $SU(2)$ $\mathcal{N} = 2$ theory in subtle but interesting ways. These differences can be used to determine the correct $SO(3)$ Seiberg-Witten curve, as we now explain. In subsection 2.2, we will see how this can be better understood in terms of gauging the $\mathbb{Z}_2^{[1]}$ one-form symmetry of the $SU(2)$ theory, which can be done directly at the level of the SW curve.

**The $SO(3)$ normalisation.** The simplest and most important difference between the $SU(2)$ and the $SO(3)$ theories is that the semi-classical Higgs mechanism gives us two distinct normalisations of the low-energy 'electric' $U(1)$, accounting for the fact that the spin-$\frac{1}{2}$ representation of $SU(2)$ does not exists in the $SO(3)$ theory. Conversely, there are 'twice as many' magnetic fluxes allowed in the $SO(3)$ theory than in the $SU(2)$ theory; in particular, $SO(3)$ bundles on $S^2$ (e.g. a sphere linking the monopole worldline) can have non-trivial Stiefel–Whitney class.

Relatedly, the $\theta$-angle of the $SO(3)$ theory is now valued in $[0, 4\pi)$, instead of $[0, 2\pi)$ for the $SU(2)$ $\theta$-angle. As such, there actually exists two distinct $SO(3)$ theories, denoted by $SO(3)_{\pm}$, which are related by a shift of the $\theta$-angle by $2\pi$ [3]:

$$SO(3)_{-}^{\theta} = SO(3)_{+}^{\theta + 2\pi}. \tag{28}$$

Then, when comparing magnetic and electric charges between the two theories, we expect:

$$
\begin{aligned}
(m_+, q_+) &= \left(2m, \frac{q}{2}\right), \\
(m_-, q_-) &= \left(2m, \frac{q}{2} - m\right) = \left(m_+, q_+ - \frac{m_+}{2}\right),
\end{aligned}
\tag{29}
$$

where $(m_{\pm}, q_{\pm})$ and $(m, q)$ denote the magnetic-electric charges under the low-energy $U(1)$ for $SO(3)_{\pm}$ and $SU(2)$, respectively. The shift of the electric charge in the $SO(3)_{-}$ theory is due to the Witten effect [22]. The low-energy photons are related by the inverse transformation, namely:

$$
\begin{aligned}
a_D^{SO(3)_+} &= \frac{1}{2} a_D^{SU(2)}, & a^{SO(3)_+} &= 2 a^{SU(2)}, \\
a_D^{SO(3)_-} &= \frac{1}{2} a_D^{SU(2)} + a^{SU(2)}, & a^{SO(3)_-} &= 2 a^{SU(2)},
\end{aligned}
\tag{30}
$$

since the local dynamics is unaffected by the choice of global form of the gauge group – in particular, the central charge (15) is insensitive to this choice. It also follows from (30) that the effective gauge couplings of the distinct theories are related as:

$$\tau_{SO(3)_+} = \frac{1}{4}\tau_{SU(2)}, \qquad \tau_{SO(3)_-} = \tau_{SO(3)_+} + \frac{1}{2}. \tag{31}$$

**The Seiberg-Witten curve.** The $SO(3)$ curve must be related to the $SU(2)$ curve in such a way that its periods are rescaled as in (30) – to the best of our knowledge, this was first pointed out in [6]. Another important property follows from considering the $R$-symmetry of the $SO(3)$ theory. In the UV description, the classical $U(1)_R$ symmetry is broken to $\mathbb{Z}_4^{(R)}$ by the gauge-$R$ anomaly (instead of $\mathbb{Z}_8^{(R)}$ for the $SU(2)$ theory), and this $\mathbb{Z}_4^{(R)}$ acts trivially on the Coulomb branch. Since the $SO(3)$ theory can be obtained by gauging the $\mathbb{Z}_2^{[1]}$ one-form symmetry of the $SU(2)$ theory, this betrays a mixed 't Hooft anomaly between $\mathbb{Z}_2^{[1]}$ and $\mathbb{Z}_2 \subset \mathbb{Z}_8^{(R)}$ [23].

Therefore, unlike the $SU(2)$ case, the two strong-coupling singularities on the $SO(3)$ Coulomb branch are not related by a spontaneously-broken $R$-symmetry. This is also apparent from the perspective of the 4d $\mathcal{N} = 1$ supersymmetric gauge theories with $\mathfrak{g} = \mathfrak{su}(2)$ gauge algebra [3], which can be obtained from the $\mathcal{N} = 2$ theories by explicit supersymmetry breaking from $\mathcal{N} = 2$ to $\mathcal{N} = 1$ [4]. In the $\mathcal{N} = 1$ $SU(2)$ gauge theory, the classical $U(1)_R$ symmetry is broken to $\mathbb{Z}_4^{(R)}$ by the ABJ anomaly, and one has two confining vacua related by the $R$-symmetry (equivalently, by a shift $\theta \to \theta + 2\pi$). In the $\mathcal{N} = 2$ theory softly broken to $\mathcal{N} = 1$, the two vacua arise from condensation of either the magnetic monopole or the dyon at the SW points [5]. As a result, the Wilson line in the fundamental representation of $SU(2)$ has an area law (*i.e.* it is confined) in both vacua [3].

By contrast, in the $\mathcal{N} = 1$ $SO(3)$ theory, the ABJ anomaly breaks the $U(1)_R$ symmetry to $\mathbb{Z}_2^{(R)}$ (corresponding to a shift $\theta \to \theta + 4\pi$) and the two vacua are no longer physically equivalent. In particular, allowed line operators will have either an area law or a perimeter law, depending on which of the two vacua is chosen; in the $SO(3)_+$ theory, the monopole point gives us a deconfined $\mathcal{N} = 1$ vacuum with a $\mathbb{Z}_2$ gauge theory in the IR [3]. Thus, the $\mathcal{N} = 2$ $SO(3)$ theory should also have inequivalent CB singularities. Note also that these singularities should be exchanged as we go between $SO(3)_+$ and $SO(3)_-$ as in (28).

Based on the above reasoning, the correct curve for the $SO(3)$ gauge theory is uniquely determined. We are looking for an extremal rational elliptic surface with three singular fibers, one of them being $I_1^*$ (the fiber at infinity, which is fixed by the $\beta$-function of the $SO(3)$ $\mathcal{N} = 2$ gauge theory). As the remaining two singularities must be distinct, this leaves us with only one possibility [24], namely the RES $(I_1^*; I_4, I_1)$, which is also the configuration for the massless $SU(2)$ $N_f = 3$ theory [5]. In this case, we have the monodromies:

$$
\begin{aligned}
SO(3)_+ : \quad & \mathbb{M}_{I_4}^+ = ST^4 S^{-1}, \qquad & \mathbb{M}_{I_1}^+ = (ST^{-2}S)T(ST^{-2}S)^{-1}, \\
SO(3)_- : \quad & \mathbb{M}_{I_1}^- = \mathbb{M}_{I_1}^+, \qquad & \mathbb{M}_{I_4}^- = (TS)T^4(TS)^{-1},
\end{aligned}
\tag{32}
$$

as well as $\mathbb{M}_\infty = -T$, with $\mathbb{M}_\infty^{-1} = \mathbb{M}_{I_4}^+ \mathbb{M}_{I_1}$, for the $SO(3)_+$ curve and $\mathbb{M}_\infty^{-1} = \mathbb{M}_{I_1} \mathbb{M}_{I_4}^-$ for the $SO(3)_-$ curve. These monodromies and the corresponding fundamental domains can be obtained as follows. Starting with the $\Gamma^0(4)$ domain of the $SU(2)$ curve in figure 1, the map (31) between $\tau_{SU(2)}$ and $\tau_{SO(3)_+}$ is realised by the action of the matrix:

$$
\mathbb{M}_0 = \begin{pmatrix} 1 & 0 \\ 0 & 4 \end{pmatrix}.
\tag{33}
$$

Conjugating the generators of $\Gamma^0(4)$ by $\mathbb{M}_0$ leads to the $SO(3)_+$ generators listed above. This argument can be repeated for the $SO(3)_-$ curve. Note, in particular, that in this latter case the width-1 cusp of the $SO(3)_+$ curve is exchanged with the width-4 cusp after conjugation. The corresponding fundamental domains are shown in figure 2a. Note also that the coset representative $TST^2S$ for the width-one cusp at $\tau = \frac{1}{2}$ of $SO(3)_-$ leads to the same monodromy matrix as the above $\mathbb{M}_{I_1}$.

In Weierstrass form, the $SO(3)_\pm$ Seiberg-Witten curves are given by:

$$
\begin{aligned}
g_2^{SO(3)_\pm} &= \frac{u^2}{12} \pm \frac{5\Lambda^2 u}{2} + \frac{11\Lambda^4}{4}, \\
g_3^{SO(3)_\pm} &= -\frac{u^3}{216} \pm \frac{7u^2\Lambda^2}{24} + \frac{29u\Lambda^4}{24} \pm \frac{7\Lambda^6}{8},
\end{aligned}
\tag{34}
$$

with the discriminant:

$$
\Delta^{SO(3)_\pm} = \frac{1}{8}\Lambda^2 \left(u \pm \Lambda^2\right)\left(u \mp \Lambda^2\right)^4.
\tag{35}
$$

Given the explicit map between curves to be explained below, the SW differential of the $SO(3)$ theory is related to the one for the $SU(2)$ theory as:

$$
\lambda_{SO(3)_\pm} = 2\lambda_{SU(2)}.
\tag{36}
$$

We will derive these expressions (including the normalisation) in section 2.2. One can check that the leading term in the large-$u$ expansion of the physical periods reads:

$$
a(u) \approx -2i\sqrt{u}, \qquad a_D(u) \approx -\frac{\sqrt{u}}{\pi}\log\frac{\Lambda^2}{u},
\tag{37}
$$

thus reproducing the $I_1^*$ monodromy as expected. From (34) and (35), we see that the $SO(3)$ $u$-plane has two strong-coupling singularities of type $I_1$ and $I_4$ at $u = \mp\Lambda^2$ and $u = \pm\Lambda^2$, respectively. These two singularities are exchanged as we shift $\theta \to \theta + 2\pi$ (i.e. $\Lambda^2 \to -\Lambda^2$). Note that the $I_4$ singularity is an undeformable singularity in the sense of [9]. This is also expected because, in the $SO(3)$ normalisation of the charges, one of the two SW singularities corresponds to having a massless hypermultiplet of charge 2 coupled to the low-energy photon (in the appropriate duality frame).[4]

---

[4]Recall that an undeformable $I_n$ singularity corresponds to a massless hypermultiplet of charge $\sqrt{n}$.

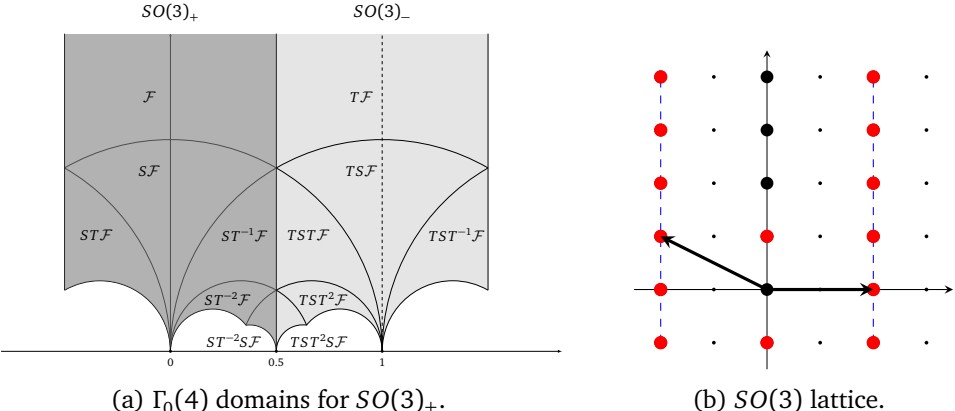

(a) $\Gamma_0(4)$ domains for $SO(3)_\pm$.     (b) $SO(3)$ lattice.

Figure 2: LEFT: Fundamental domains for $\Gamma_0(4)$ on the upper-half-plane $\mathcal{H} \cong \{\tau\}$, for $SO(3)_+$ (left) and $SO(3)_-$ (right), related by $\tau_{SO(3)_-} = \tau_{SO(3)_+} + \frac{1}{2}$. Note that this exchanges the bulk width-1 cusp with the width-4 cusp. RIGHT: The charge lattice $\Gamma$ of the $SO(3)$ theory (small dots), with the allowed dynamical particle charges $\gamma \in \widehat{\Gamma}_{SO(3)}$ shown in black (large dots) and with the weak-coupling BPS spectrum coloured in red. We are using the charge normalisation $(m_\pm, q_\pm)$ for $SO(3)_\pm$.

We also note that the $SO(3)$ Coulomb branch is a modular curve for the modular group $\Gamma_0(4)$, with the fundamental domain shown in figure 2a. Thus, $u = u(\tau)$ becomes a modular function for this congruence subgroup [8, 18], with the $SO(3)_+$ expression given by

$$\frac{u(\tau)}{\Lambda^2} = 1 + \frac{8\vartheta_3(\tau)^2 \vartheta_4(\tau)^2}{\left(\vartheta_3(\tau)^2 - \vartheta_4(\tau)^2\right)^2} = 1 + \frac{1}{8}\left(\frac{\eta(\tau)}{\eta(4\tau)}\right)^8, \tag{38}$$

which can be also found from (19) by performing the transformation $\tau \to \frac{1}{4}\tau$. The $SO(3)_-$ expression is found by sending $\Lambda^2 \to -\Lambda^2$. Alternatively, one can use the relative $\mathfrak{su}(2)$ curve, to be introduced in the next subsection.

**BPS states and BPS lines for $SO(3)_+$.** The global structure of the gauge group should not affect the BPS spectrum, and indeed the BPS quiver remains the same. This is because the change of normalisation (29) leaves the Dirac pairing invariant. In particular,

$$\langle \gamma_M, \gamma_D \rangle = 2, \tag{39}$$

in all cases. For the $SO(3)_+$ theory, the light states at the SW singularities are now:

$$\gamma_M = (2, 0), \qquad \gamma_D = (-2, 1). \tag{40}$$

Here, the monopole corresponds to a light hypermultiplet of charge 2 in the $S$-dual description of the $I_4$ singularity, as anticipated, while the dyon that becomes light at the $I_1$ singularity still corresponds to a charge-1 hypermultiplet in the appropriate duality frame.

As before, the set of allowed BPS states form a sublattice $\widehat{\Gamma}_{SO(3)_+}$ of the magnetic-electric lattice $\Gamma \cong \mathbb{Z}^2$, and the allowed BPS lines have charges $\gamma_L$ such that:

$$\langle \gamma_L, \gamma \rangle \in \mathbb{Z}, \qquad \forall \gamma \in \widehat{\Gamma}_{SO(3)_+}, \qquad \text{and} \qquad \gamma_L \in \Gamma. \tag{41}$$

We then find that the lines $(0, 0)$ and $(1, 0)$ cannot be screened. Correspondingly, we have a magnetic one-form symmetry:

$$\Gamma^{[1]} \equiv \mathrm{Hom}\left(\Gamma/\widehat{\Gamma}_{SO(3)_+}, U(1)\right) = \mathbb{Z}_2^{[1]}, \tag{42}$$

which is generated by the element $g^{\mathbb{Z}_2^{[1]}} = \left(\frac{1}{2}, 0\right)$, similarly to (26).

**BPS states and BPS lines for $SO(3)_-$.** Similar comments hold in the case of the $SO(3)_-$ gauge theory. The SW singularities now correspond to the light states:

$$\gamma_M = (2, -1), \qquad \gamma_D = (-2, 2). \tag{43}$$

Here, we see that the 'monopole' is actually a dyon and corresponds to the $I_1$ singularity, while the light dyon $(-2, 2)$ gives rise to the $I_4$ singularity. In the $SO(3)_-$ normalisation of the charges, $\gamma = (m_-, q_-) \in \Gamma$, we again have that all the allowed lines are simply $\gamma_L \in \Gamma$, and that the line $(1, 0)$ cannot be screened. Importantly, this exactly corresponds to the line $(\lambda_m, \lambda_e) = (1, 1)$ in the notation of [3], as explained in appendix B.

**Torsion section and one-form symmetry.** The $SO(3)_\pm$ curve has torsion MW group:

$$\Phi(\mathcal{S}) = \Phi_{\text{tor}}(\mathcal{S}) = \mathbb{Z}_4, \qquad P_{\mathbb{Z}_2}^\pm = \left( -\frac{u}{6} \mp \frac{\Lambda^2}{2}, 0 \right), \tag{44}$$

with the additional $\mathbb{Z}_4$ sections:

$$P_{\mathbb{Z}_4} = P_1 = \left( \frac{u}{12} - \frac{3\Lambda^2}{4}, \frac{iu\Lambda}{\sqrt{2}} - \frac{i\Lambda^3}{\sqrt{2}} \right), \qquad P_3 = -P_1, \qquad 2P_{\mathbb{Z}_4} = P_{\mathbb{Z}_2}^+, \tag{45}$$

for $SO(3)_+$, and similarly for the $SO(3)_-$ curve. The $\mathbb{Z}_2 \subset \mathbb{Z}_4$ subgroup corresponds to the one-form symmetry under which the magnetic line $\gamma_L = (m_\pm, q_\pm) = (1, 0)$ is charged. We will give a physical interpretation of $P_{\mathbb{Z}_4}$ in section 2.3.

### 2.1.3 The relative $\mathfrak{su}(2)$ curve

Historically speaking, the very first Seiberg-Witten curve for the $\mathfrak{su}(2)$ gauge theory, as presented in [4], was neither the $SU(2)$ nor the $SO(3)$ curve discussed above, but rather a third curve given by:

$$y^2 = (x - \Lambda^2)(x + \Lambda^2)(x + u), \tag{46}$$

so that the Coulomb branch is a modular curve for $\Gamma(2)$, with [7, 8, 18]:

$$\frac{u(\tau)}{\Lambda^2} = \frac{2\vartheta_3(\tau)^4 - \vartheta_2(\tau)^4}{\vartheta_2(\tau)^4} = 1 + \frac{1}{8} \left( \frac{\eta\left(\frac{\tau}{2}\right)}{\eta(2\tau)} \right)^8. \tag{47}$$

Note, in particular, that $\tau_{\mathfrak{su}(2)} = \frac{1}{2}\tau_{SU(2)} = 2\tau_{SO(3)_+} = 2\tau_{SO(3)_-} - 1$.[5] This corresponds to yet another normalisation of the charges, with:

$$\widehat{\gamma} = (\widehat{m}, \widehat{q}) = \left( m, \frac{1}{2}q \right) = \left( \frac{1}{2}\lambda_m, \frac{1}{2}\lambda_e \right), \tag{48}$$

where $(m, q)$ are the $SU(2)$ charges, and $(\lambda_m, \lambda_q)$ denotes the weight basis of [3] reviewed in appendix B. In particular, the W-boson now has charge $\widehat{\gamma}_W = (0, 1)$. Additional care is needed to interpret this charge normalisation correctly. The basic issue is that the charges $\widehat{\gamma}$ corresponds exactly to the 'allowed charge lattice' $\widehat{\Gamma}$, and as such it does not allow for a maximal consistent set of BPS lines.

---

[5]Note that to obtain the $SO(3)_-$ version of (38), one can first do a **T**-transformation on the relative curve (47), and then rescale $\tau$ by a factor of 2, as indicated.

**The Seiberg-Witten curve.** By a simple change of coordinates,

$$x \to 2x - \frac{u}{3}, \qquad y \to \sqrt{2}y, \tag{49}$$

we bring (46) to the Weierstrass normal form. We have:

$$g_2^{\mathfrak{su}(2)} = \frac{u^2}{3} + \Lambda^4, \qquad g_3^{\mathfrak{su}(2)} = -\frac{u^3}{27} + \frac{u\Lambda^4}{3}, \tag{50}$$

with the discriminant:

$$\Delta^{\mathfrak{su}(2)} = \Lambda^4 \left(u^2 - \Lambda^4\right)^2. \tag{51}$$

We shall call this particular SW curve *the relative $\mathfrak{su}(2)$ curve*. Note that we chose a particular rescaling of the $(x, y)$ coordinates in (49), for reasons that will become clear momentarily.

The relative $\mathfrak{su}(2)$ curve has an $I_2^*$ singularity at infinity, which is the expected monodromy given the charge normalisation (48). The strong coupling singularities are now two undeformable $I_2$ singularities. This means that the low-energy description at each SW singularity should be in terms of a light hypermultiplet coupled to the low-energy photon with an electric charge $\sqrt{2}$. In the normalisation (48), we have the monopole $\widehat{\gamma}_M = (1, 0)$ and the dyon $\widehat{\gamma}_M = (-1, 1)$ becoming massless at the SW singularities, which seems to be a contradiction. In fact, $\widehat{\Gamma}$ should be identified with the homology lattice of the elliptic curve, with the allowed states corresponding to homology 1-cycles $[\widehat{\gamma}] = \widehat{m}\gamma_B + \widehat{q}\gamma_A$, but the correct Dirac pairing is related to the homology pairing by a factor of 2. We then have:

$$\langle \widehat{\gamma}_1, \widehat{\gamma}_2 \rangle = 2(\widehat{m}_1 \widehat{q}_2 - \widehat{q}_1 \widehat{m}_2) = 2[\widehat{\gamma}_1] \cdot [\widehat{\gamma}_2], \tag{52}$$

to be compared to (16). The Dirac pairing is a physical choice of polarisation of the elliptic curve [13, 14]. Here, the polarisation is fixed once we require that we should have the same BPS quiver (and thus the same BPS spectrum) as for the $SU(2)$ curve. Therefore, the relative $\mathfrak{su}(2)$ curve is not principally polarised (unlike the $SU(2)$ and $SO(3)$ SW curves, which are principally polarised). Finally, let us define the physical charge:

$$\widetilde{\gamma} \equiv \sqrt{2}\widehat{\gamma} = \frac{1}{\sqrt{2}}(2m, q). \tag{53}$$

In this normalisation, we indeed find the monopole and dyons:

$$\widetilde{\gamma}_M = \sqrt{2}(1, 0), \qquad \widetilde{\gamma}_D = \sqrt{2}(-1, 1), \tag{54}$$

with the 'correct' Dirac pairing $\langle \widetilde{\gamma}_M, \widetilde{\gamma}_D \rangle = 2$, and as expected from the presence of $I_2$ singularities.

**Torsion section and defect group.** The relative $\mathfrak{su}(2)$ curve (50) takes the simple form:

$$y^2 = 4(x - x_1)(x - x_3)(x - x_3), \tag{55}$$

with the rational roots:

$$x_1 = -\frac{u}{3}, \qquad x_2 = \frac{u}{6} + \frac{\Lambda^2}{2}, \qquad x_3 = \frac{u}{6} - \frac{\Lambda^2}{2}, \tag{56}$$

which satisfy $x_1 + x_2 + x_3 = 0$. It follows that we have three torsion sections of order 2:

$$P_i = (x_i, 0), \qquad i = 1, 2, 3, \tag{57}$$

which satisfy:

$$P_i + P_j = \begin{cases} \mathcal{O}, & \text{if } i = j, \\ P_k, & \text{if } i \neq j, \ i \neq k \neq j, \end{cases} \tag{58}$$

with $\mathcal{O}$ the zero-section. We also use the notation $P_+ = P_2$, $P_- = P_3$ (and $x_+ = x_2$, $x_- = x_3$). These sections generate the Mordell-Weil group:

$$\Phi(\mathcal{S}) = \mathbb{Z}_2 \oplus \mathbb{Z}_2. \tag{59}$$

While we identified (part of) the Mordell-Weil group of the 'absolute' curves (for $SU(2)$ and $SO(3)_\pm$) with the corresponding $\mathbb{Z}_2^{[1]}$ one-form symmetries, we shall identify the MW group of the relative $\mathfrak{su}(2)$ curve with the defect group of the $\mathfrak{su}(2)$ gauge theory. We will give more evidence for this identification in the following.

The four sections $P \in \mathbb{Z}_2 \oplus \mathbb{Z}_2$ (including the zero-section) correspond to the four points $z = 0, \frac{1}{2}, \frac{\tau}{2}, \frac{\tau+1}{2}$ on the elliptic fiber, as we will discuss in more detail below. It is interesting to note that the modular group $\Gamma(2)$ of the relative $\mathfrak{su}(2)$ curve is the kernel of the homomorphism $\mathrm{SL}(2,\mathbb{Z}) \to \mathrm{SL}(2,\mathbb{Z}_2) \cong S_3$, where $S_3$ exchanges the three points $P_i \neq \mathcal{O}$ [25]. This interpretation of $\Gamma(2)$ will be particularly relevant when discussing the relative $\mathcal{N} = 2^*$ curve in section 3.

## 2.2 Gauging one-form symmetries along torsion sections

We just discussed how different choices of the SW curve for the pure $\mathfrak{su}(2)$ gauge theory determine the normalisation of the BPS spectrum, and hence determine the set of allowed line operators. Let us now give a concrete prescription to explicitly gauge any one-form symmetry at the level of the SW curve. We will do this in the form of three conjectures that are inspired by elementary properties of isogenies (see appendix A.2 for further mathematical background). These conjectures allow us, in particular, to derive the $SO(3)_\pm$ curves from the $SU(2)$ curve, and vice versa.

### 2.2.1 Isogenies, torsion sections and gauging: General conjectures

The torsion rational sections in the Mordell-Weil group $\Phi = \mathrm{MW}(\mathcal{S})$ conjecturally determine the one-form symmetry of the underlying rank-one 4d $\mathcal{N} = 2$ theory [7]. In that previous work, we only studied Coulomb branches with maximally-deformable singularities; in that case, the fully mass-deformed SW curve only had $I_1$ singularities. In the present work, we must allow for more general deformation patterns – the physically-allowed deformation patterns were classified in [9]. Then, the generic mass deformation of the curve contains a number of undeformable singularities (such as the $I_4$ singularity of the $SO(3)$ curve given above), but heuristic arguments suggests that, in all cases, the one-form symmetry is encoded in the torsion sections that do not 'interact' with the flavour symmetry.[6] In a general rank-1 theory, we have a distinct rational elliptic surfaces at any fixed value of the relevant and/or marginal couplings of the theory. Let $\mathcal{S}_{\mathrm{mass}}$ denote the fully mass-deformed SW geometry, at generic values of the masses (and/or marginal couplings). We then propose the following conjectures:

**Conjecture I. (Defect group.)**  *The Seiberg-Witten curve of any relative rank-one theory $\mathcal{T}_{\mathrm{rel}}$ is given by a non-principally-polarised elliptic curve. In this case, the defect line group of the theory is isomorphic to the torsion part of the MW group of the mass-deformed curve:*

$$\mathbb{D} \cong \Phi_{\mathrm{tor}}(\mathcal{S}_{\mathrm{mass}}). \tag{60}$$

---

[6]At the massless point, the flavour symmetry is encoded in the (deformable) singularities on the Coulomb branch. At generic masses, the flavour group is broken down to a maximal torus $U(1)^f$, and the corresponding $U(1)$ background vector multiplets arise from the free generators of the Mordell-Weil group, $\Phi_{\mathrm{free}} \cong \mathbb{Z}^f$ [7,11].

In all cases, this corresponds to $\mathbb{D} \cong \mathbb{Z}_N \oplus \mathbb{Z}_N$ for $N = 2$ or $N = 3$.

**Conjecture II. (One-form symmetry.)** *The Seiberg-Witten curve of any absolute rank-one theory $\mathcal{T}$ must be a principally-polarised elliptic curve. Then, the one-form symmetry $\Gamma^{[1]}$ of $\mathcal{T}$ is isomorphic to a subgroup of the torsion part of the Mordell-Weil group of the mass-deformed curve:*

$$\Gamma^{[1]} \subseteq \Phi_{\text{tor}}(\mathcal{S}_{\text{mass}}). \tag{61}$$

*This subgroup can be identified by relating the absolute curve to the relative curve, as we will explain momentarily.*

Let us consider an absolute theory $\mathcal{T}$ with a one-form symmetry $\Gamma^{[1]} = \mathbb{Z}_N^{[1]}$. Then, by Conjecture II, its SW curve $E$ admits a rational section $P_{\mathbb{Z}_N}$ of order $N$. Any such torsion section defines an automorphism $t_{P_{\mathbb{Z}_N}}$ of the RES $\mathcal{S}$, corresponding to translation by $P_{\mathbb{Z}_N}$ along every smooth fiber. Then, the quotient $\mathcal{S}/\langle t_P \rangle$ defines an $N$-to-1 homomorphism on the smooth fibers, which extends in a well-understood fashion to the singular fibers. (See appendix A.2 for further details.) This $N$-isogeny along the torsion section of the curve $E$ therefore gives us the new Seiberg-Witten curve:

$$E_{\text{rel}} = E/\langle t_{P_{\mathbb{Z}_N}} \rangle. \tag{62}$$

We claim that this 'relative curve' is the SW curve for the relative version of the theory $\mathcal{T}$. In particular, we expect that, for generic masses, the torsion sections of this relative curve span the group:

$$\Phi_{\text{tor}}(\mathcal{S}_{\text{mass}}) \cong \mathbb{Z}_N \oplus \mathbb{Z}_N, \tag{63}$$

so that Conjecture I holds true. Now, consider any torsion section $P'$ of the relative curve that generates a $\mathbb{Z}_N$ subgroup of (63). By performing a further $N$-isogeny along this section, we obtain a new principally-polarised SW curve:

$$E' = E_{\text{rel}}/\langle t_{P'} \rangle. \tag{64}$$

For a particular choice of 'inverse' isogeny, we obtain the curve of the theory $\mathcal{T}$ we started with, while more generally we obtain a new, distinct, curve for a different absolute theory with the same local dynamics. The relation (64) identified the subgroup (61) as the one generated by the torsion section in $E = E'$ that generates the dual isogeny.

**Preserving the Dirac pairing: Absolute and relative curves.** The isogeny defined by a quotient along a torsion section $P$,

$$\psi : E \mapsto E/\langle t_P \rangle, \tag{65}$$

preserves the holomorphic one-form $\frac{dx}{y}$. At the same time, it rescales one particular linear combination of the geometric periods $\omega_a$, $\omega_D$ defined in (11) by a factor of $1/N$. It follows that the Dirac pairing defined as in (16) is rescaled by a factor of $N$. On the other hand, we always have the freedom of rescaling the curve $E$ by an overall factor of $\alpha \in \mathbb{C}^*$ – that is, we may rescale the two periods as $(\omega_D, \omega_a) \rightarrow (\alpha\omega_D, \alpha\omega_a)$, which amounts to the rescaling $(g_2, g_3) \rightarrow (\alpha^{-4}g_2, \alpha^{-6}g_3)$, in which case the new Dirac pairing after the $N$-isogeny becomes:

$$\langle \gamma_1, \gamma_2 \rangle \rightarrow \frac{N}{\alpha^2} \langle \gamma_1, \gamma_2 \rangle. \tag{66}$$

We can therefore preserve the Dirac pairing if and only if we set $\alpha = \sqrt{N}$. This is what we will do in the following: The isogenies (62) and (64) will henceforth be understood as a composition of the quotient isogeny (65) with the rescaling by $\alpha = \sqrt{N}$. Note that, given

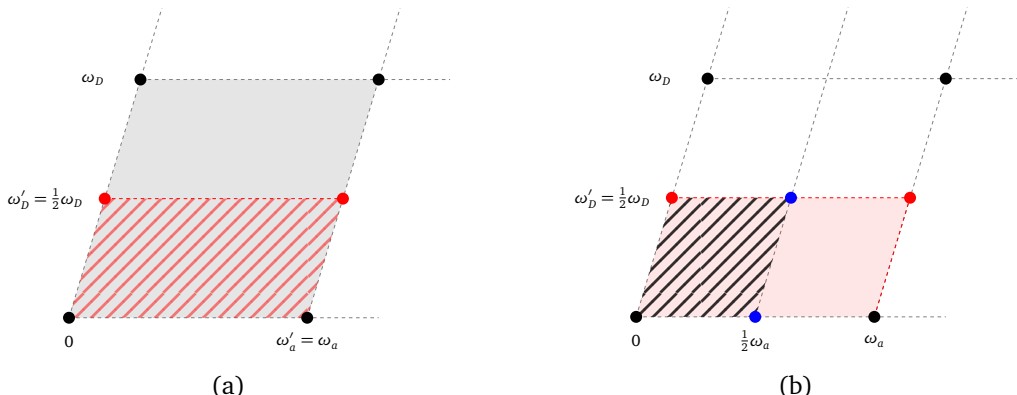

Figure 3: 2-isogenies of complex tori. ($a$) 2-Isogeny $\psi : \mathbb{C}/L_\tau \to \mathbb{C}/L'_{\tau'}$, with $\tau' = \frac{\tau}{2}$ realized as a cyclic quotient map on $L_\tau$ lattice, on the $\tau$ direction. ($b$) The dual isogeny $\hat\psi : \mathbb{C}/L'_{\tau'} \to \mathbb{C}/L_\tau$ is realised as a cyclic quotient map along the real axis, leading to a new lattice isomorphic to $L_\tau$ upon expansion to standard basis.

a principally polarised SW curve for an absolute theory, the rescaling then implies that the relative curve is not principally polarised.[7]

Given this discussion, we can state our conjecture about how the gauging of a one-form symmetry affects the SW geometry:

**Conjecture III. (Gauging $\Gamma^{[1]}$.)** *At the level of the rank-one SW geometry, the gauging of a one-form symmetry $\Gamma^{[1]} = \mathbb{Z}_N^{[1]}$ is the composition of two $N$-isogenies generated by torsion sections of order $N$:*

$$E(\mathcal{T}) \xdashrightarrow{\quad N\text{-isogeny}\quad} E_{\text{rel}} \xdashrightarrow{\quad N\text{-isogeny}\quad} E\left(\mathcal{T}/\Gamma^{[1]}\right). \tag{67}$$

In general, there can be several consistent 'gaugings' of $E(\mathcal{T}/\Gamma^{[1]})$, as in the case of the $SO(3)_\pm$ theories, which is reflected in the choice of a torsion section in the relative curve. As we just explained, these $N$-isogenies include a rescaling by $\alpha = \sqrt{N}$ in order to preserve the Dirac pairing.

### 2.2.2 Gauging along 2-isogenies: Relations amongst the $\mathfrak{su}(2)$ gauge theories

Let us now focus on the case $\Gamma^{[1]} = \mathbb{Z}_2$, which is the one relevant for the pure $\mathfrak{su}(2)$ gauge theories. In Weierstrass normal form, any $\mathbb{Z}_2$ torsion section takes the form $P = (x_0, 0)$, where $x_0$ is a rational root of the polynomial $f(x) \equiv 4x^3 - g_2 x - g_3$. Hence, if we have any such rational sections, we have either one such section or three of them. The 2-isogeny along $P$ is given explicitly by the Vélu formula (reviewed in appendix A.2), which tells us that the new curve $E/\langle t_P \rangle$ is given by:

$$g'_2 = -g_2 + 15x_0^2, \qquad g'_3 = \frac{1}{8}\left(g_3 - 7g_2 x_0 + 84x_0^3\right). \tag{68}$$

Here, we included the rescaling by $\alpha = \sqrt{2}$, as discussed above. Using this formula, it is straightforward to check that the $SU(2)$, $SO(3)_\pm$ and $\mathfrak{su}(2)$ curves are all related by such 2-isogenies. In the rest of this section, we discuss these relations in some more detail.

---

[7]Any elliptic curve admits a principal polarisation. What we are saying is that the principal polarisation of the 'relative curve', which would correspond to $\alpha = 1$, does not give us the physical Dirac pairing. This agrees with the general discussion in [13].

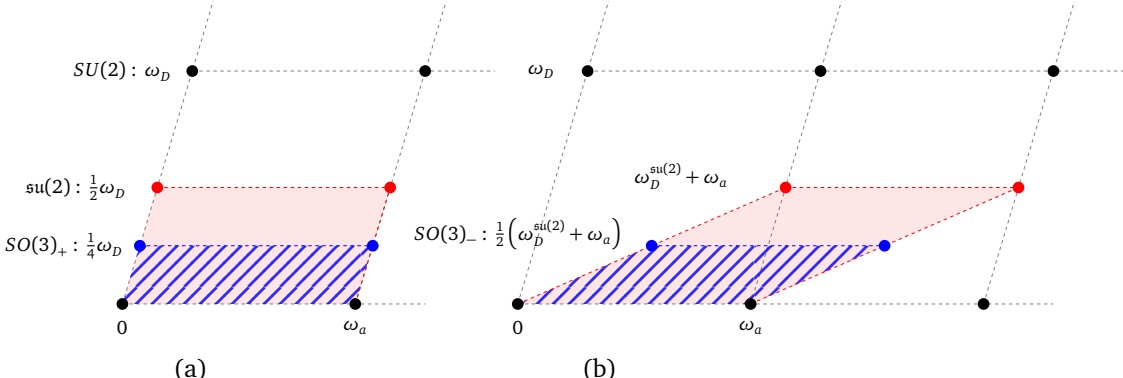

Figure 4: Isogenies of complex tori associated to the $SU(2)$, $\mathfrak{su}(2)$ and $SO(3)_\pm$ curves. The fundamental domain of the $L_{\mathfrak{su}(2)}$ lattice is depicted in red, while those of the $L_{SO(3)_\pm}$ lattices correspond to the dashed regions in (a) and (b), respectively.

**The $SU(2)$ curve.** Let us start with the $\Gamma^0(4)$ curve for the 4d pure $SU(2)$ theory, with the Weierstrass form (8). The rational elliptic surface has a $\mathbb{Z}_2$ torsion section (27), corresponding to the rational root $x_0 = \frac{u}{3}$ of $f(x)$. Note that there are 2 more distinct roots:

$$x_\pm = \frac{1}{6}\left(-u \pm 3\sqrt{u^2 - \Lambda^4}\right). \tag{69}$$

On general grounds, these three roots must correspond to the values of the Weierstrass $\wp$ function at the half-periods of the associated lattice $L = \mathbb{Z}\,\omega_a + \mathbb{Z}\,\omega_D$ spanned by the geometric periods. (See appendix A.3 for a review of relevant facts about elliptic curve, and equation (A.29) in particular.) Recalling that $\tau = \frac{\omega_D}{\omega_a}$ by definition, we can check that $P_{\mathbb{Z}_2} = (x_0, 0)$ maps to the half-period $\frac{1}{2}\omega_D$ on the complex plane:

$$\wp\left(\frac{\omega_D}{2}\right) = \frac{u}{3}. \tag{70}$$

It is then particularly easy to visualise the 2-isogeny along $P_{\mathbb{Z}_2}$ on the torus $\mathbb{C}/L$, since it simply corresponds to a further quotienting by that half-period, as shown in figure 3. Note that, under this 2-isogeny, the effective gauge coupling of the $SU(2)$ theory maps to:

$$\tau_{SU(2)} \mapsto \tau_{\mathfrak{su}(2)} = \frac{\tau_{SU(2)}}{2}, \tag{71}$$

irrespective of the choice of rescaling parameter $\alpha$.

**The $\mathfrak{su}(2)$ curve.** One easily checks that the 2-isogeny along the $\mathbb{Z}_2$ torsion section of the $SU(2)$ curve (8) gives us precisely the relative $\mathfrak{su}(2)$ curve (50). Keeping the rescaling parameter $\alpha$ arbitrary, we obtain the quotient curve:

$$g_2(u) = 4\alpha^{-4}\, g_2^{\mathfrak{su}(2)}(u), \qquad g_3(u) = 8\alpha^{-6}\, g_3^{\mathfrak{su}(2)}(u). \tag{72}$$

Here, we have the new periods $\alpha\left(\frac{\omega_D}{2}, \omega_a\right)$, which means that the magnetic-electric charges $(m, q)$ of the $SU(2)$ theory are mapped to $\frac{1}{\alpha}(2m, q)$, and we then recover (53) by setting $\alpha = \sqrt{2}$. Let $L_{\mathfrak{su}(2)}$ denote the period lattice of the relative curve. The $\mathbb{Z}_2 \oplus \mathbb{Z}_2$ torsion sections (57) correspond to the three half-periods. In this case, we have:

$$\wp\left(\frac{\omega_a^{\mathfrak{su}(2)}}{2}\right) = x_1, \qquad \wp\left(\frac{\omega_D^{\mathfrak{su}(2)}}{2}\right) = x_2, \qquad \wp\left(\frac{\omega_a^{\mathfrak{su}(2)} + \omega_D^{\mathfrak{su}(2)}}{2}\right) = x_3, \tag{73}$$

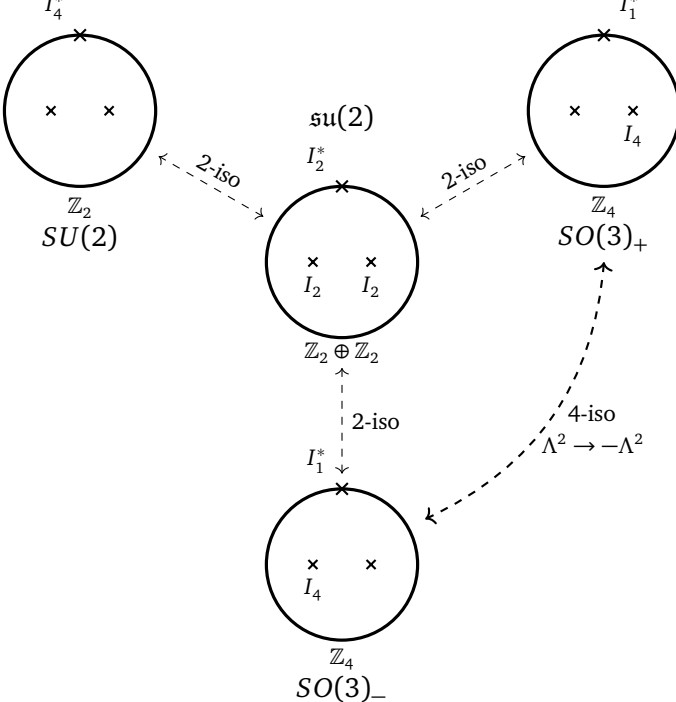

Figure 5: Summary of the relations between the relative $\mathfrak{su}(2)$ curve (in the middle) and the three absolute curves. Here each circle represents the Coulomb branch with its singularity structure (unlabelled crosses are $I_1$ singularities). The torsion MW group $\Phi_{\text{tor}}(\mathcal{S})$ is indicated under each CB. The direct relation between $SO(3)_+$ and $SO(3)_-$ is either through a 4-isogeny or through the explicit substitution $\Lambda^2 \to -\Lambda^2$. The two operations can be combined as discussed in subsection 2.3.

with $x_i$ given in (56). Performing a quotient of the generic fiber along any of the three $\mathbb{Z}_2$ section is again straightforward. For instance, the quotient along $P_1$ gives us back the $SU(2)$ curve, since it now corresponds to a quotient along the half-period $\frac{\omega_a}{2}$, as shown in figure 3.

Let us also note that the section $P_1$ intersects both $I_2$ singular fibers of the SW geometry non-trivially, while $P_2$, $P_3$ each intersect a distinct $I_2$ fibre. According to general results on isogenies (see appendix A.2), this implies that the quotient along $P_1$ gives us back the two $I_1$ singularities, as expected, while the quotient along $P_2$ or $P_3$ gives us an $I_1$ and an $I_4$ singularity. This of course corresponds to the $SO(3)$ curves discussed above. The relations between the different curves are shown in figure 5.

**The $SO(3)_\pm$ curves.** The isogeny along the $P_+ = P_2$ section gives us a quotient along the half-period $\frac{1}{2}\omega_D^{\mathfrak{su}(2)}$. This leads to a period lattice $L_{SO(3)_+}$ with $\tau^{SO(3)} = \frac{1}{2}\tau^{\mathfrak{su}(2)} = \frac{1}{4}\tau_{SU(2)}$, as depicted in figure 4. Using the explicit 2-isogeny (68), we exactly recover the principally-polarised $SO(3)_+$ curve (34). Note that this indeed leads to the $SO(3)_+$ periods given in (30).

Next, let us consider the isogeny along $P_- = P_3$, which is a further quotient of the relative curve $\mathbb{C}/L_{\mathfrak{su}(2)}$ quotient along the 'diagonal' half-period $\frac{1}{2}\omega_3^{\mathfrak{su}(2)}$, with $\omega_3^{\mathfrak{su}(2)} \equiv \omega_a^{\mathfrak{su}(2)} + \omega_D^{\mathfrak{su}(2)}$. To perform this quotient, it is most convenient to first perform a $T$ transformation on the $L_{\mathfrak{su}(2)}$ lattice, to obtain a shifted lattice $\widetilde{L}_{\mathfrak{su}(2)}$ with:

$$\tau_{\mathfrak{su}(2)} \mapsto \tilde{\tau}_{\mathfrak{su}(2)} = \tau_{\mathfrak{su}(2)} + 1 = \frac{\omega_D^{\mathfrak{su}(2)} + \omega_a^{\mathfrak{su}(2)}}{\omega_a^{\mathfrak{su}(2)}} = \frac{\omega_3^{\mathfrak{su}(2)}}{\omega_a^{\mathfrak{su}(2)}}. \tag{74}$$

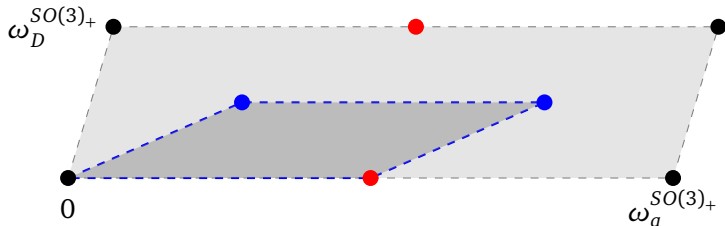

Figure 6: Explicit depiction of the 4-isogeny on the generic fiber of the $SO(3)_+$ curve, with the $\mathbb{Z}_4$ translation generated by the blue dots, and $P_{\mathbb{Z}_2} = 2P_{\mathbb{Z}_4}$ generated by the red dot. The result of the 4-isogeny is shown in dark grey. After a rescaling by $\alpha = 2$, this gives us the $SO(3)_-$ curve.

The quotient by $\frac{1}{2}\omega_3^{\mathfrak{su}(2)}$ is then straightforward, as shown in figure 4. We can also check that the explicit 2-isogeny (68) leads precisely to the $SO(3)_-$ curve given in (34). Note that the shift (74) also matches with (31).

Note that the charge lattice $\Lambda \cong \mathbb{Z}^2$ spanned by $\lambda \equiv (\lambda_m, \lambda_e)$ is not realised by any of the homology lattices of these curves. Instead, the homology lattice of the relative $\mathfrak{su}(2)$ curve only allows for the states $(\lambda_m, \lambda_e) \in 2\mathbb{Z} \oplus 2\mathbb{Z}$, according to (48). In other words, we see that the lattice $\Lambda$ corresponds to the lattice of 'half-periods' of the relative curve $E_{\text{rel}}$. As we perform any 2-isogeny along a $\mathbb{Z}_2$ torsion section $P$, some half-period becomes a period in the new curve $E = E_{\text{rel}}/\langle t_P \rangle$. In this sense, once we accept Conjecture I above, the mathematics of isogenies completely parallels the standard discussion of allowed defect lines [3] – for rank-1 SW geometry, reading between the lines is equivalent to reading between the rational sections.

## 2.3 $\mathbb{Z}_4$ torsion and non-invertible symmetry in the $SO(3)$ gauge theory

Let us further investigate the $\mathbb{Z}_4$ torsion sections of the $SO(3)$ curves. It turns out that a 4-isogeny (including a rescaling $\alpha = \sqrt{4}$) along the torsion section (45) exchanges the two $SO(3)_\pm$ curves:

$$\psi_{\mathbb{Z}_4} : E(SO(3)_+) \mapsto \sqrt{4} \circ E(SO(3)_+)/\langle t_{P_{\mathbb{Z}_4}} \rangle = E(SO(3)_-), \tag{75}$$

as shown in figure 5. This is depicted in figure 6. We have also seen that a shift of the $\theta$-angle by $2\pi$ (corresponding to $\Lambda^2 \to -\Lambda^2$) exchanges the two $SO(3)$ curves. In particular:

$$\theta \to \theta + 2\pi : E(SO(3)_-) \mapsto E(SO(3)_+). \tag{76}$$

Hence, the composition of (76) with (75) is a symmetry of the $SO(3)_+$ curve, which we denote by:

$$\hat{\mathcal{N}} \equiv (\theta \to \theta + 2\pi) \circ \psi_{\mathbb{Z}_4}. \tag{77}$$

We should therefore interpret this action in terms of the physics of the $SO(3)_+$ gauge theory.

As mentioned above, the $SU(2)$ $\mathcal{N} = 2$ gauge theory has a mixed anomaly between the $\mathbb{Z}_2^{[1]}$ one-form symmetry and a $\mathbb{Z}_2^{[0]}$ subgroup of the $\mathbb{Z}_8$ $R$-symmetry. The corresponding 5d anomaly theory takes the form [23]:

$$\frac{i\pi}{2} \int_{\mathcal{M}_5} \mathbf{z} \cup \mathcal{P}(B), \tag{78}$$

where $\mathbf{z}$ is a $\mathbb{Z}_8$ background gauge field, while $B$ is the $\mathbb{Z}_2^{[1]}$ background gauge field and $\mathcal{P}(B)$ is its Pontryagin square. When gauging $\mathbb{Z}_2^{[1]}$ to obtain the $SO(3)_+$ theory, we loose the action of

the spontaneously-broken $\mathbb{Z}_2^{[0]}$ in the infrared description (indeed, the $SO(3)$ Coulomb branch geometry is not symmetric under $u \to -u$). Interestingly, due to the particular form the of the mixed anomaly (78), the $\mathbb{Z}_2^{[0]}$ symmetry of the $SU(2)$ theory implies the existence of a non-invertible symmetry of the $SO(3)_+$ $\mathcal{N} = 2$ gauge theory [26], denoted by $\mathcal{N}$. We propose that the action $\widehat{\mathcal{N}}$ defined in (77) is precisely the imprint of this non-invertible symmetry $\mathcal{N}$ into the Mordell-Weil group of the $SO(3)_+$ curve (and similarly for the $SO(3)_-$ theory).

## 3 The 4d $\mathcal{N} = 2^*$ $\mathfrak{su}(2)$ curves

Another interesting example of a 4d $\mathcal{N} = 2$ gauge theory with a $\mathbb{Z}_2$ one-form symmetry is the $\mathcal{N} = 2^*$ theory, namely the $\mathfrak{su}(2)$ vector multiplet coupled to one massive adjoint hypermultiplet. In the massless limit, this gives us the 4d $\mathcal{N} = 4$ SYM theory with gauge group $\mathfrak{su}(2)$, which is superconformal. The massless curve has the singularity structure $(I_0^*; I_0^*)$.

In the classification programme of 4d $\mathcal{N} = 2$ SCFTs of [9, 27–29], there are three SW geometries with this massless limit. The first one has a fully-deformable $I_0^*$ in the bulk (with the deformation pattern being $I_0^* \to 6I_1$). It corresponds to the $SU(2)$ $N_f = 4$ gauge theory [5]; see [30] for a recent detailed analysis of that SW geometry. The other two deformation patterns correspond to the $\mathcal{N} = 2^*$ theory, having only three singularities in the maximally-deformed phase; two of those correspond to the monopole and dyon points of the pure $\mathcal{N} = 2$ SYM theory, with the additional one being due to the adjoint hypermultiplet becoming light. The two distinct SW geometries have deformation patterns $I_0^* \to 2I_1, I_4$ and $I_0^* \to 3I_2$, respectively. Crucially, they are related to one another by 2-isogenies. This gives us another instance of the general discussion of section 2.2: The $(I_0^*; 3I_2)$ SW geometry corresponds to the relative $\mathfrak{su}(2)$ $\mathcal{N} = 2^*$ curve, while the $(I_0^*, 2I_1, I_4)$ corresponds to the absolute $\mathcal{N} = 2^*$ curve. This interpretation was first discussed in [13]. Here, we further elaborate on this crucial distinction. In particular, we explain the explicit relation between the curves and we expound on the action of $S$-duality in the presence of the mass deformation.

### 3.1 The relative $\mathfrak{su}(2)$ $\mathcal{N} = 2^*$ curve

The $(I_0^*; 3I_2)$ $\mathcal{N} = 2^*$ geometry was originally discussed in [4], where it was written as:

$$y^2 = \prod_{i=1}^{3} \left( x + e_i(\tau_{\text{uv}})\tilde{u} - \frac{1}{4} e_j^2(\tau_{\text{uv}}) m^2 \right), \qquad \tilde{u} = u + \frac{1}{8} e_1(\tau_{\text{uv}}) m^2. \qquad (79)$$

Here, the CB parameter $u$ is defined as in (7) for the pure $SU(2)$ $\mathcal{N} = 2$ theory, and $e_i \equiv e_i(\tau_{\text{uv}})$ are modular forms of the marginal UV coupling, $\tau_{\text{uv}}$, with the property that $\sum_{i=1}^{3} e_i = 0$. Various useful identities for these modular forms are collected in appendix A.4. The shifted Coulomb branch parameter $\tilde{u}$ is invariant under the $\text{SL}(2, \mathbb{Z})$ action of $S$-duality on the UV gauge coupling, $\tau_{\text{uv}}$, of the $\mathcal{N} = 4$ theory.

By a simple change of coordinates, we can bring this curve to Weierstrass normal form. Here, it is important to rescale the curve by a factor of $\alpha = \sqrt{2}$, similarly to the discussion in section 2.1.3. We then have found our *relative $\mathfrak{su}(2)$ $\mathcal{N} = 2^*$ curve*:

$$y^2 = 4 \prod_{i=1}^{3} (x - \widehat{f}_i), \qquad \widehat{f}_i \equiv -\frac{1}{2} e_i \tilde{u} + \frac{1}{8} m^2 \left( e_i^2 - \frac{1}{3} \sum_{k=1}^{3} e_k^2 \right), \qquad (80)$$

where the roots $\widehat{f}_i$ obviously satisfy $\sum_{i=1}^{3} \widehat{f}_i = 0$. This can be written in terms of Eisenstein

series $E_k$ in $q_{\mathrm{uv}} = e^{2\pi i \tau_{\mathrm{uv}}}$, as follows:

$$
\begin{aligned}
g_2(\widetilde{u}) &= \frac{1}{27}\left( 9E_4\widetilde{u}^2 - \frac{3}{2}E_6 m^2 \widetilde{u} + \frac{1}{16}E_4^2 m^4 \right), \\
g_3(\widetilde{u}) &= -\frac{1}{729}\left( 27E_6\widetilde{u}^3 - \frac{27}{4}E_4^2 m^2 \widetilde{u}^2 + \frac{9}{16}E_4 E_6 m^4 \widetilde{u} - \frac{1}{64}(2E_6^2 - E_4^3)m^6 \right).
\end{aligned}
\tag{81}
$$

The discriminant reads:

$$
\Delta^{\mathfrak{su}(2)\,\mathcal{N}=2^*} = \Delta_0(\tau_{\mathrm{uv}})\prod_{i=1}^{3}\left( \widetilde{u} + \frac{1}{4}e_i\, m^2 \right)^2, \qquad \Delta_0(\tau_{\mathrm{uv}}) \equiv \frac{1}{27}\left( E_4^3 - E_6^2 \right), \tag{82}
$$

therefore the three $I_2$ singular fibers, $I_2^{(i)}$, are located at $\widetilde{u}_i \equiv -\frac{1}{4}e_i m^2$. The $\mathbb{Z}_2 \oplus \mathbb{Z}_2$ torsion sections are given by:

$$
P_i = \left( \frac{1}{24}(e_i^2 + 2e_j e_k)m^2 - \frac{1}{2}e_i\widetilde{u},\, 0 \right), \qquad j,k \neq i, \quad j \neq k, \tag{83}
$$

satisfying $2P_i = \mathcal{O}$, and $P_i + P_j = P_k$ for $i \neq j$ and $k \neq i,j$. One can check that $P_i$ intersects the node of the $I_2^{(j)}$ singular fibres if $j \neq i$. This implies that the 2-isogeny along the section $P_i$ will give us a new curve with bulk singularities $(2I_1, I_4)$.

### 3.1.1 BPS states and charge normalisation

Analogously to the case of the pure $\mathfrak{su}(2)$ relative theory, the light BPS states for this theory have physical charges:

$$
\gamma_1 \equiv \gamma_A = \sqrt{2}(0,-1), \qquad \gamma_2 \equiv \gamma_M = \sqrt{2}(1,0), \qquad \gamma_3 \equiv \gamma_D = \sqrt{2}(-1,1), \tag{84}
$$

corresponding to the adjoint hypermultiplet and the $SU(2)$ monopole and dyon, respectively. The BPS state $\gamma_i$ becomes massless at the point $\widetilde{u}_i$. The charge normalisation is the same as in (53). We then have the well-known BPS quiver for the $\mathcal{N}=2^*$ theory:

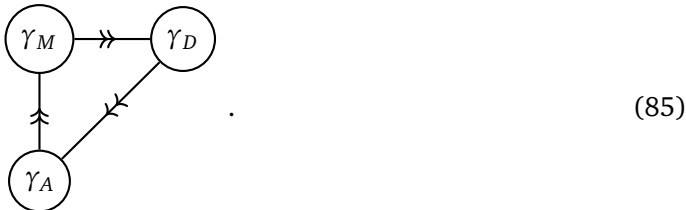

$$\tag{85}$$

This matches the recent discussion in [8], where the BPS quiver was derived from the $\Gamma(2)$ fundamental domain of the theory [30]. This domain is shown in figure 7a, where we also indicate a slightly different basis of light BPS states. Note, however, that a mutation on the dyon node $\gamma_D$ of the BPS quiver leads us back to (84).

### 3.1.2 The $S$-duality action

The 4d $\mathcal{N}=4$ SYM theory enjoys an exact $S$-duality, which is realised explicitly as an invariance of the relative SW curve under $\mathrm{SL}(2,\mathbb{Z})$ transformations of $\tau_{\mathrm{uv}}$, the UV gauge coupling. Let us denote by $\mathbf{S}$ and $\mathbf{T}$ the standard generators of this $S$-duality group. Let us recall that, in the case of the absolute theories $SU(2)$ and $SO(3)_\pm$, we have the relations [3]:

$$
\mathbf{T}\;\circlearrowright\; SU(2) \xleftrightarrow{\;\;\mathbf{S}\;\;} SO(3)_+ \xleftrightarrow{\;\;\mathbf{T}\;\;} SO(3)_- \;\circlearrowleft\; \mathbf{S}. \tag{86}
$$

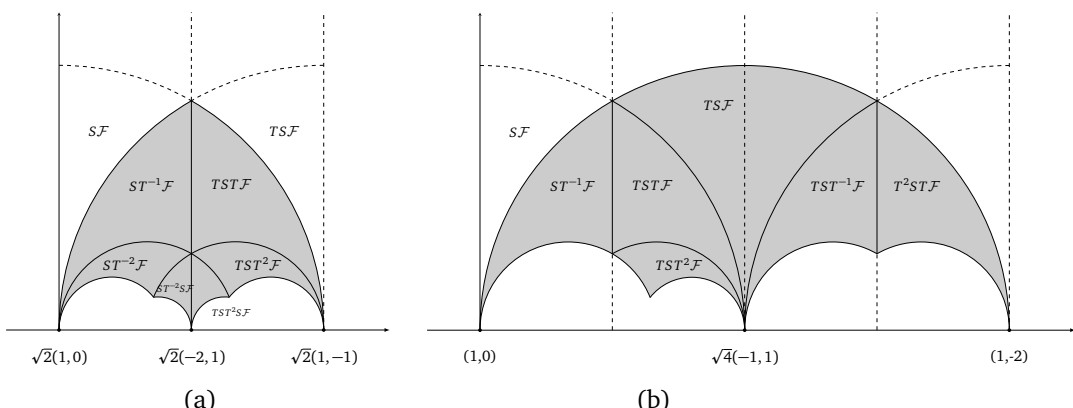

$$\sqrt{2}(1,0) \qquad \sqrt{2}(-2,1) \qquad \sqrt{2}(1,-1) \qquad\qquad (1,0) \qquad \sqrt{4}(-1,1) \qquad (1,-2)$$

(a) $\qquad\qquad\qquad\qquad$ (b)

Figure 7: Fundamental domains for the (a) relative ($\Gamma(2)$) and (b) absolute $SU(2)$ ($\Gamma^0(4)$) SW curves of the 4d $\mathcal{N} = 2^*$ theory.

We also note that **S** and **T** act on the modular forms $e_i$ as:

$$\mathbf{S} : (e_1, e_2, e_3) \mapsto \left(\tau_{\text{uv}}^2 e_2, \tau_{\text{uv}}^2 e_1, \tau_{\text{uv}}^2 e_3\right), \qquad \mathbf{T} : (e_1, e_2, e_3) \mapsto (e_1, e_3, e_2), \qquad (87)$$

and on the mass parameter as:

$$\mathbf{S} : m \mapsto \frac{m}{\tau_{\text{uv}}}, \qquad \mathbf{T} : m \mapsto m. \qquad (88)$$

Recall that this $S$-duality is best understood from the point of view of the 6d $\mathcal{N} = (2,0)$ realisation of the theory, wherein the compactification torus has modular parameter $\tau_{\text{uv}}$ and the mass $m$ arises as a background flat connection. The curve (80) is manifestly invariant under the full SL$(2,\mathbb{Z})$ $S$-duality action. The CB parameter $\tilde{u}$ is now a function of both $\tau$ and $\tau_{\text{uv}}$. Its explicit form was found in [30, 31] and can be written in terms of the modular $\lambda$-function, which is the Hauptmodul of $\Gamma(2)$ – see appendix A.4. A possible solution for $\tilde{u} = \tilde{u}(\tau, \tau_{\text{uv}})$ is given below, which is chosen to be consistent with the pure $\mathfrak{su}(2)$ limit, as we discuss momentarily:

$$\tilde{u}(\tau, \tau_{\text{uv}}) = \frac{1}{12} m^2 \vartheta_3(\tau_{\text{uv}})^4 \frac{\lambda(\tau_{\text{uv}})^2 + 2\left(\lambda(\tau) - 1\right)\lambda(\tau_{\text{uv}}) - \lambda(\tau)}{\lambda(\tau_{\text{uv}}) - \lambda(\tau)}, \qquad \lambda = \frac{\vartheta_2^4}{\vartheta_3^4}. \qquad (89)$$

An important difference compared to SQCD with $N_f = 4$, however, is that the mass parameter transforms non-trivially under SL$(2,\mathbb{Z})$ transformations, as mentioned above. As such, $\tilde{u}(\tau, \tau_{\text{uv}})$ becomes a bimodular form under $\Gamma(2)_\tau \times \Gamma(2)_{\tau_{\text{uv}}}$, of weights $(0,0)$. Importantly, it is only the action of $\gamma \in$ SL$(2,\mathbb{Z})$ on both $\tau$ and $\tau_{\text{uv}}$ simultaneously that leaves $\tilde{u}(\tau, \tau_{\text{uv}})$ invariant. Thus, $\tilde{u}(\tau, \tau_{\text{uv}})$ is a bimodular form for the triple $(\Gamma(2), \Gamma(2); \text{SL}(2,\mathbb{Z}))$, in the notation of [30].

The expression (89) emphasizes another important aspect. In the massless limit, we have the singularity structure $(I_0^*; I_0^*)$, as can be readily seen from (81). In this case, the effective gauge coupling is constant over the whole Coulomb branch. From (89), it is clear that this can only happen for:

$$\tau(u)|_{m=0} = \tau_{\text{uv}}. \qquad (90)$$

This point will be important when discussing the absolute curves.

The light states (84) are given in the $SU(2)$ frame, and $S$-duality acts on the charges as:

$$\mathbf{S} : \gamma = (m, q) \mapsto (-q, m), \qquad \mathbf{T} : \gamma = (m, q) \mapsto (m, q - m). \qquad (91)$$

This means that, going from the $SU(2)$ to the $SO(3)_+$ description, we exchange the monopole state $\gamma_M$ with the adjoint hypermultiplet state $\gamma_A$ and leave the dyon unaffected, while going from $SO(3)_+$ to $SO(3)_-$ we exchange the dyon and the monopole and leave the adjoint hypermultiplet unaffected.

### 3.1.3 Weak coupling limit and flavour decoupling

Let us consider the limit to the pure $\mathfrak{su}(2)$ $\mathcal{N} = 2$ gauge theory. This limit was discussed in [4, 31, 32]. While the three $I_2$ singularities are exchanged by $S$-duality, the flavour decoupling limit $m \to \infty$ must obviously break this symmetry, sending one $I_2$ singularity to infinity:

$$\left(I_0^*; 3I_2\right) \to \left(I_2^*; 2I_2\right), \tag{92}$$

in order to obtain the relative $\mathfrak{su}(2)$ curve. In our conventions above, the pure $SU(2)$ Coulomb branch parameter is:

$$u = \widetilde{u} - \frac{1}{8} e_1(\tau_{\mathrm{uv}}) m^2, \tag{93}$$

where we fix an $S$-duality frame in which $\tau_{\mathrm{uv}}$ is interpreted as the $SU(2)$ gauge coupling. At small $q_{\mathrm{uv}}$, the modular forms $e_i$ have an expansion:

$$e_1 = \frac{2}{3} + 16q_{\mathrm{uv}} + 16q_{\mathrm{uv}}^2 + \cdots, \quad e_2 = -\frac{1}{3} - 8q_{\mathrm{uv}}^{1/2} - 8q_{\mathrm{uv}} + \cdots, \quad e_3 = -\frac{1}{3} + 8q_{\mathrm{uv}}^{1/2} - 8q_{\mathrm{uv}} + \cdots.$$

The flavour decoupling limit is given by:

$$\tau_{\mathrm{uv}} \to i\infty, \qquad m \to \infty, \qquad \Lambda^2 \equiv 2m^2 \sqrt{q_{\mathrm{uv}}} \quad \text{fixed}, \tag{94}$$

where $q_{\mathrm{uv}}$ and the dynamical scale $\Lambda$ of the pure gauge theory are matched at the scale set by the adjoint mass. One easily checks that the relative $\mathcal{N} = 2^*$ curve (81) reduces to the pure $\mathfrak{su}(2)$ relative curve (50) in this limit. One finds:

$$g_2 \approx g_2^{\mathfrak{su}(2)} + \frac{32u\Lambda^4}{3m^2}, \qquad g_3 \approx g_3^{\mathfrak{su}(2)} + \frac{4\Lambda^4(5u^2 + 3\Lambda^4)}{9m^2}, \tag{95}$$

up to higher-order terms in $1/m$. Let us note that the solution (89) is chosen such that $u(\tau, \tau_{\mathrm{uv}})$ is consistent with the $u(\tau)$ parameter of the $\mathfrak{su}(2)$ curve in (47).

Moreover, in this limit, the $I_2^{(i)}$ singularities are located at:

$$u^{(1)} = -\frac{m^2}{4} - \frac{3\Lambda^4}{2m^2}, \qquad u^{(2)} = \Lambda^2 + \frac{\Lambda^6}{m^4}, \qquad u^{(3)} = -\Lambda^2 - \frac{\Lambda^6}{m^4}, \tag{96}$$

in agreement with the identification of BPS states given in (84). In particular, $u^{(1)}$ is the location of the hypermultiplet states, which in the weak-coupling limit becomes massless at $4u \approx -m^2$, namely at $2a \pm m = 0$. Defining $u$ as in (93), we took the weak-coupling limit in the '$SU(2)$ frame', but since the three singularities are SL$(2, \mathbb{Z})$-covariant, the other $S$-dual frames give us the same exact result, once we define $u$ appropriately. Thus, the $\mathcal{N} = 2^*$ relative curve gives us the relative $\mathfrak{su}(2)$ curve, as expected from (92). Next, we will consider the absolute $\mathcal{N} = 2^*$ curve. In that case, the weak coupling limit will depend on the duality frame, giving rise to the $SU(2)$ and $SO(3)_\pm$ absolute curves in the appropriate limits.

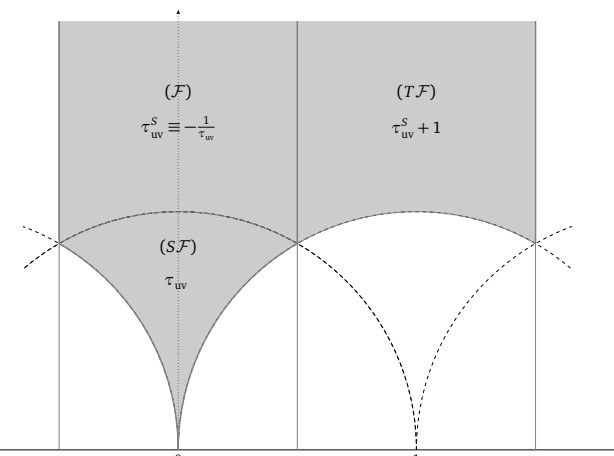

Figure 8: Fundamental domain for $\Gamma^0(2)$, the S-duality group for the $\mathcal{N} = 4$ SYM theory with gauge algebra $\mathfrak{su}(2)$. Here a point on the upper-half-plane is identified with $\tau_{\mathrm{uv}}^S \equiv -\frac{1}{\tau_{\mathrm{uv}}}$, where $\tau_{\mathrm{uv}}$ is the $SU(2)$ gauge coupling and $\tau_{\mathrm{uv}}^S$ is the $SO(3)_+$ gauge coupling in the $S$-dual frame.

## 3.2 The absolute $\mathcal{N} = 2^*$ curve

The $\mathcal{N} = 2^*$ Seiberg-Witten curve with deformation pattern $I_0^* \to I_4, 2I_1$ first appeared in [33]. It was further studied in [27], where it was written as:

$$
\begin{aligned}
g_2(v) &= \frac{1}{3}\left((1+3\Theta)v^2 + 8\mu_1^2\Theta v + 4\mu^4\,\Theta^2\right), \\
g_3(v) &= \frac{1}{27}\left((9\Theta-1)v^3 + 3\mu^2\Theta(5+3\Theta)v^2 + 24\mu^4\Theta^2 v + 8\mu^6\Theta^3\right).
\end{aligned}
\tag{97}
$$

Here, $v$ is a Coulomb branch parameter, $\mu$ is a mass and $\Theta$ a marginal coupling, with the conformal point recovered at $\mu = 0$.[8] The discriminant reads:

$$
\Delta(v) = \Theta(1-\Theta)^2 v^4(v^2 + 2\mu^2 v + \mu^4\Theta).
\tag{98}
$$

This curve has $\Phi_{\mathrm{tor}} = \mathbb{Z}_2$, with the 2-torsion section given by:

$$
P = \left(-\frac{2}{3}(v + \mu^2\Theta), 0\right).
\tag{99}
$$

This curve can be obtained from the relative curve $E_{\mathrm{rel}}$ in (81) using the explicit 2-isogeny along any of the sections $P_i$ in (83). In this way, we obtain the absolute theories:

$$
E[SU(2)] = E_{\mathrm{rel}}/\langle t_{P_1}\rangle, \qquad E[SO(3)_+] = E_{\mathrm{rel}}/\langle t_{P_2}\rangle, \qquad E[SO(3)_-] = E_{\mathrm{rel}}/\langle t_{P_3}\rangle,
\tag{100}
$$

where we include the rescaling by $\alpha = \sqrt{2}$ as usual. Indeed, the absolute curves are principally polarised while the relative curve was not. Focussing on the $SU(2)$ duality frame with the CB parameter (93), we find the relations:

$$
v = -\frac{3}{2}e_1\left(u + \frac{3}{8}e_1 m^2\right), \qquad \mu = \frac{3}{4}e_1 m,
\tag{101}
$$

with the marginal parameter $\Theta$ given in terms of the UV gauge coupling by:

$$
\Theta(\tau_{\mathrm{uv}}) = \frac{8}{9} + \frac{4}{9}\frac{e_2 e_3}{e_1^2} = \frac{4\vartheta_4^4\left(\vartheta_2^4 + \vartheta_4^4\right)}{\left(\vartheta_3^4 + \vartheta_4^4\right)^2},
\tag{102}
$$

---

[8]In [27], the parameters $v$, $\Theta$ and $\mu$ are denoted by $u$, $\alpha^2$, and $m$, respectively. We also rescaled the curve by a factor $\alpha = \sqrt{2}$.

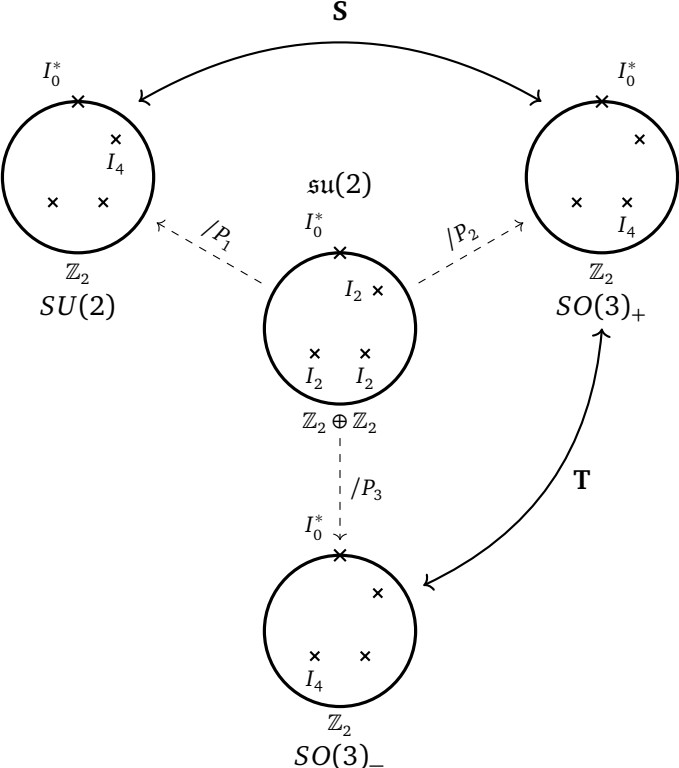

Figure 9: Coulomb branch geometries for the $\mathcal{N} = 2^*$ curves. From the relative curve with $\mathbb{Z}_2 \oplus \mathbb{Z}_2$ torsion, whose CB geometry is shown in the middle, we obtain the three absolute curves as indicated. The absolute curves are also related to each other by the action of the $S$-duality group $\Gamma^0(2)$ acting on the UV gauge coupling.

written here in terms of the Jacobi functions $\vartheta(\tau_{\mathrm{uv}})$ reviewed in appendix A.4, which satisfy the relation $\vartheta_3^4 = \vartheta_2^4 + \vartheta_4^4$. Note that $\Theta(\tau_{\mathrm{uv}})$ is invariant under $T$-transformations, as well as under $ST^2S$-transformations. Thus, this is a modular function for $\Gamma_0(2)$, an index-3 congruence subgroup of $\mathrm{PSL}(2, \mathbb{Z})$, which is indeed the $S$-duality group of the $SU(2)\,\mathcal{N} = 4$ SYM theory [3]. To see how this becomes manifest, consider again the massless limit of the curve $\mu \to 0$. Then, $\tau(u)|_{\mu=0} = 2\tau_{\mathrm{uv}}$ throughout the whole Coulomb branch, and, upon some rescaling of the SW curve, the only dependence on $\tau_{\mathrm{uv}}$ will be through the $\Theta$ function.

The map (101)-(102) was already worked out in [27, 33], and the correct interpretation of the two distinct SW curves in terms of the relative and absolute theories was first given in [13]. In $S$-duality frames for $\tau_{\mathrm{uv}}$ that correspond to $SO(3)_\pm$, the map between parameters can be obtained from (101)-(102) by following the transformations (86). In particular, one finds:

$$\Theta_S = \frac{8}{9} + \frac{4}{9}\frac{e_1 e_3}{e_2^2} = \frac{4\vartheta_2^4\vartheta_3^4}{\left(\vartheta_2^4 + \vartheta_3^4\right)^2}, \qquad \Theta_{TS} = \frac{8}{9} + \frac{4}{9}\frac{e_1 e_2}{e_3^2} = -\frac{4\vartheta_2^4\vartheta_4^4}{\left(\vartheta_2^4 - \vartheta_4^4\right)^2}, \qquad (103)$$

where $\Theta_S$ is obtained from $\Theta$ by a $S$-transformation, and similarly for $\Theta_{TS}$. Here, $\Theta_S$ is a modular function for $\Gamma^0(2)$, and it is written in terms of the $\tau_{\mathrm{uv}}^S$ parameter that spans the upper-half-plane shown in figure 8.

### 3.2.1 The absolute $SU(2)$ $\mathcal{N}=2^*$ curve

Following the 2-isogeny from the relative curve in the $SU(2)$ frame, we find the absolute curve:

$$
\begin{aligned}
g_2^{SU(2)\mathcal{N}=2^*} &= \frac{1}{768}\Big(64u^2\left(\vartheta_2^8 + 16\vartheta_4^4\vartheta_2^4 + 16\vartheta_4^8\right) + 16m^2u\left(\vartheta_2^4 + 2\vartheta_4^4\right)\vartheta_2^8 \\
&\quad + m^4\left(\vartheta_2^8 - 12\vartheta_4^4\vartheta_2^4 - 12\vartheta_4^8\right)\vartheta_2^8\Big), \\
g_3^{SU(2)\mathcal{N}=2^*} &= \frac{1}{110592}\Big(512u^3\left(\vartheta_2^4 + 2\vartheta_4^4\right)\left(\vartheta_2^8 - 32\vartheta_4^4\vartheta_2^4 - 32\vartheta_4^8\right) \\
&\quad + 192m^2u^2\left(\vartheta_2^8 - 8\vartheta_4^4\vartheta_2^4 - 8\vartheta_4^8\right)\vartheta_2^8 \\
&\quad + 24m^4u\left(\vartheta_2^{12} + 14\vartheta_4^4\vartheta_2^8 + 36\vartheta_4^8\vartheta_2^4 + 24\vartheta_4^{12}\right)\vartheta_2^8 \\
&\quad + m^6\left(\vartheta_2^8 + 36\vartheta_4^4\vartheta_2^4 + 36\vartheta_4^8\right)\vartheta_2^{16}\Big),
\end{aligned}
\tag{104}
$$

with the discriminant:

$$
\Delta^{SU(2)\mathcal{N}=2^*} = \frac{\vartheta_2^{16}\vartheta_3^4\vartheta_4^4}{16}\left(u + \frac{m^2}{8}(\vartheta_3^4 + \vartheta_4^4)\right)^4\left(u^2 - \frac{m^4}{64}\vartheta_2^8\right).
\tag{105}
$$

Here, all Jacobi theta functions depend on the $\tau_{\text{uv}}$ parameter of the relative curve. That is, the isogeny implemented as a quotient by the $\mathbb{Z}_2$ torsion subgroup generated by $P_1$ will only act on $\tau$. However, this creates a 'mismatch' between the IR and UV couplings, as we will see momentarily.

It is straightforward to take the weak-coupling and large-mass limit (94) on this curve, and one reproduces exactly the absolute $SU(2)$ curve (8):

$$
g_2^{SU(2)\mathcal{N}=2^*} \approx g_2^{SU(2)} + \frac{8u\Lambda^4}{3m^2}, \qquad g_3^{SU(2)\mathcal{N}=2^*} \approx g_3^{SU(2)} - \frac{4\Lambda^4(2u^2 - 3\Lambda^4)}{9m^2}.
\tag{106}
$$

In that limit, the three singularities located at:

$$
u^{(1)} = -\frac{m^2}{8}\left(\vartheta_3^4 + \vartheta_4^4\right), \qquad u^{(2)} = \frac{m^2}{8}\vartheta_2^4, \qquad u^{(3)} = -\frac{m^2}{8}\vartheta_2^4,
\tag{107}
$$

have the same weak-coupling limit as in (96). This identifies the $I_4$ singularity as the adjoint hypermultiplet singularity, so that the $m \to \infty$ limit realises the geometric limit $(I_0^*; I_4, 2I_2) \to (I_4^*; 2I_1)$.

To see how the $S$-duality group manifests itself on the curve, we first look at the function $u = u\left(\tau_{SU(2)}, \tau_{\text{uv}}\right)$. For this, we introduce the Hauptmodul of $\Gamma^0(4)$, as in (19):

$$
f(\tau) = \frac{\vartheta_2(\tau)^4 + \vartheta_3(\tau)^4}{2\vartheta_2(\tau)^2\vartheta_3(\tau)^2}.
\tag{108}
$$

Then, we find that a solution consistent with the decoupling limit to (19) and (89) is given by:

$$
u\left(\tau_{SU(2)}, \tau_{\text{uv}}\right) = \frac{1}{8}m^2\vartheta_2(\tau_{\text{uv}})^4\frac{\left(f(\tau_{SU(2)}) + 1\right)\lambda(\tau_{\text{uv}}) - 2f(\tau_{SU(2)})}{\left(f(\tau_{SU(2)}) + 1\right)\lambda(\tau_{\text{uv}}) - 2},
\tag{109}
$$

with the modular $\lambda$ function as defined in (89) and appendix A.4. This is now a bimodular form under $\Gamma^0(4)_\tau \times \Gamma(2)_{\tau_{\text{uv}}}$. This, in fact, was to be expected from the $u(\tau, \tau_{\text{uv}})$ expression (89) of the relative curve, which was a bimodular form under $\Gamma(2) \times \Gamma(2)$, since the 2-isogeny acts by:

$$
\tau_{\mathfrak{su}(2)} \mapsto \tau_{SU(2)} = 2\tau_{\mathfrak{su}(2)}.
\tag{110}
$$

However, the isogeny does not affect the UV parameter $\tau_{\mathrm{uv}}$. As a result, in the massless limit we have the 'mismatch' in couplings: $\tau_{SU(2)}(u) = 2\tau_{\mathrm{uv}}$, throughout the entire Coulomb branch. For this reason, we can introduce a renormalized parameter $\tau^{\mathrm{uv}}_{SU(2)} = 2\tau_{\mathrm{uv}}$. We should stress, however, that the S-duality of the $\mathcal{N} = 4$ theory acts on the $\tau_{\mathrm{uv}}$ parameter of the relative curve, rather than on $\tau^{\mathrm{uv}}_{SU(2)}$. As such, we would expect the $SU(2)$ curve to be invariant under simultaneous $\mathbf{T}^2$-transformations on $\tau_{SU(2)}$ and $\tau^{\mathrm{uv}}_{SU(2)}$, which correspond to $\mathbf{T}$-transformations on the parameters of the relative curve.

To see this invariance in the new normalization, we first notice that $\lambda\left(\frac{\tau}{2}\right) = \frac{2}{1+f(\tau)}$, which leads to:

$$u(\tau, \tau_{\mathrm{uv}}) = \frac{1}{2}m^2 \vartheta_2(\tau_{\mathrm{uv}})^2 \, \vartheta_3(\tau_{\mathrm{uv}})^2 \, \frac{f(\tau_{\mathrm{uv}})f(\tau)-1}{f(\tau_{\mathrm{uv}})-f(\tau)}, \tag{111}$$

where $\tau$ and $\tau_{\mathrm{uv}}$ correspond to $\tau_{SU(2)}$ and $\tau^{\mathrm{uv}}_{SU(2)}$, respectively. Thus, $\frac{u(\tau,\tau_{\mathrm{uv}})}{m^2}$ becomes now a bimodular form under the triple $\left(\Gamma^0(4), \Gamma^0(4); \Gamma^0(2)\right)$ [30]. Moreover, $u_{SU(2)}$ is, in particular, invariant under a simultaneous $\mathbf{T}^2$-transformation as expected. Hence, this explains the $\mathbf{T}$-invariance of the $SU(2)$ global form in (86). As we will see momentarily, the $\mathbf{S}$-transformation will lead to the $SO(3)_+$ global form.

Let us also mention that, in the new normalization for the UV coupling, the curve (104) can be expressed as:

$$g_2^{SU(2)\mathcal{N}=2^*} = \frac{1}{3}\vartheta_2^4\vartheta_3^4\Big(4u^2(-3+4f^2) + 4m^2\vartheta_2^2\vartheta_3^2 uf + m^4\vartheta_2^4\vartheta_3^4(4-3f^2)\Big),$$

$$g_3^{SU(2)\mathcal{N}=2^*} = \frac{1}{27}\vartheta_2^6\vartheta_3^6\Big(2fu + m^2\vartheta_2^2\vartheta_3^2\Big)\Big(4u^2\left(8f^2-9\right) - 4m^2\vartheta_2^2\vartheta_3^2 uf + m^4\vartheta_2^4\vartheta_3^4\left(8-9f^2\right)\Big),$$

$$\tag{112}$$

which makes the $\Gamma^0(4)$ dependence on $\tau^{\mathrm{uv}}_{SU(2)}$ manifest.

A fundamental domain for the absolute $\mathcal{N} = 2^*$ $SU(2)$ curve is, in fact, the $\Gamma^0(4)$ domain of the pure $SU(2)$ theory shown in figure 1a, with the distinction that the width-4 cusp at infinity will now correspond to the adjoint hypermultiplet. Thus, the light BPS states for this theory are:

$$\gamma_1 \equiv \gamma_A = (0,-2), \qquad \gamma_2 \equiv \gamma_M = (1,0), \qquad \gamma_3 \equiv \gamma_D = (-1,2), \tag{113}$$

reproducing the $\mathcal{N} = 2^*$ quiver (85). This is, of course, in agreement with the fundamental domain shown in 7b, with the two bases being related by a mutation on the $\gamma_D$ node of the quiver.

### 3.2.2 The absolute $SO(3)_\pm$ $\mathcal{N} = 2^*$ curves

The action of the $S$-duality group permutes the global structure of the $\mathfrak{su}(2)$ gauge group as in (86), with the UV gauge coupling transforming as shown in figure 8. Note that, here, we are referring to the UV coupling in the normalization of the relative theory $\tau^{\mathrm{uv}}_{\mathfrak{su}(2)}$. We can thus easily work out the absolute SW curves in the $SO(3)_\pm$ duality frames. The simplest way to derive the $SO(3)_+$ curve is to follow the isogeny along the $P_2$ section of the relative curve. The relations between relative and absolute curves is summarised in figure 9. We can equivalently obtain the $SO(3)_+$ curve through an $S$-duality transformation of the absolute curve written above in the $SU(2)$ frame. The only subtlety is that the duality-invariant Coulomb-branch parameter is $\tilde{u}$ and not $u$, so that under $\mathbf{S}$ we have the non-trivial transformation of $u$ as:

$$\mathbf{S} : u_{SU(2)} \mapsto u_{SO(3)_+} + \frac{1}{8}(e_1 - e_2)m^2 = u_{SO(3)_+} + \frac{1}{8}\vartheta_3^4 m^2. \tag{114}$$

Note that the $\mathbf{S}$ transformations acts simultaneously on the $(\tau, \tau_{\mathrm{uv}})$ couplings, which, here, are the couplings of the relative curve.

As for the absolute $SU(2)$ curve, we can bring the curve in an $SO(3)_+$ normalization, where:

$$\tau_{SO(3)_+} = \frac{1}{2}\tau_{\mathfrak{su}(2)} = \frac{1}{4}\tau_{SU(2)}\,. \tag{115}$$

Doing this for both the UV and IR couplings, we have:

$$
\begin{aligned}
g_2^{SO(3)_+\,\mathcal{N}=2^*} &= \frac{1}{3072}\Big(256u^2(\vartheta_3^8 - \vartheta_3^4\vartheta_4^4 + \vartheta_4^8) - 32m^2\vartheta_3^4\vartheta_4^4(\vartheta_3^4 + \vartheta_4^4)u \\
&\qquad + m^4\vartheta_3^4\vartheta_4^4(-3\vartheta_3^8 + 10\vartheta_3^4\vartheta_4^4 - 3\vartheta_4^8)\Big), \\
g_3^{SO(3)_+\,\mathcal{N}=2^*} &= \frac{1}{442368}\Big(4(\vartheta_3^4 + \vartheta_4^4)u - m^2\vartheta_3^4\vartheta_4^4\Big)\Big(m^4\vartheta_3^4\vartheta_4^4(-9\vartheta_3^8 + 14\vartheta_3^4\vartheta_4^4 - 9\vartheta_4^8) \\
&\qquad + 256u^2(2\vartheta_3^8 - 5\vartheta_3^4\vartheta_4^4 + 2\vartheta_4^8) + 32m^2\vartheta_3^4\vartheta_4^4(\vartheta_3^4 + \vartheta_4^4)u\Big),
\end{aligned}
\tag{116}
$$

with the discriminant:

$$\Delta^{SO(3)_+\,\mathcal{N}=2^*} = \frac{\vartheta_2^8\vartheta_3^8\vartheta_4^8}{256}\left(u^2 - \frac{1}{64}m^4\vartheta_3^4\vartheta_4^4\right)\left(u - \frac{1}{16}m^2(\vartheta_3^4 + \vartheta_4^4)\right)^4\,. \tag{117}$$

Next, we introduce the $\Gamma_0(4)$ modular function:

$$\tilde{f}(\tau) = 1 + \frac{8\vartheta_3(\tau)^2\vartheta_4(\tau)^2}{\left(\vartheta_3(\tau)^2 - \vartheta_4(\tau)^2\right)^2}\,. \tag{118}$$

Note that $\lambda(2\tau) = 2\left(1 + \tilde{f}(\tau)\right)^{-1}$. We then find that, in the $SO(3)_+$ normalization, the CB parameter can be expressed as:

$$u_{SO(3)_+}(\tau, \tau_{\mathrm{uv}}) = -\frac{1}{8}m^2\vartheta_3(\tau_{\mathrm{uv}})^2\vartheta_4(\tau_{\mathrm{uv}})^2\frac{2 + \tilde{f}(\tau_{\mathrm{uv}}) + \tilde{f}(\tau)}{\tilde{f}(\tau) - \tilde{f}(\tau_{\mathrm{uv}})}\,. \tag{119}$$

As such, $u(\tau, \tau_{\mathrm{uv}})$ is a bimodular form under the triple $(\Gamma_0(4), \Gamma_0(4); \Gamma_0(2))$, meaning that it is invariant under $\Gamma_0(2)$ under simultaneous transformations on $(\tau, \tau_{\mathrm{uv}})$, as also shown in [30].

The expression (119) can be obtained directly from (116). Alternatively, as already alluded to, we can perform an **S**-transformation on the $SU(2)$ form (111). We mentioned before that S-duality acts on the $\tau_{\mathfrak{su}(2)}$ couplings of the relative curve. As such, under an **S** transformation, we still have:

$$\tau_{SU(2)} = \frac{1}{2}\tau_{\mathfrak{su}(2)} \quad \mapsto \quad \frac{1}{2}\left(-\frac{1}{\tau_{\mathfrak{su}(2)}}\right) = -\frac{1}{\tau_{SO(3)_+}}\,, \tag{120}$$

for the both the UV and IR coupling. Thus, using the identity $f\left(-\frac{1}{\tau}\right) = \frac{3 + \tilde{f}(\tau)}{-1 + \tilde{f}(\tau)}$, one finds that the $SO(3)_+$ CB parameter (119) is precisely the **S** transformation of the $SU(2)$ form (111).[9]

We can similarly write down the $SO(3)_-\,\mathcal{N} = 2^*$ curve, which is obtained from the relative curve by an isogeny along $P_3$. It is also simply obtained by a **T** transformation on the $SO(3)_+$ $\mathcal{N} = 2^*$ curve, which, in the $SO(3)$ normalization, acts as:

$$\tau_{SO(3)_-} = \tau_{SO(3)_+} + \frac{1}{2}\,. \tag{121}$$

Moreover, since the $\tilde{u}$ parameter is the one invariant under $\mathrm{SL}(2,\mathbb{Z})$, we have $u_{SO(3)_-} = \tilde{u} - \frac{1}{8}e_3 m^2$, from which we find:

$$u_{SO(3)_-}(\tau, \tau_{\mathrm{uv}}) = -\frac{m^2}{32}\left(\vartheta_3^2(\tau_{\mathrm{uv}}) + \vartheta_4^2(\tau_{\mathrm{uv}})\right)^2\frac{-2 + \tilde{f}(\tau_{\mathrm{uv}}) + \tilde{f}(\tau)}{\tilde{f}(\tau) - \tilde{f}(\tau_{\mathrm{uv}})}\,. \tag{122}$$

---

[9]Note that the overall rescaling comes from the transformation of the mass term in (88), which involves the coupling of the relative curve.

Finally, the weak-coupling and large-mass limit on the $SO(3)_\pm$ $\mathcal{N} = 2^*$ curves sends one $I_1$ singularity to infinity, thus reproducing the absolute pure $SO(3)_\pm$ curves (34):

$$
\begin{aligned}
g_2^{SO(3)_\pm \mathcal{N}=2^*} &\approx g_2^{SO(3)_\pm} \pm \frac{2\Lambda^2\left(5u \pm 3\Lambda^2\right)\left(3u \pm 5\Lambda^2\right)}{3m^2}, \\
g_3^{SO(3)_\pm \mathcal{N}=2^*} &\approx g_3^{SO(3)_\pm} \pm \frac{\Lambda^2\left(21u^3 \pm 173\Lambda^2 u^2 + 231\Lambda^4 u \pm 87\Lambda^6\right)}{18m^2}.
\end{aligned}
\tag{123}
$$

Indeed, labelling the singular points $u^{(i)}$ $SO(3)_+$ curve, at which the discriminant (117) vanishes, as in (107), the weak coupling limit in the $SO(3)_+$ duality frame reproduces (96). Similar considerations hold for the $SO(3)_-$ curve by simply exchanging $u^{(2)}$ and $u^{(3)}$.

We have already mentioned the fundamental domains for the $\mathfrak{su}(2)$ and $SU(2)$ curves, which are shown in figure 7. We can also work out the domains for $SO(3)_+$ and $SO(3)_-$ from the transformation of $\tau$, as discussed around (33). For completeness, we list below the positions of the cusps on the upper half-plane, which can be used to reproduce the expected light BPS states:

$$
\begin{array}{|c||c|c|c|}
\hline
G & \text{cusp positions} & \text{cusp widths} & \tau \\
\hline\hline
SU(2) & (0,1,2) & (1,4,1) & \tau_{SU(2)} = 2\tau_{\mathfrak{su}(2)} \\
\hline
\mathfrak{su}(2) & \left(0,\tfrac{1}{2},1\right) & (2,2,2) & \tau_{\mathfrak{su}(2)} \\
\hline
SO(3)_+ & \left(0,\tfrac{1}{4},\tfrac{1}{2}\right) & (4,1,1) & \tau_{SO(3)_+} = \tfrac{1}{2}\tau_{\mathfrak{su}(2)} \\
\hline
SO(3)_- & \left(\tfrac{1}{2},\tfrac{3}{4},1\right) & (1,1,4) & \tau_{SO(3)_-} = \tau_{SO(3)_+} + \tfrac{1}{2} \\
\hline
\end{array}
\tag{124}
$$

# 4 Five-dimensional $E_1[\mathfrak{su}(2)]$ theories on $S^1$

In the rest of this paper, we will generalise the previous discussion to SW geometries for higher-dimensional supersymmetric field theories compactified to 4d. Let us first consider any rank-one 5d SCFT $\mathcal{T}_{5d}$ on a circle $S^1$ of finite radius $\beta$, giving us a 4d $\mathcal{N} = 2$ KK theory $\mathcal{T}_{KK} \equiv D_{S^1}\mathcal{T}_{5d}$ with a one-dimensional Coulomb branch. In this case, the Coulomb-branch order parameter is the dimensionless expectation value of a half-BPS line, $W$, wrapping the circle. It is denoted by:

$$
U = \langle W \rangle. \tag{125}
$$

Let us further assume that the 5d theory has a one-form symmetry $\Gamma_{5d}^{[1]} = \mathbb{Z}_n$. Upon circle compactification, we obtain both a 1-form and a 0-form symmetry in the 4d KK theory, which we denote by:

$$
\Gamma_{5d}^{[1]} = \mathbb{Z}_n \qquad \rightarrow \qquad \Gamma^{[1]} = \mathbb{Z}_n^{[1]}, \qquad \Gamma^{[0]} = \mathbb{Z}_n^{[0]}. \tag{126}
$$

If $\Gamma_{5d}^{[1]}$ is non-anomalous, we can gauge either or both of the symmetries $\Gamma^{[1]}$ and $\Gamma^{[0]}$, leading to a rich structure of 4d $\mathcal{N} = 2$ KK theories. In particular, we have the following commutative diagram of absolute theories:

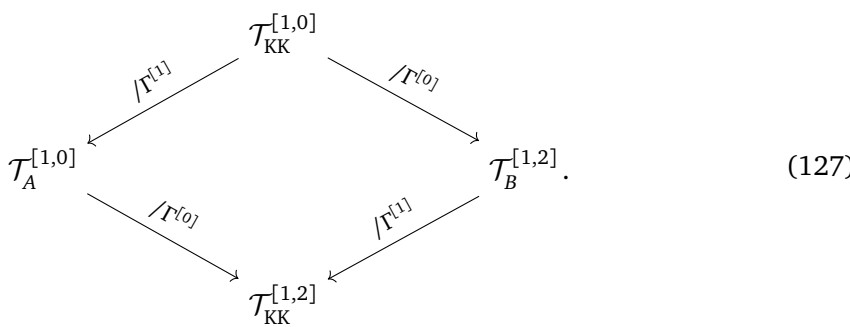

$$
\tag{127}
$$

Here, $\mathcal{T}_A$ and $\mathcal{T}_B$ denote the theory obtained by gauging either the 1-form or the 0-form symmetry of the original theory, $\mathcal{T}_A \equiv \mathcal{T}_{KK}/\Gamma^{[1]}$ and $\mathcal{T}_B \equiv \mathcal{T}_{KK}/\Gamma^{[0]}$, and the superscript denotes the discrete symmetries of the theory. In particular, we have $\mathcal{T}_{KK}^{[1,2]}$ with a 1-form and a 2-form symmetry, which corresponds to the circle reduction of the 5d SCFT $\mathcal{T}_{5d}/\Gamma_{5d}^{[1]}$, which indeed has a 2-form symmetry.[10] At the level of the SW geometry, the gauging of the one-form symmetry is performed through isogenies along torsion sections, as explained in the previous section. (In particular, there can be several consistent gaugings of $\Gamma^{[1]}$.) The gauging of the 0-form symmetry, on the other hand, is easily understood as a 'folding' of the $U$-plane, following the careful analysis of [34]. The 0-form symmetry acts on the $U$-plane by a phase dictated by the charge of the 5d line $W$ under $\Gamma_{5d}^{[1]}$. We thus interpret $\Gamma^{(0)}$ as an accidental $R$-symmetry of the 4d $\mathcal{N} = 2$ KK theory.[11]

## 4.1 The $\mathfrak{su}(2)$ curves for the 5d $E_1$ theory: Absolute and relative

We start with considering the $E_1$ SCFT [35]. In five-dimensions, it admits a deformation to a 5d $\mathcal{N} = 1$ gauge theory with $\mathfrak{su}(2)$ gauge group. This theory has a global form with a one-form symmetry $\Gamma_{5d}^{[1]} = \mathbb{Z}_2$, and that absolute 5d SCFT admits a real-mass deformation to a $SU(2)_0$ 5d gauge theory. Here, we will denote its circle compactification by $D_{S^1}E_1 \equiv E_1[SU(2)]^{[1,0]}$. We then expect the following commutative diagram of 4d $\mathcal{N} = 2$ theories:

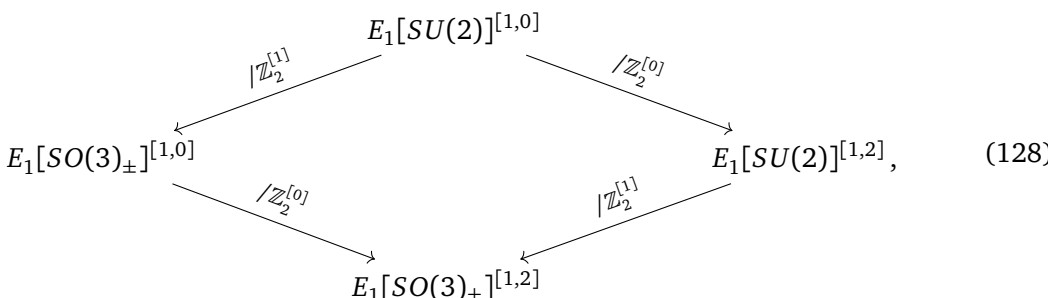

$$\tag{128}$$

giving us a concrete realisation of (127). Let us also introduce the shorthand notation:

$$(E_1)_0^{[p,q]} \equiv E_1[SU(2)]^{[p,q]}, \qquad (E_1)_\pm^{[p,q]} \equiv E_1[SO(3)_\pm]^{[p,q]}. \tag{129}$$

The $E_1$ SW geometry was first proposed by Nekrasov [36] and further studied in [37]. More recently, the $U$-plane has been studied from closely related perspectives in [7, 38–40]. The $E_1[SU(2)]^{[1,0]}$ curve is given by:

$$g_2^{(E_1)_0^{[1,0]}} = \frac{1}{12}\left(U^4 - 8(1+\lambda)U^2 + 16\left(1-\lambda+\lambda^2\right)\right),$$
$$g_3^{(E_1)_0^{[1,0]}} = -\frac{1}{216}\left(U^6 - 12(1+\lambda)U^4 + 24\left(2+\lambda+2\lambda^2\right)U^2 - 32\left(2-3\lambda-3\lambda^2+2\lambda^3\right)\right),$$
$$\tag{130}$$

and with discriminant $\Delta = \lambda^2\left(U^4 - 8(1+\lambda)U^2 + 16(1-\lambda)^2\right)$, in the notation of [7]. Here, the dimensionless parameter $\lambda$ is the exponentiated inverse 5d gauge coupling, and $\lambda \to 1$ is the 5d strong-coupling limit (giving us the curve for the undeformed $E_1$ SCFT on a circle).

At $\lambda \neq 0$, we have the singularities $(I_8; 4I_1)$: The $I_8$ fiber at infinity, corresponding to the 5d beta function [7], and four $I_1$ singularities on the Coulomb branch where a single BPS particle becomes massless. This corresponds to two copies of the 4d $SU(2)$ SW points,

---

[10]Recall that gauging a $p$-form symmetry in $d$ dimensions results in a dual $(d-p-2)$-form symmetry.

[11]It would be a subgroup of the classical $U(1)_r$ in 4d $\mathcal{N} = 2$ Lagrangian theories. Recall that the 5d $\mathcal{N} = 1$ theory only has an $SU(2)_R$ symmetry, since the 5d Coulomb branch is real.

viewed as holomomy saddles in the small circle limit [40]. These two copies are related by the spontaneously-broken symmetry $\mathbb{Z}_2^{[0]}$ inherited from the 5d one-form symmetry, which acts on the CB parameter as:

$$\mathbb{Z}_2^{[0]} \; : \; U \to -U \,. \tag{131}$$

At $\lambda = 1$, two of the $I_1$ singularities merge into an $I_2$ singularity, from which a non-trivial Higgs branch $\mathcal{M}_H = \mathbb{C}^2/\mathbb{Z}_2$ emerges. This reproduces the quantum Higgs branch of the 5d SCFT. The Coulomb branch for this $(I_8; I_2, 2I_1)$ massless SW curve is a modular curve for $\Gamma^0(8)$. The light BPS states at these singularities have magnetic-electric charges:

$$\gamma_1 = (1, 0), \qquad \gamma_2 = \gamma_3 = (-1, 2), \qquad \gamma_4 = (1, -4), \tag{132}$$

where the charges $\gamma = (m, q)$ are given in the same $SU(2)$ normalisation as section 2.1, with the Dirac pairing given by the homology lattice as in (17). We can then read off the well-known 5d BPS quiver [7, 8, 16]:



$$\tag{133}$$

The $\mathbb{Z}_2^{[1]}$ one-form symmetry of the $E_1[SU(2)]^{[1,0]}$ theory is the 'electric' one-form symmetry that preserves the BPS states (132). It acts by a sign on a 'Wilson line' with charge $\gamma_L = (0, 1)$. This line descends from the fundamental 'Wilson line' $W$ of the 5d SCFT transverse to the compactification circle.[12]

Let us recall that, at $\lambda \neq 1$, we have a non-trivial Mordell-Weil group $\Phi = \mathbb{Z} \oplus \mathbb{Z}_2$, with $\Phi_{\text{tor}} = \mathbb{Z}_2$ identified with the one-form symmetry. The non-trivial torsion section reads:

$$P_{\mathbb{Z}_2} = \left( \frac{1}{12} \left( U^2 - 4\lambda - 4 \right), 0 \right). \tag{134}$$

At $\lambda = 1$, the torsion MW group enhances to $\mathbb{Z}_4$, with the generator:

$$P_{\mathbb{Z}_4} = \left( \frac{1}{12} (U^2 + 4), -U \right), \tag{135}$$

with the above $\mathbb{Z}_2$ as the subgroup generated by $P_{\mathbb{Z}_2}|_{\lambda=1} = 2P_{\mathbb{Z}_4}$. The $\mathbb{Z}_2$ that arises as the cokernel of the inclusion of the 'generic' $\mathbb{Z}_2$ (generated by (134)) inside $\mathbb{Z}_4$ is related to the global form of the flavour symmetry group $SO(3)_F$ that acts on the quantum Higgs branch [7, 41, 42].

**The relative theory $E_1[\mathcal{R}_A]$.** We would like to gauge the electric $\mathbb{Z}_2^{[1]}$ symmetry by successive isogenies, as in the 4d $SU(2)$ case. We first obtain the relative SW curve $E_1[\mathcal{R}_A]$ by quotienting the $E_1[SU(2)]^{[1,0]}$ curve along the section (134). This gives us:

$$g_2^{\mathcal{R}_A} = \frac{1}{48} \left( U^4 - 8U^2(1+\lambda) + 16 \left( 1 + 14\lambda + \lambda^2 \right) \right),$$

$$g_3^{\mathcal{R}_A} = -\frac{1}{1728} \left( U^6 - 12U^4(1+\lambda) + 48U^2 \left( 1 - 10\lambda + \lambda^2 \right) - 64 \left( 1 - 33\lambda - 33\lambda^2 + \lambda^3 \right) \right), \tag{136}$$

---

[12]In the 5d $SU(2)_0$ massive phase, the line $W$, which must exist as a 5d line at the SCFT point, flows to the ordinary supersymmetric Wilson line.

with a discriminant $\Delta^{\mathcal{R}_A} = \frac{1}{64}\lambda\left(U^4 + 16(-1+\lambda)^2 - 8U^2(1+\lambda)\right)^2$. At $\lambda \neq 1$, this relative curve has the CB singularities $(I_4; 4I_2)$, while at $\lambda = 1$ we have $(I_4; 2I_2, I_4)$ – two $I_2$ singularities merge to give the $I_4$. Here, the $I_2$ singularities are undeformable, corresponding to the normalised charges:

$$\widetilde{\gamma}_1 = \sqrt{2}(1,0), \qquad \gamma_3 = \gamma_4 = \sqrt{2}(-1,1), \qquad \gamma_4 = \sqrt{2}(1,-2), \tag{137}$$

exactly as in (53). Note also that the massless curve is modular, with monodromy group $\Gamma^0(4) \cap \Gamma(2)$, which is conjugate to $\Gamma^0(8)$ in $\mathrm{PSL}(2,\mathbb{R})$ [8]. For $\lambda \neq 1$, we have $\Phi_{\mathrm{tor}} = \mathbb{Z}_2 \oplus \mathbb{Z}_2$, which we again identify with the defect group of the 4d $\mathcal{N} = 2$ KK theory. For $\lambda = 1$, this enhances to $\Phi_{\mathrm{tor}} = \mathbb{Z}_4 \oplus \mathbb{Z}_2$. The $\mathbb{Z}_2 \oplus \mathbb{Z}_2$ sections of the relative curve $\mathcal{R}_A$ are given by:

$$P_1 = \left(-\frac{1}{12}\left(U^2 - 4\lambda - 4\right), 0\right), \qquad P_\pm = \left(\frac{1}{24}\left(U^2 - 4\lambda - 4 \pm 24\sqrt{\lambda}\right), 0\right), \tag{138}$$

with $P_2 \equiv P_+$, $P_3 \equiv P_-$.

**The $E_1[SO(3)_\pm]^{[1,0]}$ curves.** Given the relative curve $\mathcal{R}_A$, we can obtain three principally polarised SW geometries by performing the 2-isogenies:

$$E_1[SU(2)]^{[1,0]} \cong \mathcal{R}_A/P_1, \qquad E_1[SO(3)_\pm]^{[1,0]} \cong \mathcal{R}_A/P_\pm. \tag{139}$$

Thus, in addition to the $E_1[SU(2)]^{[1,0]}$ curve discussed above, we find the $E_1[SO(3)_\pm]^{[1,0]}$ curves:

$$g_2^{(E_1)_\pm^{[1,0]}} = \frac{1}{192}\left(U^4 - 8U^2\left(1 \mp 30\lambda^{\frac{1}{2}} + \lambda\right) + 16\left(1 \mp 60\lambda^{\frac{1}{2}} + 134\lambda - 60\lambda^{\frac{3}{2}} + \lambda^2\right)\right),$$

$$g_3^{(E_1)_\pm^{[1,0]}} = \frac{1}{13824}\left(-U^6 + 12U^4\left(1 \pm 42\lambda^{\frac{1}{2}} + \lambda\right) - 48U^2\left(1 \pm 84\lambda^{\frac{1}{2}} - 346\lambda \pm 84\lambda^{\frac{3}{2}} + \lambda^2\right)\right.$$
$$\left. + 64\left(1 \pm 126\lambda^{\frac{1}{2}} - 1041\lambda \pm 1764\lambda^{\frac{3}{2}} - 1041\lambda^2 \pm 126\lambda^{\frac{5}{2}} + \lambda^3\right)\right). \tag{140}$$

The discriminant reads:

$$\Delta^{(E_1)_\pm^{[1,0]}} = \pm\frac{\sqrt{\lambda}}{4096}\left(U - 2 \mp 2\sqrt{\lambda}\right)^4\left(U + 2 \mp 2\sqrt{\lambda}\right)\left(U - 2 \pm 2\sqrt{\lambda}\right)\left(U + 2 \pm 2\sqrt{\lambda}\right)^4. \tag{141}$$

The singularity structure is $(I_2; 2I_1, 2I_4)$. The $I_1$ and $I_4$ singularities are exchanged when going between the $SO(3)_+$ and $SO(3)_-$ theory, which corresponds to a sign flip $\sqrt{\lambda} \to -\sqrt{\lambda}$. In the massless limit $\lambda \to 1$, we have:

$$\Delta^{(E_1)_+^{[1,0]}}\Big|_{\lambda=1} = \frac{(U-4)^4 U^2 (U+4)^4}{4096}, \qquad \Delta^{(E_1)_-^{[1,0]}}\Big|_{\lambda=1} = -\frac{(U-4)U^8(U+4)}{4096}, \tag{142}$$

so that we obtain either an $I_2$ or an $I_8$ singularity at the origin. The low energy dynamics of this singularity is in terms of a $U(1)$ vector multiplet coupled to two hypermultiplets of charge 1 or 2, respectively. Indeed, using the $SO(3)_\pm$ normalisation (29), we see that (132) becomes:

$$\begin{aligned}
\gamma_{1+} &= (2,0), & \gamma_{2+} = \gamma_{3+} &= (-2,1), & \gamma_{4+} &= (2,-2), \\
\gamma_{1-} &= (2,-1), & \gamma_{2-} = \gamma_{3-} &= (-2,2), & \gamma_{4-} &= (2,-3),
\end{aligned} \tag{143}$$

for $E_1[SO(3)_+]^{[1,0]}$, and $E_1[SO(3)_-]^{[1,0]}$, respectively, so that the mutually-local particles $\gamma_2$ and $\gamma_3$ give us either the $I_2$ or the $I_8$ singularity at the origin, where they become massless. In either case, the low energy description at $U = 0$ reproduces the expected 5d Higgs branch

$\mathbb{C}^2/\mathbb{Z}_2$. Moreover, in the $E_1[SO(3)_-]^{[1,0]}$ theory, we have a residual $\mathbb{Z}_2$ gauge symmetry that survives everywhere on the Higgs branch. The structure of the $U$-plane in each case is summarised in figure 10.

Finally, note that the massless curves are still modular; for $SO(3)_+$, the monodromy group is $\Gamma_0(4) \cap \Gamma(2)$, which is again conjugate to $\Gamma^0(8)$ in $PSL(2,\mathbb{R})$ [8]. Similarly, the monodromy group for $SO(3)_-$ is also conjugate to $\Gamma^0(8)$. The positions of the cusps and their widths are given below, and can be found from the transformation of $\tau$, as discussed around (33):

$$
\begin{array}{|c||c|c|c|}
\hline
\mathcal{T} & \text{cusp positions} & \text{cusp widths} & \tau \\
\hline\hline
E_1[SU(2)]^{[1,0]} & (\infty; 0, 2, 4) & (8; 1, 2, 1) & \tau_{SU(2)} = 2\tau_{\mathfrak{su}(2)} \\
\hline
\mathcal{R}_A & (\infty; 0, 1, 2) & (4; 2, 4, 2) & \tau_{\mathfrak{su}(2)} \\
\hline
E_1[SO(3)_+]^{[1,0]} & \left(\infty; 0, \frac{1}{2}, 1\right) & (2; 4, 2, 4) & \tau_{SO(3)_+} = \frac{1}{2}\tau_{\mathfrak{su}(2)} \\
\hline
E_1[SO(3)_-]^{[1,0]} & \left(\infty; \frac{1}{2}, 1, \frac{3}{2}\right) & (2; 1, 8, 1) & \tau_{SO(3)_-} = \tau_{SO(3)_+} + \frac{1}{2} \\
\hline
\end{array}
\tag{144}
$$

These reproduce the BPS states in (143), as expected.

**Limit to 4d $\mathcal{N} = 2$ pure $\mathfrak{su}(2)$.** Let us also discuss the 4d limit of the $E_1$ curves discussed so far. It is obtained by restoring the dependence on the radius $\beta$ of the fifth direction, and by taking the small-radius limit:

$$
\beta \to 0, \qquad U = 2 + (2\pi\beta)^2 u, \qquad \sqrt{\lambda} = 2\pi^2\beta^2\Lambda^2,
\tag{145}
$$

with $u$ and $\Lambda$ finite. We should also rescale the curve with the factor $\alpha = 2\pi\beta$ before taking the limit. Then, the relative curve $\mathcal{R}_A$ in (136) reproduces the pure $\mathfrak{su}(2)$ curve (50), and similarly for the absolute curves.

## 4.2 Discrete gaugings and $E_1$ $\mathcal{N} = 2$ KK theories

Let us now discuss the theories at the bottom right of (128), which are also shown in figure 10. They are all obtained by gauging a discrete 0-form symmetry, $\Gamma^{[0]} = \mathbb{Z}_2^{[0]}$. This is true also for the relative theories. We have:

$$
(E_1)_\bullet^{[1,2]} = (E_1)_\bullet^{[1,0]}/\mathbb{Z}_2^{[0]}, \qquad \mathcal{R}_B = \mathcal{R}_A/\mathbb{Z}_2^{[0]},
\tag{146}
$$

with $\bullet \in \{0, +, -\}$. Discrete gaugings of Seiberg-Witten geometries were discussed in detail in [34], to which we refer for more details. Here, we would like to gauge the accidental $R$-symmetry (131). At the level of the SW curves in Weierstrass normal form, this corresponds to a so-called base change. This amounts to the simple replacement:

$$
g_2(U) \to V^2 g_2\left(\sqrt{V}\right), \qquad g_3(U) \to V^3 g_2\left(\sqrt{V}\right),
\tag{147}
$$

where $V = U^2$ is the new Coulomb branch parameter, and we included a quadratic twist $(g_2, g_3) \to (V^2 g_2, V^3 g_3)$. At $\lambda \neq 0$, the discrete gauging introduces an undeformable $I_0^*$ singularity at the origin, whose low energy dynamics is indeed the $U(1)/\mathbb{Z}_2$ theory [34]. Discrete gaugings as base changes were also discussed in [11], in the case of theories with a four-dimensional UV completion. See [43] for a mathematical classification of possible base changes.

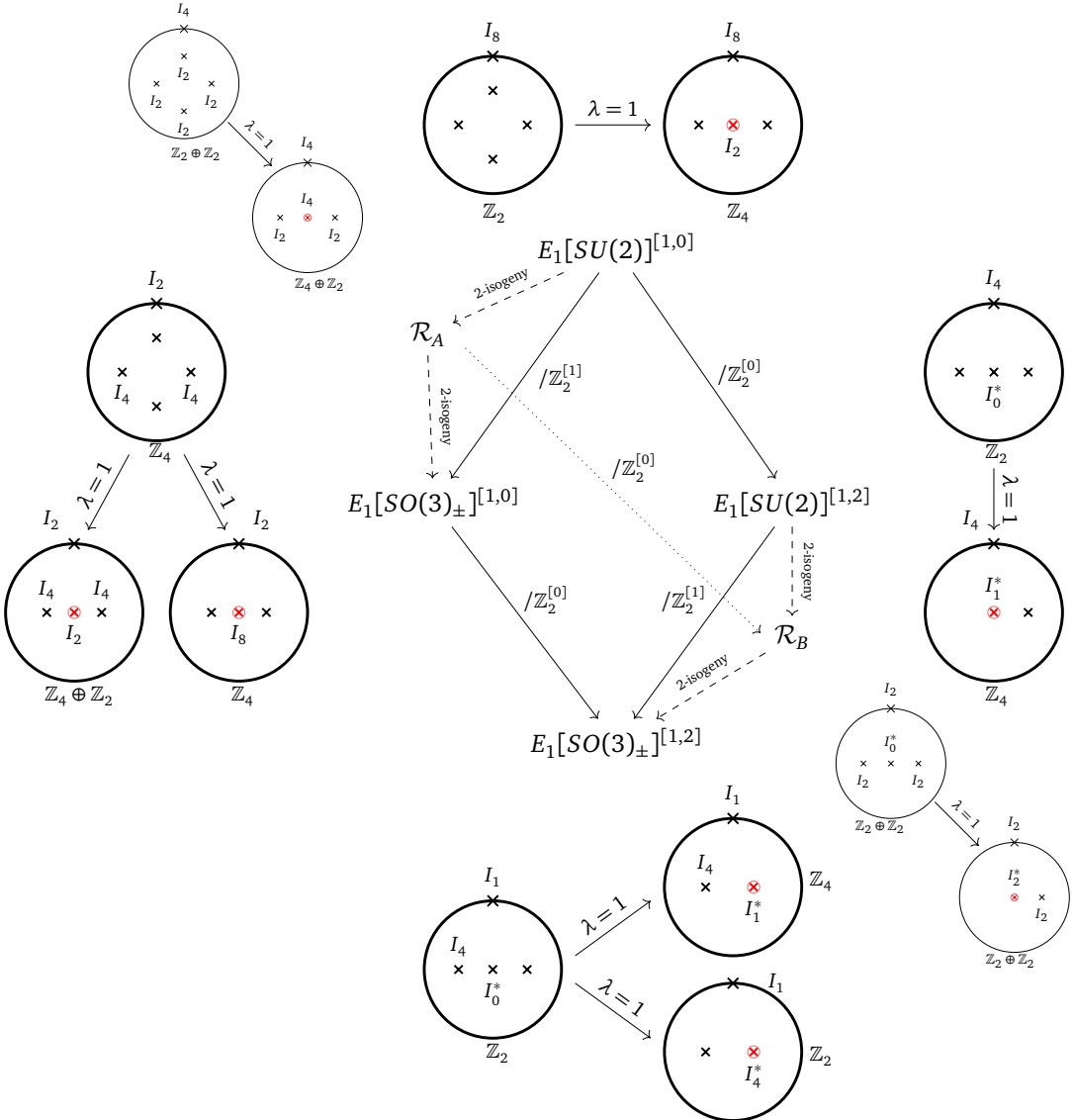

Figure 10: Summary of the relations between the six absolute $E_1$ 4d $\mathcal{N} = 2$ KK theories through gauging. Next to each theory, we give a sketch of the Coulomb branch with its various singularities. We also have the relation $\mathcal{R}_B = \mathcal{R}_A / \mathbb{Z}_2^{[0]}$ between the relative curves, as explained in the main text. For the $SO(3)_\pm$ theories, it is understood that exchanging $SO(3)_+$ with $SO(3)_-$ interchanges the $I_1$ and $I_4$ singularities, which then gives us distinct massless limits (for $\lambda \to 1$) as indicated. The MW torsion is indicated for every rational elliptic surface. The circled singularities are the ones from which a Higgs branch emanates.

**The $E_1[SU(2)]^{[1,2]}$ curve.**   Let us first consider the $\mathbb{Z}_2^{[0]}$ gauging of the $E_1[SU(2)]^{[1,0]}$ theory. The discretely-gauged theory, $E_1[SU(2)]^{[1,2]}$, retains an electric 1-form symmetry and gains a 2-form symmetry $\Gamma^{[2]} = \mathbb{Z}_2^{[2]}$ acting on surface operators. By performing the base change (147), we obtain the Seiberg-Witten curve:

$$g_2^{(E_1)_0^{[1,2]}} = \frac{1}{12}V^2\left(V^2 - 8V(1+\lambda) + 16\left(1 - \lambda + \lambda^2\right)\right),$$
$$g_2^{(E_1)_0^{[1,2]}} = \frac{1}{216}V^3\left(-V^3 + 12V^2(1+\lambda) - 24V\left(2 + \lambda + 2\lambda^2\right) + 32\left(2 - 3\lambda - 3\lambda^2 + 2\lambda^3\right)\right),$$
(148)

as a one-parameter family on the $V$-plane (for a fixed value of $\lambda$), with the discriminant:

$$\Delta^{(E_1)_0^{[1,2]}} = V^6\lambda^2\left(16 - 8V + V^2 - 32\lambda - 8V\lambda + 16\lambda^2\right).$$
(149)

The singularity structure is $(I_2; I_0^*, 2I_1)$. In the limit $\lambda \to 1$, one $I_1$ singularity merges with the $I_0^*$ singularity at $V = 0$, giving us the $(I_2; I_1^*, I_1)$ curve. This curve has a unique $\mathbb{Z}_2$ torsion section:

$$P_{\mathbb{Z}_2} = \left(\frac{1}{12}V(V - 4\lambda - 4), 0\right),$$
(150)

which is simply inherited from (134). The torsion part of the Mordell-Weil group enhances to $\Phi = \mathbb{Z}_4$ at $\lambda = 1$. This is generated by the section:

$$P_{\mathbb{Z}_4} = \left(\frac{1}{12}V(V + 4), -V^2\right),$$
(151)

which descends from (135).

**The relative theory $E_1[\mathcal{R}_B]$.**   By performing a 2-isogeny on the $E_1[SU(2)]^{[1,2]}$ curve along the section (150), we obtain the relative curve $\mathcal{R}_B$:

$$g_2^{\mathcal{R}_A} = \frac{V^2}{48}\left(V^2 - 8V(1+\lambda) + 16\left(1 + 14\lambda + \lambda^2\right)\right),$$
$$g_3^{\mathcal{R}_A} = -\frac{V^3}{1728}\left(V^3 - 12V^2(1+\lambda) + 48V\left(1 - 10\lambda + \lambda^2\right) - 64\left(1 - 33\lambda - 33\lambda^2 + \lambda^3\right)\right),$$
(152)

with a discriminant $\Delta^{\mathcal{R}_A} = \frac{1}{64}\lambda V^6\left(V^2 - 8V + 16 - 32\lambda - 8V\lambda + 16\lambda^2\right)^2$. This can also be obtained by gauging the $\mathbb{Z}_2^{[0]}$ symmetry of the relative theory $\mathcal{R}_A$, as one can readily check at the level of the curves:

$$\mathcal{R}_B = \mathcal{R}_A/\mathbb{Z}_2^{[0]} = E_1[SU(2)]^{[1,2]}/\mathbb{Z}_2^{[1]}.$$
(153)

The singularity structure of this curve is $(I_2; I_0^*, 2I_2)$, with the further degeneration to $(I_2; I_2^*, I_2)$ at $\lambda = 1$. For $\lambda \neq 1$, we have $\Phi_{\text{tor}} = \mathbb{Z}_2 \oplus \mathbb{Z}_2$ generated by:

$$P_1 = \left(-\frac{V}{12}(V - 4\lambda - 4), 0\right), \qquad P_\pm = \left(\frac{V}{24}\left(V - 4\lambda - 4 \pm 24\sqrt{\lambda}\right), 0\right).$$
(154)

**The $E_1[SO(3)_\pm]^{[1,2]}$ curves.**   Given the relative curve $\mathcal{R}_B$, we obtain the three absolute curves through isogenies along (154), namely:

$$E_1[SU(2)]^{[1,2]} \cong \mathcal{R}_B/P_1, \qquad E_1[SO(3)_\pm]^{[1,2]} \cong \mathcal{R}_B/P_\pm.$$
(155)

Equivalently, these curves can be obtained by gauging the $\mathbb{Z}_2^{[0]}$ symmetry of the theories $(E_1)_\bullet^{[1,0]}$ in (139), as summarised in figure 10. The $E_1[SO(3)_\pm]^{[1,2]}$ curves read:

$$g_2^{(E_1)_\pm^{[1,2]}} = \frac{V^2}{192}\left(V^2 - 8V\left(1 \mp 30\lambda^{\frac{1}{2}} + \lambda\right) + 16\left(1 \mp 60\lambda^{\frac{1}{2}} + 134\lambda - 60\lambda^{\frac{3}{2}} + \lambda^2\right)\right),$$

$$g_3^{(E_1)_\pm^{[1,2]}} = \frac{V^3}{13824}\left(-V^3 + 12V^2\left(1 \pm 42\lambda^{\frac{1}{2}} + \lambda\right) - 48V\left(1 \pm 84\lambda^{\frac{1}{2}} - 346\lambda \pm 84\lambda^{\frac{3}{2}} + \lambda^2\right)\right.$$
$$\left. + 64\left(1 \pm 126\lambda^{\frac{1}{2}} - 1041\lambda \pm 1764\lambda^{\frac{3}{2}} - 1041\lambda^2 \pm 126\lambda^{\frac{5}{2}} + \lambda^3\right)\right).$$
$$(156)$$

At $\lambda \neq 1$, we have the singularity structure $(I_1; I_0^*, I_1, I_4)$ in either case, with the $I_1$ and $I_4$ singularity exchanged between the $(E_1)_+^{[1,2]}$ and the $(E_1)_-^{[1,2]}$ curves. At $\lambda = 1$, we have a singularity $I_1^*$ at the origin for $SO(3)_+$ theory, while we have a singularity $I_4^*$ for the $SO(3)_-$ theory. These singularities simply describe the $\mathbb{Z}_2^{[0]}$ gauging of the $I_2$ and $I_8$ singularities of the massless $(E_1)_+^{[1,0]}$ curves. Indeed, given a $I_{2n}$ singularity with a $I_{2n} \to 2I_n$ deformation pattern, which describes a $U(1)$ theory with two massless hypermultiplets of charge $\sqrt{n}$, gauging a $\mathbb{Z}_2$ $R$-symmetry (together with an appropriate $S$-duality to preserve supersymmetry) gives us an $I_n^*$ singularity [34].

In summary, we have shown that there exist six distinct absolute 4d $\mathcal{N} = 2$ KK theories that correspond to the rank-one 5d SCFT $E_1$ on a circle, and we have provided the Seiberg-Witten geometry for each of them. Moreover, there exists two distinct relative theories, $\mathcal{R}_A$ and $\mathcal{R}_B$, which are related by a discrete gauging. Of the six absolute theories, only two have a straightforward 5d uplift. Indeed, $E_1[SU(2)]^{[1,0]}$ is the direct dimensional reduction of the $E_1$ SCFT with the electric 1-form symmetry $\Gamma_{5d}^{[1]}$, while $E_1[SO(3)_+]^{[1,2]}$ is the direct dimensional of the $E_1$ SCFT with the magnetic 2-form symmetry $\Gamma_{5d}^{[2]}$:

$$E_1[SU(2)]^{[1,0]} \equiv D_{S^1}E_1^{[1]}, \qquad E_1[SO(3)_+]^{[1,2]} \equiv D_{S^1}E_1^{[2]}. \qquad (157)$$

In the M-theory geometric engineering of these theories, the electric lines charged under $\Gamma_{5d}^{[1]}$ arise from M2-branes wrapping relative 2-cycles, and the magnetic surface operators charged under $\Gamma_{5d}^{[2]}$ arise from M5-branes wrapping relative 4-cycles [44,45].

# 5 Global structures of the $E_0$ theory

In addition to the $E_1$ theory, there is only one more rank-one 5d SCFT with a non-trivial one-form symmetry. This is the $E_0$ theory, which is geometrically engineered in M-theory at the complex cone over $\mathbb{P}^2$ [46] and possesses a 1-form symmetry $\Gamma_{5d}^{[1]} = \mathbb{Z}_3$ [44,45]. This 5d SCFT has no continuous flavour symmetry, hence it does not admit any relevant deformation [47].

The Seiberg-Witten curve for the $E_0$ theory on a circle is often discussed in the context of topological string theory on the local $\mathbb{P}^2$ geometry, since it gives its mirror Type-IIB description [48,49] – see *e.g.* [50–54] for various related approaches and results. Interestingly, there are two physically-distinct presentation of the one-dimensional moduli space of local $\mathbb{P}^2$.[13] One can parameterise it by the $E_0$ Coulomb-branch parameter $U$, which results in a moduli space with three conifold singularities that are rotated by a $\mathbb{Z}_3$ 0-form symmetry [7], or one can gauge this $\mathbb{Z}_3^{[0]}$ symmetry to obtain a moduli space described by a parameter $V = U^3$. On

---

[13]Here we mean the complexified Kähler moduli space in IIA, quantum-corrected by worldsheet instantons. By mirror symmetry, this is equivalent to the complex-structure moduli space of the SW curve.

the $V$-plane, we have one conifold and one $\mathbb{Z}_3$ orbifold singularity (this is in the string-theory terminology – the conifold point is an $I_1$ singularity in the mirror curve). This latter description is most often encountered in the topological-string literature. Both descriptions are physical and correspond to different global forms of the 4d $\mathcal{N} = 2$ KK theory $D_{S^1}E_0$, as we now explain.

As it regards to the possible 4d global structures, the most important physical difference between the $E_1$ and the $E_0$ theory is that the 5d 1-form symmetry of the latter is anomalous [55]. There is a cubic anomaly for $\Gamma_{5d}^{[1]} = \mathbb{Z}_3$ (corresponding to a 6d term of the schematic form $B^3$ for the background gauge field $B$), which gives us a mixed anomaly between $\mathbb{Z}_3^{[0]}$ and $\mathbb{Z}_3^{[1]}$ in the 4d KK theory.[14] Hence, we can still gauge $\mathbb{Z}_3^{[1]}$ or $\mathbb{Z}_3^{[0]}$ in 4d, but not both at the same time. This mixed anomaly should be reflected in the allowed global structure as seen by the SW curves. Starting from the KK theory for the 'electric' form of the $E_0$ theory in 5d, denoted by $(E_0)_e^{[1,0]}$, the diagram (127) truncates, and we expect:

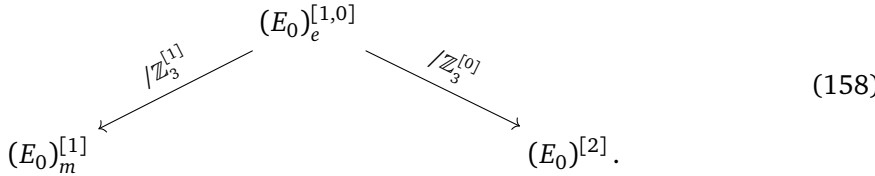

$$\tag{158}$$

Indeed, we will show that there exists three distinct 'magnetic' KK theories, $(E_0)_m^{[1]} = (E_0)_e^{[1,0]}/\mathbb{Z}_3^{[1]}$, with a magnetic 1-form symmetry and no 0-form symmetry, and that there exists a unique discreetly gauged theory, $(E_0)^{[2]} = (E_0)_e^{[1,0]}/\mathbb{Z}_3^{[0]}$, which has no "gauge-able" 1-form symmetry and whose CB is the $V$-plane mentioned in the previous paragraph.

## 5.1 Absolute and relative $E_0$ curves

**Absolute electric curve.** Let us start our discussion with the 5d $E_0$ 'electric' theory on a circle, whose Coulomb branch we already studied in [7] – it was first studied in [37]. The theory does not admit any gauge-theory deformation, nor any relevant deformations at all. It has a SW geometry given by:

$$g_2^{(E_0)_e^{[1,0]}} = \frac{3}{4}U\left(9U^3 - 8\right), \qquad g_3^{(E_0)_e^{[1,0]}} = -\frac{1}{8}\left(27U^6 - 36U^3 + 8\right), \tag{159}$$

with discriminant $\Delta^{(E_0)_e^{[1,0]}} = 27(U^3 - 1)$. We thus have the singularity structure $(I_9; 3I_1)$. The KK theory has a $\mathbb{Z}_3^{[1]}$ 1-form symmetry encoded in the MW group of the SW geometry, $\Phi(\mathcal{S}) = \mathbb{Z}_3$. It also has a $\mathbb{Z}_3^{[0]}$ 0-form symmetry that is spontaneously broken on the Coulomb branch, arising as an accidental $R$-symmetry that rotates the three $I_1$ singularities. Note also that this curve is modular, with monodromy group $\Gamma^0(9)$ [7, 8], and that we have:

$$U(\tau) = 1 + \frac{1}{3}\left(\frac{\eta\left(\frac{\tau}{9}\right)}{\eta(\tau)}\right)^3. \tag{160}$$

Given a natural choice of fundamental domain of $\Gamma^0(9)$, one finds a possible basis of light BPS states:

$$\gamma_1 = (1, 0), \qquad \gamma_2 = (-1, 3), \qquad \gamma_3 = (1, -6). \tag{161}$$

---

[14]The 5d anomaly theory has the schematic form $\int B_2^2 B_1$, where $B_2$ and $B_1$ are background gauge fields for $\mathbb{Z}_2^{[1]}$ and $\mathbb{Z}_2^{[0]}$, respectively.

The corresponding BPS quiver reads:

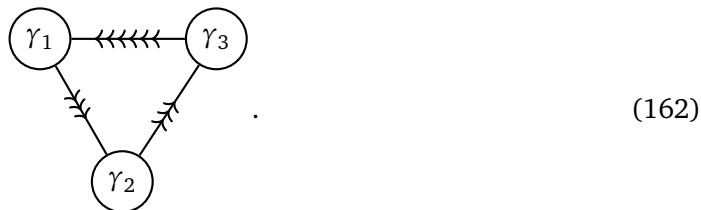

$$(162)$$

The $\mathbb{Z}_3$ torsion sections of the SW curve are given by:

$$P_1 = \left(\frac{3}{4}U^2, 1\right), \qquad P_2 = \left(\frac{3}{4}U^2, -1\right), \tag{163}$$

which satisfy $P_i + P_j = P_{i+j \bmod 3}$, with $P_0 = \mathcal{O}$ the zero-section.

**Gauging $\mathbb{Z}_3^{[1]}$ along 3-isogenies.** Given any RES with a 3-torsion section $P_1$, which generates a $\mathbb{Z}_3 \subset \Phi_{\text{tor}}$ with:

$$P_1 = (x_0, y_0), \qquad P_2 = (x_0, -y_0), \tag{164}$$

we can then construct a 3-isogeny along these sections. The explicit form of the new curve is again given by the Vélu formula, as reviewed in appendix A.2. This reads:

$$g'_2 = -g_2 + \frac{40x_0^2}{3}, \qquad g'_3 = \frac{1}{27}\left(g_3 - 14g_2 x_0 + 28\left(6x_0^3 + y_0^2\right)\right), \tag{165}$$

where we included the rescaling by $\alpha = \sqrt{3}$ in order to preserve the Dirac pairing.

**Relative $E_0$ curve.** Starting with the absolute $(E_0)_e^{[1,0]}$ curve (159) and quotienting along the sections (163), we obtain the relative curve for the $E_0$ theory:

$$g_2^{(E_0)^{\text{rel}}} = \frac{3}{4}U\left(U^3 + 8\right), \qquad g_3^{(E_0)^{\text{rel}}} = -\frac{1}{8}\left(U^6 - 20U^3 - 8\right). \tag{166}$$

Its discriminant is:

$$\Delta^{(E_0)^{\text{rel}}} = 27\left(U^3 - 1\right)^3. \tag{167}$$

Hence the $U$-plane of the relative theory has the singularity structure $(I_3; 3I_3)$, with three undeformable $I_3$ singularities at $U^3 = 1$. This is again modular, with modular group $\Gamma(3)$ which is conjugate to $\Gamma^0(9)$ in $\text{PSL}(2, \mathbb{R})$. We immediately find the modular function from (160), upon using the fact that $\tau_{\text{rel}} = \frac{\tau}{3}$:

$$U(\tau) = 1 + \frac{1}{3}\left(\frac{\eta\left(\frac{\tau}{3}\right)}{\eta(3\tau)}\right)^3. \tag{168}$$

Moreover, a basis of BPS states preserving the Dirac pairing now reads:

$$\gamma_1 = \sqrt{3}(1, 0), \qquad \gamma_2 = \sqrt{3}(-1, 1), \qquad \gamma_3 = \sqrt{3}(1, -2), \tag{169}$$

which can be also determined from the standard fundamental domain of $\Gamma(3)$ [8]. Note, in particular, that under this 3-isogeny the periods are mapped as $(a_D^{\text{rel}}, a^{\text{rel}}) = \left(\frac{1}{\sqrt{3}}a_D, \sqrt{3}a\right)$. The relative curve (166) has the MW group $\Phi(\mathcal{S}_{\text{rel}}) = \mathbb{Z}_3 \oplus \mathbb{Z}_3$. This generated by the torsion sections:

$$P_1 = \left(-\frac{3}{4}U^2, i(U^3 - 1)\right), \qquad P_2 = \left(-\frac{3}{4}U^2, -i(U^3 - 1)\right), \tag{170}$$

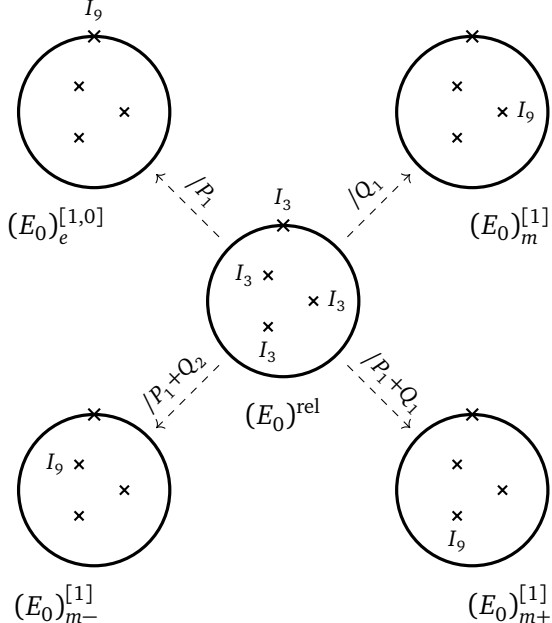

Figure 11: Structure of the $U$-plane and gauging of the one-form symmetry for the $D_{S^1}E_0$ theory. The central configuration is the $U$-plane of the relative curve, with $\mathbb{Z}_3 \oplus \mathbb{Z}_3$ torsion, from which the four absolute curves can be obtained by a 3-isogeny. Note that all absolute curves have $\mathbb{Z}_3$ torsion.

and:

$$Q_1 = \left(\frac{1}{4}(U+2)^2, \sqrt{3}(U^2+U+1)\right), \qquad Q_2 = \left(\frac{1}{4}(U+2)^2, -\sqrt{3}(U^2+U+1)\right), \quad (171)$$

which each span a $\mathbb{Z}_3$ subgroup of $\Phi(\mathcal{S}_{\text{rel}})$. There are two other $\mathbb{Z}_3$ subgroups of $\Phi(\mathcal{S}_{\text{rel}})$, given by $\{P_1+Q_1, P_2+Q_2\}$, and $\{P_1+Q_2, P_2+Q_1\}$, respectively. We can then consider 3-isogenies generated by these torsion sections. First, a 3-isogeny of the relative curve along $\{P_1, P_2\}$ in (170) returns us back to the $E_0$ curve (159). For the other three cases, we find a new 'magnetic' curve. This is depicted in figure 11.

**The 'magnetic' curve $(E_0)_e^{[1,0]}/\mathbb{Z}_3^{[1]} \equiv (E_0)_m^{[1]}$.** Let us now perform a 3-isogeny of the relative curve along $\{Q_1, Q_2\}$ in (171). (The other two possibilities lead to similar results.) This gives us the new curve:

$$\begin{aligned}
g_2^{(E_0)_m^{[1]}} &= \frac{1}{12}\left(U^4 + 80U^3 + 240U^2 + 248U + 160\right), \\
g_3^{(E_0)_m^{[1]}} &= \frac{1}{216}\left(-U^6 + 168U^5 + 1848U^4 + 4556U^3 + 6384U^2 + 4704U + 2024\right),
\end{aligned} \quad (172)$$

which has a discriminant:

$$\Delta^{(E_0)_m^{[1]}} = \frac{1}{3}(U-1)^9\left(U^2+U+1\right). \quad (173)$$

The singularity structure on this $U$-plane is $(I_1; I_9, 2I_1)$, with the $I_9$ at $U=1$ and the $I_1$ singularities at $U = e^{\pm\frac{2\pi i}{3}}$. In particular, we see that we have lost the $\mathbb{Z}_3^{[0]}$ accidental $R$-symmetry, as expected from our general discussion above. Instead, the would-be $\mathbb{Z}_3^{[0]}$ symmetry acting on $U$ would give us the other two magnetic curves, as shown in figure 11. (This is similar to the

case of the pure $SO(3)_{\pm}$ gauge theory, in which case we had a mixed anomaly between the 0-form and 1-form $\mathbb{Z}_2$ symmetries of the $SU(2)$ gauge theory.)

This particular 3-isogeny leads to $\tau_m = \frac{\tau_{\text{rel}}}{3}$, from which we can immediately find an expression for $U = U(\tau)$, as before. The new monodromy group is $\Gamma_0(9)$, which is conjugate to $\Gamma^0(9)$ (by **S** in $\text{PSL}(2, \mathbb{Z})$). Thus, a basis of BPS states preserving the Dirac pairing reads:

$$\gamma_1 = (3, 0), \qquad \gamma_2 = (-3, 1), \qquad \gamma_3 = (3, -2). \tag{174}$$

Note that the magnetic-electric charges and periods in the 'electric' and 'magnetic' theories, $(E_0)_e^{[1,0]}$ and $(E_0)_m^{[1]}$, are related by:

$$(m_{D,m}, q_m) = \left(3m_{D,e}, \frac{q_e}{3}\right), \qquad (a_{D,m}, a_m) = \left(\frac{a_{D,e}}{3}, 3a_e\right). \tag{175}$$

The charge normalisation for the other two magnetic theories shown in figure 11 can also be understood in terms of the Witten effect, and one finds:

$$(m_{D,m\pm}, q_{m\pm}) = \left(3m_{D,e}, \frac{q_m}{3} \mp m_{D,e}\right), \qquad (a_{D,m}, a_m) = \left(\frac{a_{D,e}}{3} \pm a_e, 3a_e\right). \tag{176}$$

In each of the three magnetic normalisation, the magnetic line $(1, 0)$ is unscreened, and it is thus charged under the magnetic one-form symmetry $\mathbb{Z}_3^{[1]}$. The latter is responsible for the SW curve (172) having $\mathbb{Z}_3$ torsion. Indeed, its SW geometry corresponds to the same rational elliptic surface as the one for $(E_0)_e^{[1,0]}$, but with a different fiber at infinity. The torsion sections read:

$$P_1 = \left(-\frac{1}{4}(U+2)^2, \frac{i}{3\sqrt{3}}(U-1)^3\right), \qquad P_2 = \left(-\frac{1}{4}(U+2)^2, -\frac{i}{3\sqrt{3}}(U-1)^3\right). \tag{177}$$

One can check that the 3-isogeny along $\{P_1, P_2\}$ leads us back to the relative curve (166).

## 5.2 Gauging $\mathbb{Z}_3^{[0]}$: The $\mathbb{Z}_3$ orbifold point on the $V$-plane

As we have already seen, the SW curve for the $(E_0)_e^{[1,0]}$ theory has a $\mathbb{Z}_3$ 0-form symmetry, which acts on the CB by exchanging the three $I_1$ cusps. This symmetry is inherited from the 5d one-form symmetry, and can be gauged by performing a base change, as in (147):

$$g_2(U) \to V^{\frac{8}{3}} g_2\left(V^{\frac{1}{3}}\right), \qquad g_3(U) \to V^4 g_2\left(V^{\frac{1}{3}}\right), \tag{178}$$

where $V = U^3$ and we again introduced a quadratic twist to maintain the fiber at infinity as $F_\infty = I_3$.

$$g_2^{(E_0)^{[2]}} = \frac{3}{4} V^3 (9V - 8), \qquad g_3^{(E_0)^{[2]}} = -\frac{1}{8} V^4 \left(27V^2 - 36V + 8\right), \tag{179}$$

with discriminant $\Delta^{(E_0)^{[2]}} = 27(V-1)V^8$. The resulting CB geometry on the $V$-plane has singular fibers $(I_3; IV^*, I_1)$. This is the same configuration as the massless $E_6$ curve [7,37], but the interpretation of the $IV^*$ here is very different. The $IV^*$ is undeformable, and corresponds simply to a $\mathbb{Z}_3^{[0]}$ discrete gauging of the free $U(1)$ gauge theory at $U = 0$ [34]. This $IV^*$ singularity is the standard $\mathbb{C}^3/\mathbb{Z}_3$ orbifold point on the Kähler moduli space of the local $\mathbb{P}^2$ geometry in Type IIA string theory.

Given our discussion above, a slightly puzzling feature of the $(E_0)^{[2]}$ curve is that it still has a non-trivial MW group, $\Phi(\mathcal{S}) = \mathbb{Z}_3$, generated by the sections:

$$P_1 = \left(\frac{3}{4}V^2, V^2\right), \qquad P_2 = \left(\frac{3}{4}V^2, -V^2\right). \tag{180}$$

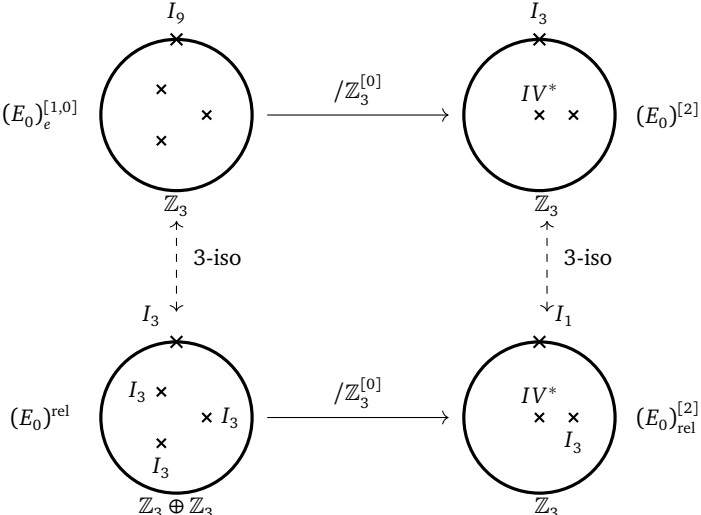

Figure 12: Structure of the $U$-plane versus $V$-plane, and gauging of the zero-form symmetry for the $D_{S^1}E_0$ theory.

Interestingly, a 3-isogeny along $\{P_1, P_2\}$ leads to the same rational elliptic surface with a distinct fiber at infinity, namely the $(I_1; IV^*, I_3)$ configuration. It reads:

$$g_2^{(E_0)_{\text{rel}}^{[2]}} = \frac{3}{4}V^3(V + 8), \qquad g_3^{(E_0)_{\text{rel}}^{[2]}} = -\frac{1}{8}V^4\left(V^2 - 20V - 8\right), \tag{181}$$

with discriminant $\Delta^{(E_0)_{\text{rel}}^{[2]}} = 27(V - 1)^3 V^8$. This is still interpreted as a 'relative curve', in the sense that it can also be obtained as the $\mathbb{Z}_3$ folding (178) of the $U$-plane for the relative curve (166) – see figure 12. Unlike the proper relative curve, however, the torsion group of $(E_0)_{\text{rel}}^{[2]}$ is still $\mathbb{Z}_3$ (not $\mathbb{Z}_3 \oplus \mathbb{Z}_3$). We tentatively interpret this situation as follows. The 'electric' theory $(E_0)_e^{[1,0]}$ has a mixed anomaly $B_2^2 B_1$, so that the anomalous variation of the theory under the $p$-form symmetries $\mathbb{Z}_3^{[p]}$, denoted schematically by $\delta_{(p)}S$, takes the form $\delta_{(0)}S \sim \lambda_0 B_2^2$ and $\delta_{(1)}S \sim \lambda_1 B_2 B_1$. Hence, if we gauge $\mathbb{Z}_3^{[1]}$ (making $B_2$ dynamical), the $\mathbb{Z}_3^{[0]}$ symmetry disappears entirely, but if we gauge $\mathbb{Z}_3^{[0]}$ (making $B_1$ dynamical) there still exists a $\mathbb{Z}_3^{[1]}$ symmetry if we set $B_2 = 0$. We call this situation a "non-gaugeable" $\mathbb{Z}_3^{[1]}$ symmetry. It would be desirable to understand this rather subtle point much better.

# 6 Six-dimensional M-string theory on $T^2$

In section 4, we studied the global forms of the $E_1$ theory, which is the UV completion of the 5d $\mathcal{N} = 1$ $SU(2)$ gauge theory. The next natural step is to consider coupling the 5d gauge theory to an adjoint hypermultiplet, which preserves the five-dimensional one-form symmetry. This enhances the supersymmetry to 5d $\mathcal{N} = 2$ for a massless adjoint; the theory with a massive adjoint is called the 5d $\mathcal{N} = 1^*$ theory. It is well-known, however, that the UV completion of the 5d $\mathcal{N} = 2$ theory is the six-dimensional $\mathcal{N} = (2, 0)$ $A_1$ SCFT [56], also known as the (rank-one) M-string theory – see e.g. [57–59]. The Seiberg-Witten curve for the M-string theory has been discussed in the context of integrable systems – see e.g [36, 60, 61].

The 6d $\mathcal{N} = (2, 0)$ theory on a torus will be denoted by:

$$M[G]^{[p,q]} \equiv D_{T^2}[\text{6d M-string SCFT}], \tag{182}$$

with the same notation as in section 4.1. This is now a 4d $\mathcal{N} = 2$ theory with two distinct KK charges. The effective theory also depends explicitly on the modular parameter, $\tau_{\mathrm{uv}}$, of the compactification torus. The 6d $\mathcal{N} = (2,0)$ theory is intrinsically a relative theory. It has a 2-form symmetry which is 'self-dual' so that, upon circle compactification to 5d, we must choose either the 5d $\mathcal{N} = 1^*$ theory with the 1-form symmetry (the $SU(2)$ theory) or the one with the 2-form symmetry (the $SO(3)$ theory) – see *e.g.* [62,63] for recent discussions. Hence, the allowed global structures of the 4d $\mathcal{N} = 2$ KK theory for the M-string theory are the same as for the $E_1$ theory, and we can draw the same diagram relating the allowed global forms of the 4d KK theory:

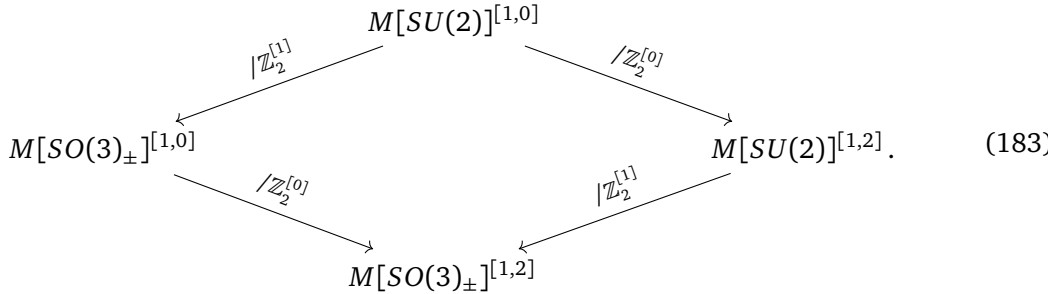

(183)

There is an important difference, however, in that the three theories with fixed 4d $[p,q]$ symmetry are actually one and the same theory in different $S$-duality frames, just like for the 4d $\mathcal{N} = 2^*$ theory discussed in section 3. In the rest of this section, we briefly explore this structure. We should also note that all other $\mathfrak{su}(2)$ SW curves discussed in this paper can be obtained from the M-string SW curve by taking appropriate limits. This is summarised in figure 13 below.

## 6.1 Relative curve $\mathcal{R}_A^M$ for the M-string

We start by presenting the relative curve for the M-string theory in Weierstrass normal form. It is most compactly written as:

$$
g_2^{\mathcal{R}_A^M} = \frac{1}{27} \left( 9E_4 \left( \widetilde{U}^2 - 1 \right)^2 - 6E_6 M \left( \widetilde{U}^2 - 1 \right) + E_4^2 M \right),
$$
$$
g_3^{\mathcal{R}_A^M} = -\frac{1}{729} \left( 27E_6 \left( \widetilde{U}^2 - 1 \right)^3 - 27E_4^2 M \left( \widetilde{U}^2 - 1 \right)^2 + 9E_4 E_6 M^2 \left( \widetilde{U}^2 - 1 \right) - \left( 2E_6^2 - E_4^3 \right) M^3 \right).
$$
(184)

Here, $\widetilde{U}$ is a dimensionless Coulomb branch parameter, and the mass parameter $M$ is related to the actual mass, $m$, as:

$$
M(i\beta m; \tau_{\mathrm{uv}}) = \frac{1}{\wp(i\beta m; \tau_{\mathrm{uv}})},
$$
(185)

in terms of the Weierstrass $\wp$ function for the compactification torus. Since $m$ arises as a background flat connection on $T^2$, we have the periodicities:

$$
i\beta m \sim i\beta m + 1 \sim i\beta m + \tau_{\mathrm{uv}}.
$$
(186)

We shall still call $\tau_{\mathrm{uv}}$ the UV gauge coupling. Note also that $M(0; \tau_{\mathrm{uv}}) = 0$. The discriminant of the relative curve (184) reads:

$$
\Delta^{\mathcal{R}_A^M} = \Delta_0(\tau_{\mathrm{uv}}) \prod_{i=1}^{3} \left( \widetilde{U}^2 - 1 + e_i M \right)^2, \qquad \Delta_0(\tau_{\mathrm{uv}}) \equiv \frac{1}{27} \left( E_4^3 - E_6^2 \right),
$$
(187)

with $\Delta_0(\tau_{\mathrm{uv}})$ defined as in (82). Hence, the $\widetilde{U}$-plane has the singularity structure $(I_0; 6I_2)$, which specialises to $(I_0; 2I_0^*)$ in the massless limit. The fact that we have a $I_0$ fiber (*i.e.* a

smooth elliptic fiber) at infinity is because of the 6d UV completion [37]. The bulk singularity structure is also expected from the point of view of the holonomy saddles of the 5d $\mathcal{N} = 1^*$ gauge theory, since it corresponds to two 'copies' of the 4d $\mathcal{N} = 2^*$ theory [40]. For generic mass parameter, we thus have six $I_2$ singularities.[15] A very interesting limit arises at $M \to 1/e_i$, namely for the mass parameter $m$ hitting the half-periods of the compactification torus:

$$i\beta m \in \left\{ \frac{1}{2}, \frac{\tau_{\text{uv}}}{2}, \frac{\tau_{\text{uv}} + 1}{2} \right\}. \tag{188}$$

In such a limit, two $I_2$ singularities from the two distinct 'holonomy saddles' merge into a single $I_4$, giving us the singularity structure $(I_0; I_4, 4I_2)$.

**S-duality invariance.** The M-string theory relative curve is invariant under the full $SL(2, \mathbb{Z})$ group acting on the UV gauge coupling, exactly as was the case for the $\mathcal{N} = 2^*$ relative curve. Under **S**, we have:

$$\mathbf{S} \; : \; i\beta m \mapsto \frac{i\beta m}{\tau_{\text{uv}}}, \tag{189}$$

while $\widetilde{U}$ is S-duality invariant. Then, $M$ transforms as a modular form of weight $-2$:

$$\mathbf{S} \; : \; M(i\beta m; \tau_{\text{uv}}) \to \tau_{\text{uv}}^{-2} M(i\beta m; \tau_{\text{uv}}), \tag{190}$$

so that $g_2^{\mathcal{R}_A^M}$ and $g_3^{\mathcal{R}_A^M}$ transform as modular forms of weight 4 and 6, respectively.

### 6.1.1 Massive limit and 4d limits

**The $E_1$ limit.** The M-string theory should have a 5d limit corresponding to integrating out the adjoint (in the 5d $\mathcal{N} = 1^*$ description). This limit is well-understood from the point of view of integrable systems. The M-string theory SW geometry corresponds to the Ruijsenaars-Schneider (a.k.a. relativistic Calogero-Moser) integrable system, the $E_1$ SW geometry corresponds to the relativistic Toda system, and one can obtain the latter from the former [36]. Introducing the parameter $y_m \equiv e^{-2\pi\beta m}$, we take the limit:

$$q_{\text{uv}} \to 0, \qquad y \to \infty, \qquad \sqrt{\lambda} \equiv q_{\text{uv}}^{\frac{1}{2}} y \;\; \text{fixed}. \tag{191}$$

Then, a direct computation (using equation (A.43) in appendix) shows that:

$$M \to -3 + \frac{36(1 + \lambda)}{\sqrt{\lambda}} q^{\frac{1}{2}} + O(q), \tag{192}$$

in this limit. Let us also rescale the parameter $\widetilde{U}$ and the SW curve itself as:

$$\widetilde{U} = \sqrt{3} \left( \frac{q_{\text{uv}}}{\lambda} \right)^{\frac{1}{4}} U, \qquad (g_2, g_3) \to \left( \alpha^{-4} g_2, \alpha^{-6} g_3 \right), \qquad \alpha = 2\sqrt{3} \left( \frac{q_{\text{uv}}}{\lambda} \right)^{\frac{1}{4}}, \tag{193}$$

keeping $U$ finite. It is straightforward to check that, in this large-mass limit, the relative curve (184) reduces to the relative curve $\mathcal{R}_A$ for the $E_1$ theory (as given in (136) with $\lambda$ the 5d gauge coupling parameter). This limit realises the transition:

$$\mathcal{R}_A^M \to \mathcal{R}_A \; : \; (I_0; 6I_2) \to (I_4; 4I_2), \tag{194}$$

by sending the two $I_2$ singularities that sit at $\widetilde{U}^2 = 1 - e_1 M$, which correspond to the two holonomy-saddle copies of the 5d adjoint hypermultiplet, to infinity.

---

[15]It is useful to note that the curve (184) can be obtained from the 4d $\mathcal{N} = 2^*$ relative curve (81) by the substitution $\widetilde{u} \to \widetilde{U}^2 - 1$ and $m^2 \to 4M$. This is related to the fact that, from the point of view of the associated integrable systems, the 5d curves are 'relativistic uplifts' of the 4d curves [36].

$$M[G]^{[p,q]} \xrightarrow[\lambda \equiv q_{\mathrm{uv}} y^2]{q_{\mathrm{uv}} \to 0,\, m \to \infty} E_1[G]^{[p,q]}$$

$$\downarrow \beta \to 0 \qquad\qquad\qquad\qquad \downarrow \beta \to 0$$

$$\text{4d } \mathcal{N} = 2^* \, G \xrightarrow[\Lambda^4 \equiv 4m^2 q_{\mathrm{uv}}]{q_{\mathrm{uv}} \to 0,\, m \to \infty} \text{4d } \mathcal{N} = 2 \, G$$

Figure 13: Relations between the various $\mathfrak{su}(2)$ theories in 4d, 5d and 6d.

**The 4d $\mathcal{N} = 2^*$ limit.** Another important limit is the $\beta \to 0$ limit that reduces the M-string theory to the 4d $\mathcal{N} = 2^*$ theory. (This corresponds to the Calogero-Moser integrable system.) At the level of the curve, this corresponds to the deformation $(I_0; 6I_2) \to (I_0^*; 3I_2)$. Indeed, the relative 4d $\mathcal{N} = 2$ curve (82) can be obtained from (184) in the limit:

$$\beta \to 0 \,, \qquad \widetilde{U} \approx 1 - 2\beta^2 \widetilde{u} \,, \qquad M \approx -\beta^2 m^2 \,, \qquad \alpha = 2i\beta \,, \tag{195}$$

with $\alpha$ a rescaling of the curve. This limit splits each pair of singularities at $\widetilde{U}^2 = 1 - e_i M$, sending the singularities $\widetilde{U} = -\sqrt{1 - e_i M}$ to infinity.

**The $q_{\mathrm{uv}} = 0$ limit.** Given that the $I_2$ singularities of the relative $\mathcal{R}_A^M$ curve lie at $\widetilde{U}^2 = 1 - e_i M$, we can 'collide' more singularities together by considering some slightly *ad hoc* limits. Consider, for instance, the limit where $e_2(\tau_{\mathrm{uv}}) = e_3(\tau_{\mathrm{uv}})$, which happens for $q_{\mathrm{uv}} = e^{2\pi i \tau_{\mathrm{uv}}} = 0$. More precisely, we have $e_1 = \frac{2}{3}$ and $e_2 = e_3 = -\frac{1}{3}$, as can be seen from the series expansion (94). Similar limits exist for $\tau = 0$ or other points in the $SL(2, \mathbb{Z})$ orbit of $\tau = i\infty$.

To understand this limit, it is useful to consider the M-theory picture that engineers this 6d theory. That is, the M-string theory is the theory living on the worldvolume of two parallel M5-branes, with the 'M-string' being the M2-brane excitation stretched between the M5-branes. By M-theory/IIB duality, this can be also viewed as M-theory compactified on a threefold that is locally of the type $T^2 \times \mathbb{C}^2/Z_2$ [57]. Let $\beta \equiv \beta_5$ and $\widetilde{\beta} \equiv \beta_6$ be the radii of circles in $T^2$. Then, we have [57]:

$$\tau_{\mathrm{uv}} = i \frac{\widetilde{\beta}}{\beta} \,. \tag{196}$$

As a result, the $\tau_{\mathrm{uv}} \to i\infty$ limit is equivalent to $\widetilde{\beta} \to \infty$, where the size of the 6d circle blows up. We can further combine this limit with $M = 1/e_2$, leading to the configuration of singular fibers $(I_0; I_2^*, 2I_2)$. This configuration can likely help explain the $\mathfrak{sp}(2) \cong C_2$ flavour root system associated to the SW geometry at generic values of the mass. We refer to [11] for the definition of the flavour root system of the SW geometry and to [15] for the specific example at hand. While the $\mathfrak{sp}(2)$ flavour symmetry cannot be explained by the 'massless' configuration $(I_0; 2I_0^*)$, which only has a $A_1 \oplus A_1$ flavour symmetry manifest (essentially from the two holomoy saddles), the limit we just considered has an $I_2^*$ singularity whose 4d infrared interpretation is that of an IR-free theory with an $\mathfrak{su}(2)$ vector multiplet coupled to two adjoint hypermultiplets, thus enjoying an $\mathfrak{sp}(2)$ flavour symmetry. It would be desirable to better understand this subtle point.[16]

---

[16] One potential worry is that, in this limit, the $\Delta_0(\tau_{\mathrm{uv}})$ factor in the discriminant (187) vanishes; nonetheless, the 'physical discriminant' [64] obtained by factoring out this piece remains well-defined.

### 6.1.2 BPS states and 6d BPS quiver

While the CB geometry $(I_0; 6I_2)$ is not modular, we can still use the general structure of the CB singularities to derive a BPS quiver for the 4d $\mathcal{N} = 2$ KK theory – this gives us a "6d BPS quiver" for the M-string theory, in the same sense that the BPS quivers for 5d SCFTs on a circle are called 5d BPS quivers [16]. To assign magnetic-electric charges to the $I_2$ singularities, we consider the limit in which the six singularities are grouped into two holonomy saddles [40]. From the point of view of an observer that sits away from both sets of singularities on the $\widetilde{U}$-plane, we then have two copies of the 4d $\mathcal{N} = 2^*$ charges (84), with the two saddles distinguished by a sign flip:

$$\begin{aligned}
\gamma_{A,1} &= \sqrt{2}(0,-1), & \gamma_{M,1} &= \sqrt{2}(1,0), & \gamma_{D,1} &= \sqrt{2}(-1,1), \\
\gamma_{A,2} &= \sqrt{2}(0,1), & \gamma_{M,2} &= \sqrt{2}(-1,0), & \gamma_{D,2} &= \sqrt{2}(1,-1).
\end{aligned} \tag{197}$$

This gives us the 6d BPS quiver:

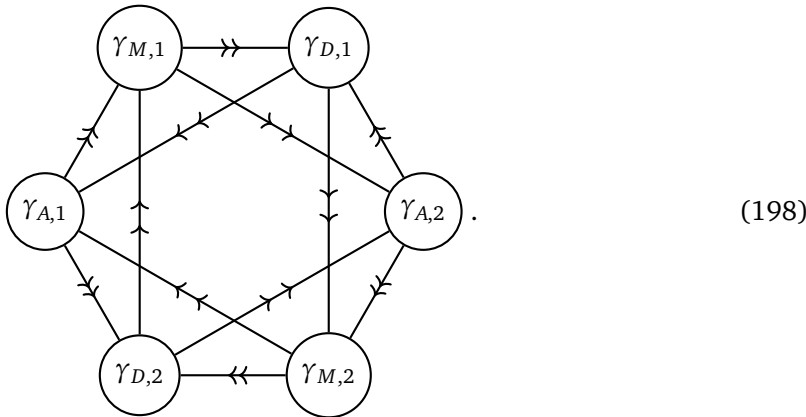

$$. \tag{198}$$

Note that this quiver contains the BPS quiver for the $E_1$ theory, which is obtained by deleting the nodes $\gamma_{A,1}$ and $\gamma_{A,2}$.[17] This 6d BPS quiver passes a number of consistency checks; for instance, one can easily check that the cokernel of the incidence matrix encodes the $\mathbb{Z}_2 \oplus \mathbb{Z}_2$ defect group, as expected on general grounds [14]. We hope to further study this quiver in future work.

## 6.2 Absolute curves for the M-string theory

5d $\mathfrak{su}(2)$ gauge theories with adjoint matter should preserve the $\mathbb{Z}_2^{[1]}$ 1-form center symmetry. Thus, as before, the KK theory corresponding to 5d $\mathcal{N} = 1^*$ will have both a 0 and a 1-form symmetry. In fact, the 6d $\mathcal{N} = (2,0)$ theory of type $A_1$ has a $\mathbb{Z}_2^{[2]}$ 2-form symmetry, which reduces to the aforementioned 0 and 1-form symmetries upon $T^2$ compactifications [62].

Let us note, however, that the reduction of the 2-form symmetry from six to five dimensions is subtle – in particular, a 5d theory obtained from the untwisted $S^1$ compactification of a 6d theory cannot simultaneously have both the 1-form and 2-form symmetries originating from 2-form symmetry of the 6d theory. One must choose which of the two is preserved in the 5d theory, as the 2-form symmetry in 6d is self-dual [62].[18] Such issues do not arise when further compactifying 5d theories on a circle. Thus, starting with the 6d $\mathcal{N} = (2,0)$ $A_1$ theory and compactifying on $S^1 \times S^1$ we have the following possibilities for higher form symmetries:

$$\mathbb{Z}_2^{[0]} \oplus \mathbb{Z}_2^{[1]}, \qquad\qquad \mathbb{Z}_2^{[1]} \oplus \mathbb{Z}_2^{[2]}, \tag{199}$$

---

[17]This $E_1$ quiver can be similarly obtained by a double-copy of the Kronecker quiver for the pure $SU(2)$ 4d $\mathcal{N} = 2$ gauge theory [40].

[18]It might be more precise to say that, in 6d, we do not have a 2-form symmetry but only a defect group [65].

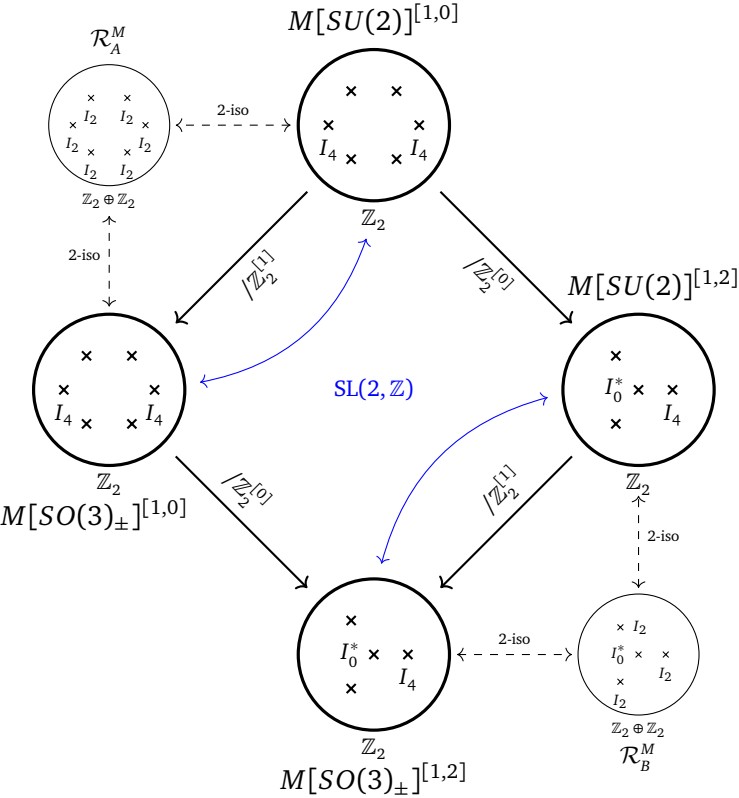

Figure 14: Global forms for the $M$-string theory on $T^2$. The MW torsion is indicated for each Coulomb branch. The blue (curved) arrows show which forms are in the same SL$(2, \mathbb{Z})$ duality orbit.

as already anticipated in (183). We thus denote by $M[G]^{[p,q]}$ the absolute theories which descend to the $E_1[G]^{[p,q]}$ theories.

**The $M[G]^{[1,0]}$ theories.** Let us first consider the models with 0 and 1-form symmetries, obtained from the relative theory $\mathcal{R}_A^M$. Following the prescription of section 2.2, we are looking at 2-isogenies generated by the torsion sections of $\mathcal{R}_A^M$. Note that since this relative M-string curve is identical to the 4d $\mathcal{N} = 2^*$ curve, upon substituting $\widetilde{u} \to \widetilde{U}^2 - 1$, and $m^2 \to 4M$, as in footnote 15, the arguments presented in section 3 have a straightforward generalisation to the M-string theory. Namely, due to the $\mathbb{Z}_2 \oplus \mathbb{Z}_2$ torsion of the relative curve, we will have three distinct global forms, which we denote by $M[SU(2)]^{[1,0]}$ and $M[SO(3)_\pm]^{[1,0]}$, respectively.

For instance, given an isogeny along $P_1$, the curve $\mathcal{R}_A^M/\langle P_1 \rangle$ will be associated to the $M[SU(2)]^{[1,0]}$ global form. Then, the SW curve for $\mathcal{R}_A^M/\langle P_2 \rangle$ is obtained from $\mathcal{R}_A^M/\langle P_1 \rangle$ by an **S** transformation on $\tau_{uv}$, while $\mathcal{R}_A^M/\langle P_3 \rangle$ follows from a further **T** transformation. These three curves are then invariant under $\Gamma^0(2)$ transformations, which is the duality group of the 4d $\mathcal{N} = 4$ theory on spin manifolds.

The generic mass-deformed curves for these three global forms are isomorphic, with singularity structure $(I_0; 2I_4, 4I_1)$, consisting again of two distinct copies of the corresponding 4d $\mathcal{N} = 2^*$ theory. As for the relative curve, the 'massless' limit $M = 0$ leads to the $(I_0; 2I_0^*)$ curve in all cases.

**The $M[G]^{[1,2]}$ theories.** The $M[G]^{[1,2]}$ theories are obtained by gauging the $\mathbb{Z}_2$ zero-form symmetry of the $M[G]^{[1,0]}$ theories. This gauging is performed similarly as in the $D_{S^1}E_1$ and

$D_{S^1}E_0$ geometries, by performing a base change. That is, one replaces $\widetilde{U} \rightarrow \sqrt{V}$, which is also accompanied by a quadratic twist such that the gauged theory will contain a $I_0^*$ fiber at the origin of the $V$-plane.

From this construction, it is clear that the curves of the $M[G]^{[1,2]}$ theories are identical to the 4d $\mathcal{N} = 2^*$ curves for the gauge group $G$, up to a quadratic twist which exchanges the $I_0^*$ fiber at infinity with the smooth fiber at the origin of the $u$-plane of the 4d theory. Thus, the gauging of the remaining 1-form symmetry follows from the 4d discussion. The global forms of the M-string theory on $T^2$ are shown in figure 14. Finally, let us note that, due to presence of the undeformable $I_0^*$ singularity, the BPS states of the $M[G]^{[1,2]}$ theories no longer admit a standard BPS quiver description.

# 7 Conclusions and outlook

In this work, we explored the network of different Seiberg-Witten curves that describe the Coulomb branch of rank-one 4d $\mathcal{N} = 2$ supersymmetric field theories with 1-form symmetries. We uncovered a self-consistent picture, wherein different global forms of the same local field theory are related by successions of isogenies between the Seiberg-Witten curves. In each case, it is important to correctly identify the *relative curve*, from which all the SW geometries for the absolute theories can be derived by performing isogenies along torsion sections. The 1-form symmetry of the absolute theories can then be read off from the torsion sections of the *absolute curves*. In the case of 5d $\mathcal{N} = 1$ theories on a circle, we also found that there are more possibilities of global structures obtained by discrete gauging of the four-dimensional 0-form symmetry inherited from the 'electric' 1-form symmetry in 5d.[19]

This analysis hopefully clears some potential confusions (at least, the authors') concerning the interpretation of various Seiberg-Witten geometries that appeared in the literature. Recently, Xie proposed a classification of rank-1 5d and 6d SCFTs with 8 supercharges [15] based on their SW geometry, using the known classification of rational elliptic surfaces [66, 67] and building on previous work in 4d by Caorsi and Cecotti [11] to study the flavour symmetry. In fact, the present paper establishes that many of the seemingly new models listed in [15] are actually different 4d $\mathcal{N} = 2$ absolute curves for the same higher-dimensional SCFT. (There remains a couple of potentially 'new' theories in the lists worked out in [15]. We hope to come back to this point in future work.)

Our discussion left open a number of questions. Most fundamentally, we did not give a first-principle derivation of the relationship between certain $N$-torsion sections and $\mathbb{Z}_N^{[1]}$-form symmetries. It is important to note that not all torsion sections of the absolute curves are directly related to the 1-form symmetries – in the case of the various 4d and 5d '$\mathfrak{su}(2)$ theories' we considered, there can also be $\mathbb{Z}_4$ factors in the Mordell-Weil group. In some cases, these extra sections are related to the global form of the flavour symmetry group [7]. Nonetheless, it seems that the true meaning of these $\mathbb{Z}_4$ rational sections, and of the corresponding isogenies, remains to be discovered. So far, we only observed that 4-isogenies relate different absolute curves amongst each other.

To answer these questions, we should probably better connect our RES approach to the geometric-engineering approach in string theory. The SW geometry can be embedded inside a local threefold in Type IIB, and this background is itself the mirror geometry for the 'standard' Type IIA geometric engineering [68, 69] – see [7] for a pedagogical review. One would then expect that the various global structures can be understood in terms of the Symmetry TFT [70]

---

[19]One could quibble about the use of the term 'global structure' here: gauging a discrete 0-form symmetry does change the algebra of local operators, albeit in a somewhat 'mild' manner. These 4d 'global structures' are directly related to proper 5d global structures, however.

captured by ten-dimensional topological terms in string theory – see *e.g.* [44,45,55,62,71–74]. This seems particularly interesting in the case of the local threefolds that engineer the 5d theories in Type IIA. To capture the structure of the 0- and 2-form symmetries in 4d, we would have to deal with defects realised by fundamental strings and NS5-branes wrapping relative cycles, in addition to the better-understood non-compact D2- and D4-branes.

Of course, a geometric-engineering approach would also more readily generalise to higher-rank theories – see *e.g.* [33,60,75–77] for early references and [78] for a review. At rank-1, we could afford ourselves of beautiful mathematical results about families of elliptic curves, which have no direct analogues for families of curves of higher genera. Recently, Argyres, Martone and Ray [13] clarified the general identification of what we call an "absolute curve" (which encodes all the lines operators of a given absolute theory) with a principally polarised abelian variety (giving us the periods of the SW geometry). They also pointed out that, for many 4d $\mathcal{N} = 2$ theories, only what we called the "relative curve" is known, and they then proceeded to discuss the additional data of a "line lattice" as a structure to supplement "by hand" if needed. It could be the case that, in general, there does not exist distinct higher-rank SW geometries for all possible global structures. We instead believe, more optimistically, that the "absolute curves" of, say, the pure $\mathfrak{su}(N)$ gauge theory, have not yet been discovered. Indeed, even the simple $SO(3)_\pm$ curves that we discussed in section 2 were not written down explicitly in the literature before, to the best of our knowledge. Very recently, the "relative and absolute" SW geometries for higher-rank 4d $\mathcal{N} = 3$ and $\mathcal{N} = 4$ SCFTs have been obtained in [79]. It would be very instructive to study the mass-deformations of these geometries.

Another interesting avenue for future research concerns the compactification of our setup to three dimensions. The resulting 3d $\mathcal{N} = 4$ Coulomb branch is essentially the Seiberg-Witten geometry itself [80], and it would be very interesting to see how the global structures studied here are reinterpreted in 3d. The translations along the elliptic fibers of the SW geometry generated by the torsion sections should become actual discrete actions on the 3d $\mathcal{N} = 4$ Coulomb branch, corresponding to a spontaneously-broken discrete 0-form symmetry inherited from the 4d 1-form symmetry. Relatedly, it would be instructive to translate our discussion to the class-$S$ language [81] (in particular, see [82] for a related discussion), and to study more explicitly the "isogenous Hitchin moduli spaces" mentioned in [2].

Last but not least, some of the theories we discussed also admit more general, categorical symmetries – see *e.g.* [83–87] for recent reviews of this rich subject. It remains an open question to precisely understand how such categorical symmetries can be imprinted into the Seiberg-Witten description of the low-energy Coulomb-branch physics. In section 2.3, we proposed that the $\mathbb{Z}_4$ torsion sections of the pure $SO(3)_\pm$ curve is likely a symptom of the non-invertible symmetry that exists in the pure $SO(3)$ gauge theory.[20] This is a potentially important relation which should be understood better.

# Acknowledgments

We are grateful to Ofer Aharony, Philip Argyres, Johannes Aspman, Lakshya Bhardwaj, Lea Bottini, Mathew Bullimore, Stefano Cremonesi, Michele Del Zotto, Elias Furrer, Dewi Gould, Andrea Grossutti, Mario Martone, Kentaro Ohmori, Sakura Schäfer-Nameki, Yuji Tachikawa, Xinyu Zhang and Boan Zhao for useful discussions and helpful comments.

**Funding information** The work of CC is supported by a University Research Fellowship Renewal 2022, "Singularities, supersymmetry and quantum invariants", of the Royal Society. CC

---

[20]We thank Lakshya Bhardwaj and Kentaro Ohmori for stimulating discussions on this point.

is also a Birmingham Fellow. The work of HM is supported by a Royal Society Research Grant for Research Fellows.

# A  Elliptic curves, isogenies and modular forms

In this appendix, we collect a variety of useful mathematical facts about rational elliptic surfaces and about modular forms, which we use extensively throughout the paper.

## A.1  Elliptic curves and Kodaira singularities

Let us first review some basic aspects of (families of) elliptic curves. As recalled in section 2, the SW geometry of rank-one 4d $\mathcal{N} = 2$ theories is a rational elliptic surface, which we may describe by the Weierstrass model (5), namely:

$$y^2 = 4x^3 - g_2(u)\,x - g_3(u)\,, \tag{A.1}$$

with the CB singularities located at the loci where the discriminant

$$\Delta(u) = g_2(u)^3 - 27g_3(u)^2\,, \tag{A.2}$$

vanishes. It is also customary to define the $J$-invariant as:

$$J(u) = \frac{g_2(u)^3}{\Delta(u)}\,, \tag{A.3}$$

which is a modular function when written in terms of the complex structure parameter $\tau$ of the elliptic fiber. This quantity is thus rather crucial for determining $u = u(\tau)$ – see *e.g.* [7, 8, 18, 19, 30] for more details. The possible singularities of the SW geometry are captured by the Kodaira classification of singular fibers, in terms of the order of vanishing of $g_2$, $g_3$ and of the discriminant:

$$g_2 \sim (u - u_*)^{\mathrm{ord}(g_2)}\,, \qquad g_3 \sim (u - u_*)^{\mathrm{ord}(g_3)}\,, \qquad \Delta \sim (u - u_*)^{\mathrm{ord}(\Delta)}\,. \tag{A.4}$$

The different types of fibers are listed in table 1, which is reproduced from [7] for the reader's convenience. There, we also list the monodromy induced by these singularities on the periods, the associated flavour symmetry if the singularity is fully deformable and the low-energy description in that case, with 'MN' standing for Minahan-Nemeschansky SCFTs [88, 89] and 'AD' for Argyres-Douglas SCFTs [90, 91]. Note also that most singularities require a fixed value for the complex structure parameter $\tau$ at the singular point, apart from $I_0^*$.

We should also recall that the monodromies can be understood in terms of light BPS states on the CB. For instance, the $I_1$ singularity occurs when a single charged particle of electromagnetic charge $(m, q)$ becomes massless – in the appropriate duality frame, the low-energy physics at that point is governed by a $U(1)$ gauge field coupled to a single massless hypermultiplet of charge 1. This induces a monodromy:

$$\mathbb{M}_*^{(m,q)} = \begin{pmatrix} 1 + mq & q^2 \\ -m^2 & 1 - mq \end{pmatrix}\,. \tag{A.5}$$

Similarly, if the low-energy description of the singularity is given by SQED with $k$ massless electrons or, more generally, with massless hypermultiplets of charges $q_j$ such that $\sum_j q_j^2 = k$, then the monodromy is conjugate to $T^k$ [4, 5], leading to an $I_k$ singularity. The low-energy physics description of all the Kodaira singularities, with any allowed deformation pattern, has been discussed in detail in [9, 27, 34].

Table 1: Kodaira classification of singular fibers and associated 4d low-energy physics.

| fiber | $\tau$ | ord($g_2$) | ord($g_3$) | ord($\Delta$) | $\mathbb{M}_*$ | 4d physics | $\mathfrak{g}$ flavour |
|---|---|---|---|---|---|---|---|
| $I_k$ | $i\infty$ | 0 | 0 | $k$ | $T^k$ | SQED | $\mathfrak{su}(k)$ |
| $I_k^*$ | $i\infty$ | 2 | 3 | $k+6$ | $PT^k$ | $SU(2)$, $N_f = 4+k > 4$ | $\mathfrak{so}(2k+8)$ |
| $I_0^*$ | $\tau_0$ | $\geq 2$ | $\geq 3$ | 6 | $P$ | $SU(2)$, $N_f = 4$ | $\mathfrak{so}(8)$ |
| $II$ | $e^{\frac{2\pi i}{3}}$ | $\geq 1$ | 1 | 2 | $(ST)^{-1}$ | AD$[A_1, A_2] = H_0$ | - |
| $II^*$ | $e^{\frac{2\pi i}{3}}$ | $\geq 4$ | 5 | 10 | $ST$ | MN $E_8$ | $\mathfrak{e}_8$ |
| $III$ | $i$ | 1 | $\geq 2$ | 3 | $S^{-1}$ | AD$[A_1, A_3] = H_1$ | $\mathfrak{su}(2)$ |
| $III^*$ | $i$ | 3 | $\geq 5$ | 9 | $S$ | MN $E_7$ | $\mathfrak{e}_7$ |
| $IV$ | $e^{\frac{2\pi i}{3}}$ | $\geq 2$ | 2 | 4 | $(ST)^{-2}$ | AD$[A_1, D_4] = H_2$ | $\mathfrak{su}(3)$ |
| $IV^*$ | $e^{\frac{2\pi i}{3}}$ | $\geq 3$ | 4 | 8 | $(ST)^2$ | MN $E_6$ | $\mathfrak{e}_6$ |

## A.2  Isogenies between rational elliptic surfaces

Let $L$, $L'$ be two lattices in $\mathbb{C}$, defined by their periods $(\omega_a, \omega_D)$ and $(\omega_a', \omega_D')$, respectively, as $L = \mathbb{Z}\omega_a + \mathbb{Z}\omega_D$, and similarly for $L'$. We say that the lattices are equivalent $L = L'$ if and only if their periods are related by (see *e.g.* lemma 1.3.1 in [92]):

$$\begin{pmatrix} \omega_D' \\ \omega_a' \end{pmatrix} = \gamma \begin{pmatrix} \omega_D \\ \omega_a \end{pmatrix}, \qquad \gamma \in \text{SL}(2,\mathbb{Z}). \tag{A.6}$$

We can also consider the case where the lattices $\alpha L$ and $L'$ are equivalent, for some $\alpha \in \mathbb{C}^*$, in which case the lattices are said to be *homothetic*. This, in particular, can be used to show that the lattice $L$ is homothetic to the lattice $L_\tau \equiv \frac{1}{\omega_a} L = \mathbb{Z} + \mathbb{Z}\tau$, for $\tau = \frac{\omega_D}{\omega_a}$, which is the more natural lattice description when discussing complex tori. Note that homothetic lattices give rise to the same elliptic curve, up to isomorphism (see *e.g.* theorem 5.35 in [93]).

More generally, we can consider an *isogeny* between complex tori, which is a non-zero holomorphic homomorphism $\psi_\alpha : \mathbb{C}/L \to \mathbb{C}/L'$. For such homomorphisms, there exists $\alpha \in \mathbb{C}^*$ with $\alpha L \subset L'$, such that:

$$\psi_\alpha(z + L) = \alpha z + L', \tag{A.7}$$

or, equivalently, $\psi_\alpha(0) = 0$. The kernel of an isogeny $\psi_\alpha$ is finite, and is given by $\ker(\psi_\alpha) = L'/\alpha L$. The degree of the isogeny is the index of $\alpha L$ in $L'$, being thus also the dimension of the kernel. For isogenous lattices, the periods are related by [92]:

$$\begin{pmatrix} \alpha \, \omega_D \\ \alpha \, \omega_a \end{pmatrix} = \tilde{\gamma} \begin{pmatrix} \omega_D' \\ \omega_a' \end{pmatrix}, \qquad \tilde{\gamma} \in M(2,\mathbb{Z}), \tag{A.8}$$

where now $\tilde{\gamma}$ is not necessarily an element of SL$(2,\mathbb{Z})$, as it was in the case of isomorphic lattices in (A.6), but any matrix with integer coefficients and positive determinant.[21] Note that the complex structure transforms as:

$$\tau = \frac{\omega_D}{\omega_a} = \tilde{\gamma}\left(\frac{\omega_D'}{\omega_a'}\right) = \tilde{\gamma}(\tau'), \tag{A.9}$$

where $\tilde{\gamma}$ acts as a fractional linear transformation on $\tau'$. Note also that the degree of the isogeny is $|\ker(\psi_\alpha)| = \det(\tilde{\gamma})$. It will often be useful to consider a change of basis such that

---

[21]This follows from $\text{Im}(\tau) \propto \det(\tilde{\gamma}) \, \text{Im}(\tau')$, with the proportionality factor being a positive number.

$\tilde{\gamma}$ in (A.8) is diagonal. Since $\alpha L \subset L'$, there exist positive integers $n_1, n_2 \in \mathbb{N}^*$, such that $\{n_1 \omega_a', n_2 \omega_D'\}$ is a basis for $\alpha L$. Thus, a basis for $L$ is given by $\{\frac{n_1}{\alpha} \omega_a', \frac{n_2}{\alpha} \omega_D'\}$. In this scenario, we have $\ker(\psi_\alpha) = \mathbb{Z}_{n_1} \oplus \mathbb{Z}_{n_2}$, giving the degree of the isogeny as $n_1 n_2$. Meanwhile, the complex structure parameter transforms as: $\tau = \frac{n_2}{n_1} \tau'$.

**Rational sections of elliptic surfaces.** The concept of isogenies between elliptic curves can be uplifted to isogenies between rational elliptic surfaces. Under this map, the smooth fibers are mapped to isogenous elliptic curves, while the singular fibers change in a non-trivial way.

Isogenies of elliptic surfaces are generated by torsion sections [94]. Each such section $P \in \mathrm{MW}(\mathcal{S})$ defines an automorphism $t_P$ of the elliptic surface $\mathcal{S}$ by translation by $P$ along every smooth fiber. If the torsion section has order $N$, then the quotient $\mathcal{S}/\langle t_P \rangle$ defines an $N$-to-1 homomorphism on the smooth fibers, which is also the degree of the isogeny. The action of $t_P$ on the multiplicative singular fibers (*i.e.* of type $I_n$) differs depending on how the torsion section intersects the singular fiber. In particular, if the section intersects the 'trivial' component $\Theta_0$ of the $I_n$ fiber, meaning that it does not intersect the 'node' of the singular curve in the Weierstrass model, then addition by $P$ leaves invariant the $n$ points at the intersections of the components of the $I_n$ fiber [94, 95]. Thus, under the quotient $\mathcal{S}/\langle t_P \rangle$, the singular fiber changes to:

$$I_n \mapsto I_{n \times N}. \tag{A.10}$$

Alternatively, $P$ can intersect non-trivially the singular fiber, in which case the automorphism rotates the components of the singular fibre by a $\frac{2\pi}{N}$ angle. As a result, the quotient by $\langle t_P \rangle$ identifies $N$ of the components of the singular fiber, leading to:

$$I_n \mapsto I_{n/N}, \tag{A.11}$$

as long as $N$ is prime, which will be our main point of interest.

**The Vélu formula.** For an elliptic curve given in Weierstrass normal form, the explicit form of the isogeny can be determined using Vélu's formula (see *e.g.* [96]). Let $P = (x_0, y_0)$ be a torsion section, with $y_0 = 0$ for the case of 2-torsion. We define:

$$t(P) = \begin{cases} 3x_0^2 - \frac{1}{4}g_2, & \text{for 2-torsion,} \\ 6x_0^2 - \frac{1}{2}g_2, & \text{for } N\text{-torsion,} \end{cases} \qquad w(P) = y_0^2 + t(P)x_0, \tag{A.12}$$

where $N \geq 3$. We would like to consider an isogeny generated by some torsion subgroup $G$. For this, we split the torsion sections into 2-torsion sections $G_2$, while the remaining sections are partitioned into two equal sized sets $G_+$ and $G_-$, with the torsion sections of $G_-$ being the inverses of those in $G_+$. Then, the isogeny $\psi : E \to E'$ is given explicitly by:

$$\psi(x, y) = \big(r(x), r'(x)y\big), \qquad r(x) \equiv x + \sum_{P \in G_2 \cup G_+} \left( \frac{t(P)}{x - x_0} + \frac{y_0^2}{(x - x_0)^2} \right). \tag{A.13}$$

We also define:

$$t(G) = \sum_{P \in G_2 \cup G_+} t(P), \qquad w(G) = \sum_{P \in G_2 \cup G_+} w(P), \tag{A.14}$$

such that the new curve $E'$ reads:

$$g_2' = g_2 + 20t(G), \qquad g_3' = g_3 + 28w(G). \tag{A.15}$$

More generally, we can also allow a rescaling by a non-zero factor $\alpha \in \mathbb{C}^*$, of the type:

$$(x, y, g_2, g_3) \mapsto \big(\alpha^{-2}x, \alpha^{-3}y, \alpha^{-4}g_2, \alpha^{-6}g_3\big). \tag{A.16}$$

Including such rescalings in the 2-isogeny, the holomorphic one-form transforms according to:

$$\frac{dx}{y} \to \frac{dx'}{y'} = \alpha \frac{dx}{y}. \tag{A.17}$$

Recall that the periods of the complex torus are computed as the integrals of this holomorphic 1-form over the 1-cycles of the torus. Thus, such rescalings correspond to rescalings of the underlying lattice by $\alpha$, as in (A.7). In section 2.2, we show that it is often important to include such rescalings in the isogeny, in order to preserve the Dirac pairing between BPS states. To do so, we choose the rescaling $\alpha$ in (A.8) such that:

$$\alpha^2 = \det \widetilde{\gamma}, \tag{A.18}$$

which implies that the periods of the two isogenous curves are related by an $SL(2, \mathbb{R})$ matrix. In such a case, if we start with a principally-polarised curves, the isogeny $\psi_\alpha$ gives us a curve which is no longer principally polarised.

## A.3 The Weierstrass elliptic function

In this section, we review some well known facts about the Weierstrass elliptic function. Given a lattice $L = \mathbb{Z}\omega_a + \mathbb{Z}\omega_D$, the Weierstrass elliptic function is a map from $\mathbb{C}/L$ to $\mathbb{C}$:

$$\wp(z, \omega_a, \omega_D) \equiv \wp(z, L) \equiv \frac{1}{z^2} + \sum_{\lambda \in L \setminus \{0\}} \left( \frac{1}{(z-\lambda)^2} - \frac{1}{\lambda^2} \right). \tag{A.19}$$

Oftentimes, we will simply use $\wp(z)$ and leave the lattice dependence implicit. Under a rescaling of the lattice by $\alpha$, we have

$$\wp(z, \alpha\omega_a, \alpha\omega_D) = \alpha^{-2}\wp(\alpha z, \omega_a, \omega_D). \tag{A.20}$$

Using the definition $L_\tau \cong \mathbb{Z} + \mathbb{Z}\tau$, we have:

$$\wp(z, \tau) \equiv \wp(z, 1, \tau) = \omega_a^2 \, \wp\left( \frac{z}{\omega_a}, \omega_a, \omega_D \right). \tag{A.21}$$

The Weierstrass $\wp$ function is a meromorphic function in $\mathbb{C}$, having double poles at each lattice point $\lambda \in L$. Moreover, from the definition it follows that $\wp$ is an even function, that is $\wp(-z) = \wp(z)$, and, moreover, we see that $\wp(z+\lambda) = \wp(z)$, for any lattice point $\lambda$ and $z \in \mathbb{C}$. The Weierstrass $\wp$ function also has a Laurent series expansion around $z = 0$, given by:

$$\wp(z) = \frac{1}{z^2} + \sum_{k=1}^{\infty} (2k+1) G_{2k+2} z^{2k}, \qquad G_k(L) = \sum_{\lambda \in L \setminus \{0\}} \lambda^{-k}, \tag{A.22}$$

where $G_k$ are sometimes referred to as Eisenstein series. The Weierstrass function satisfies the differential equation:

$$\wp'^2(z) = 4\wp^3(z) - g_2 \, \wp(z) - g_3, \tag{A.23}$$

where $g_2$ and $g_3$ are the lattice-dependent functions:

$$g_2 = 60 G_4, \qquad g_3 = 140 G_6. \tag{A.24}$$

Let us also note that the derivative $\wp'(z)$ is an odd function (since $\wp$ is even), so $\wp'(-z) = -\wp'(z)$. The functions $g_2, g_3$ are homogeneous functions of degree $-4$ and $-6$, meaning that

$$g_2(\alpha\omega_a, \alpha\omega_D) = \alpha^{-4} g_2(\omega_a, \omega_D), \qquad g_3(\alpha\omega_a, \alpha\omega_D) = \alpha^{-6} g_3(\omega_a, \omega_D), \tag{A.25}$$

for a constant factor $\alpha \neq 0$. For the upper-half-plane definition of the lattice, $L_\tau$, assuming that $\text{Im}(\tau) = \text{Im}\left(\frac{\omega_D}{\omega_a}\right) > 0$, we have the upper-half-plane functions

$$g_2(\tau) \equiv g_2(1,\tau) = \omega_a^4 g_2(\omega_a, \omega_D), \qquad g_3(\tau) \equiv g_3(1,\tau) = \omega_a^6 g_3(\omega_a, \omega_D), \qquad (A.26)$$

which become modular forms with Fourier series

$$g_2(\tau) = \frac{4\pi^4}{3}\left(1 + 240 \sum_{k=1}^{\infty} \sigma_3(k) q^k\right), \qquad g_3(\tau) = \frac{8\pi^6}{27}\left(1 - 504 \sum_{k=1}^{\infty} \sigma_5(k) q^k\right), \quad (A.27)$$

where $\sigma_a(k) = \sum_{d|k} d^a$ is the divisor function and $q = e^{2\pi i \tau}$.

The differential equation (A.23) is used to map the lattice $L \subset \mathbb{C}$ to the elliptic curve $E = \{(x,y) \,|\, y^2 = 4x^3 - g_2 x - g_3\}$, through the map:

$$z \mapsto (x,y) = \left(\wp(z), \wp'(z)\right), \qquad (A.28)$$

which is a group isomorphism between the complex torus $\mathbb{C}/L$ and $E(\mathbb{C})$, under addition modulo $L$. The roots of the cubic on the RHS of (A.23) are pairwise *distinct* and only depend on the lattice $L$, being the values of the Weierstrass function at the half-periods:[22]

$$e_1 = \wp\left(\frac{\omega_a}{2}\right), \qquad e_2 = \wp\left(\frac{\omega_D}{2}\right), \qquad e_3 = \wp\left(\frac{\omega_3}{2}\right) = \wp\left(\frac{\omega_a + \omega_D}{2}\right), \qquad (A.29)$$

with the property $e_1 + e_2 + e_3 = 0$. Note that in this case (A.23) becomes:

$$\wp'^2(z) = 4\left(\wp(z) - e_1\right)\left(\wp(z) - e_2\right)\left(\wp(z) - e_3\right). \qquad (A.30)$$

Note, thus, that the derivative of the Weierstrass elliptic function vanishes for the values of $z$ corresponding to the half-periods.

## A.4 Modular forms

The ring of modular forms for the $\text{SL}(2,\mathbb{Z})$ group is generated by the holomorphic Eisenstein series of weights 4 and 6, defined as:

$$E_4(\tau) = 1 + 240 \sum_{n=1}^{\infty} \frac{n^3 q^n}{1 - q^n}, \qquad E_6(\tau) = 1 - 504 \sum_{n=1}^{\infty} \frac{n^5 q^n}{1 - q^n}. \qquad (A.31)$$

Their **T** and **S** transformations are:

$$E_k(\tau + 1) = E_k(\tau), \qquad E_k\left(-\frac{1}{\tau}\right) = \tau^k E_k(\tau), \qquad (A.32)$$

with their zeroes at $e^{\frac{2\pi i}{3}}$ (or $e^{\frac{i\pi}{3}}$) and $i$, respectively. We introduce the Jacobi theta functions:

$$\theta_1(z;\tau) = -i \sum_{n \in \mathbb{Z}+\frac{1}{2}} (-1)^{n-\frac{1}{2}} q^{\frac{n^2}{2}} e^{2\pi i z n}, \qquad \theta_2(z;\tau) = \sum_{n \in \mathbb{Z}+\frac{1}{2}} q^{\frac{n^2}{2}} e^{2\pi i z n},$$

$$\theta_3(z;\tau) = \sum_{n \in \mathbb{Z}} q^{\frac{n^2}{2}} e^{2\pi i z n}, \qquad \theta_3(z;\tau) = \sum_{n \in \mathbb{Z}} (-1)^n q^{\frac{n^2}{2}} e^{2\pi i z n}, \qquad (A.33)$$

---

[22] This is a slight abuse of notation. With $\wp(z;\tau)$ defined as above, the half periods are at $z = \frac{1}{2}, \frac{\tau}{2}, \frac{\tau+1}{2}$.

which also have the useful product expressions:

$$
\begin{aligned}
\theta_1(z;\tau) &= -iq^{\frac{1}{8}} y^{\frac{1}{2}} \prod_{k=1}^{\infty} \left(1-q^k\right)\left(1-yq^k\right)\left(1-y^{-1}q^{k-1}\right), \\
\theta_2(z;\tau) &= q^{\frac{1}{8}} y^{\frac{1}{2}} \prod_{k=1}^{\infty} \left(1-q^k\right)\left(1+yq^k\right)\left(1+y^{-1}q^{k-1}\right), \\
\theta_3(z;\tau) &= \prod_{k=1}^{\infty} \left(1-q^k\right)\left(1+yq^{k-\frac{1}{2}}\right)\left(1+y^{-1}q^{k-\frac{1}{2}}\right), \\
\theta_4(z;\tau) &= \prod_{k=1}^{\infty} \left(1-q^k\right)\left(1-yq^{k-\frac{1}{2}}\right)\left(1-y^{-1}q^{k-\frac{1}{2}}\right),
\end{aligned}
\tag{A.34}
$$

with $y \equiv e^{2\pi i z}$. We then define the Jacobi forms as $\vartheta(\tau) \equiv \theta(0;\tau)$, namely:

$$
\vartheta_2(\tau) = \sum_{n\in\mathbb{Z}+\frac{1}{2}} q^{\frac{n^2}{2}}, \qquad \vartheta_3(\tau) = \sum_{n\in\mathbb{Z}} q^{\frac{n^2}{2}}, \qquad \vartheta_4(\tau) = \sum_{n\in\mathbb{Z}} (-1)^n q^{\frac{n^2}{2}}.
\tag{A.35}
$$

(Note that $\theta_1(0;\tau) = 0$.) Their transformations under $\mathrm{SL}(2,\mathbb{Z})$ are as follows:

$$
\begin{aligned}
\mathbf{T}: &\quad \vartheta_2 \to e^{\frac{i\pi}{4}} \vartheta_2, &\quad \vartheta_3 \to \vartheta_4, &\quad \vartheta_4 \to \vartheta_3, \\
\mathbf{S}: &\quad \vartheta_2 \to \sqrt{-i\tau}\, \vartheta_4, &\quad \vartheta_3 \to \sqrt{-i\tau}\, \vartheta_3, &\quad \vartheta_4 \to \sqrt{-i\tau}\, \vartheta_2.
\end{aligned}
\tag{A.36}
$$

Note that the zeros of these functions lie along the real axis. Moreover, they satisfy the following identity:

$$
\vartheta_2(\tau)^4 + \vartheta_4(\tau)^4 = \vartheta_3(\tau)^4.
\tag{A.37}
$$

It is sometimes useful to rewrite the theta functions in terms of Dedekind-$\eta$ quotients, as follows:

$$
\vartheta_2(\tau) = \frac{2\eta(2\tau)^2}{\eta(\tau)}, \qquad \vartheta_3(\tau) = \frac{\eta(\tau)^5}{\eta\left(\frac{\tau}{2}\right)^2 \eta(2\tau)^2}, \qquad \vartheta_4(\tau) = \frac{\eta\left(\frac{\tau}{2}\right)^2}{\eta(\tau)},
\tag{A.38}
$$

where we introduced:

$$
\eta(\tau) = q^{\frac{1}{24}} \prod_{j=1}^{\infty} (1-q^j).
\tag{A.39}
$$

From here, one finds:

$$
\vartheta_2(\tau)\vartheta_3(\tau)\vartheta_4(\tau) = 2\eta(\tau)^3.
\tag{A.40}
$$

Let us point out that the Eisenstein series can be also expressed in terms of the theta functions, with:

$$
E_4 = \frac{1}{2}\left(\vartheta_2^8 + \vartheta_3^8 + \vartheta_4^8\right), \qquad E_6 = \frac{1}{2}\left(\vartheta_3^4 - 2\vartheta_4^4\right)\left(\vartheta_4^4 - 2\vartheta_3^4\right)\left(\vartheta_3^4 + \vartheta_4^4\right).
\tag{A.41}
$$

Thus, we also have:

$$
j = 1728J = 1728\frac{E_4^3}{E_4^3 - E_6^2} = 256\frac{(\vartheta_3^8 - \vartheta_3^4\vartheta_4^4 + \vartheta_4^8)^3}{\vartheta_2^8\vartheta_3^8\vartheta_4^8}.
\tag{A.42}
$$

The Weierstrass $\wp$ function can be expressed in terms of Jacobi $\theta$-functions as:

$$
\wp(z) = \left(\vartheta_2\vartheta_3\frac{\theta_4(z)}{\theta_1(z)}\right)^2 - \frac{1}{3}\left(\vartheta_2^4 + \vartheta_3^4\right).
\tag{A.43}
$$

Then, the half-periods defined in the previous subsection read:

$$e_1(\tau) = \frac{\vartheta_3^4(\tau) + \vartheta_4^4(\tau)}{3}, \qquad e_2(\tau) = -\frac{\vartheta_2^4(\tau) + \vartheta_3^4(\tau)}{3}, \qquad e_3(\tau) = \frac{\vartheta_2^4(\tau) - \vartheta_4^4(\tau)}{3}. \quad \text{(A.44)}$$

Under $SL(2,\mathbb{Z})$ transformations, we have:

$$\begin{aligned}
\mathbf{T}: & \quad e_1 \to e_1, & e_2 \to e_3, & \quad e_3 \to e_2, \\
\mathbf{S}: & \quad e_1 \to \tau^2 e_2, & e_2 \to \tau^2 e_1, & \quad e_3 \to \tau^2 e_3.
\end{aligned} \qquad \text{(A.45)}$$

Additionally, these satisfy [31, 32]:

$$\sum_{j=1}^{3} e_j^2 = \frac{2}{3} E_4, \qquad \sum_{j=1}^{3} e_j^3 = \frac{2}{9} E_6, \qquad \prod_{j=1}^{3} e_j = \frac{2}{27} E_6. \qquad \text{(A.46)}$$

In the context of 2-isogenies, of particular interest will also be the identities:

$$\begin{aligned}
\vartheta_2(2\tau)^4 = \frac{(\vartheta_3^2 - \vartheta_4^2)^2}{4}, & \qquad \vartheta_3(2\tau)^4 = \frac{(\vartheta_3^2 + \vartheta_4^2)^2}{4}, & \qquad \vartheta_4(2\tau)^4 = \vartheta_3^2 \vartheta_4^2, \\
\vartheta_2\left(\frac{\tau}{2}\right)^4 = 4\vartheta_2^2 \vartheta_3^2, & \qquad \vartheta_3\left(\frac{\tau}{2}\right)^4 = (\vartheta_2^2 + \vartheta_3^2)^2, & \qquad \vartheta_4\left(\frac{\tau}{2}\right)^4 = (\vartheta_2^2 - \vartheta_3^2)^2,
\end{aligned} \qquad \text{(A.47)}$$

where the argument of the theta functions is $\tau$, unless otherwise specified. Finally, let us also define the modular $\lambda$-function as:

$$\lambda(\tau) = \frac{\vartheta_2(\tau)^4}{\vartheta_3(\tau)^4}, \qquad j(\tau) = 256 \frac{(1 - \lambda + \lambda^2)^3}{(1 - \lambda)^2 \lambda^2}, \qquad \text{(A.48)}$$

which is a modular function for the congruence subgroup $\Gamma(2)$, to be defined in the next subsection.

## A.5 Congruence subgroups of $PSL(2, \mathbb{Z})$

Throughout the main text we make use of certain well-known aspects about congruence subgroups. This appendix summarises relevant information about such groups, based on similar appendices from related works [7, 8]. We also refer to [97], for example, for a more detailed exposition of these concepts.

The congruence subgroups of $PSL(2, \mathbb{Z})$ are matrix subgroups defined by congruence conditions on their entries. First, the principal congruence subgroups of level $N$ of $PSL(2, \mathbb{Z})$ can be summarised as follows:

$$\Gamma(N) = \left\{ \begin{pmatrix} a & b \\ c & d \end{pmatrix} \in PSL(2, \mathbb{Z}) : \begin{pmatrix} a & b \\ c & d \end{pmatrix} = \begin{pmatrix} 1 & 0 \\ 0 & 1 \end{pmatrix} \bmod N \right\}. \qquad \text{(A.49)}$$

Other common congruence subgroups are:

$$\begin{aligned}
\Gamma_0(N) &= \left\{ \begin{pmatrix} a & b \\ c & d \end{pmatrix} \in PSL(2, \mathbb{Z}) : c = 0 \bmod N \right\}, \\
\Gamma^0(N) &= \left\{ \begin{pmatrix} a & b \\ c & d \end{pmatrix} \in PSL(2, \mathbb{Z}) : b = 0 \bmod N \right\}.
\end{aligned} \qquad \text{(A.50)}$$

The $\Gamma_0(N)$ and $\Gamma^0(N)$ subgroups are related by conjugation by $S \in PSL(2, \mathbb{Z})$.

The elements of the modular group act on the upper half-plane $\mathbb{H}$ as:

$$\tau \mapsto \frac{a\tau + b}{c\tau + d}, \qquad \forall\, \tau \in \mathbb{H}. \tag{A.51}$$

We also define a weight $k$ modular form for a subgroup $\Gamma \subset \mathrm{PSL}(2,\mathbb{Z})$ as the holomorphic function from the upper half-plane to the complex plane satisfying [97]:

$$f\left(\frac{a\tau + b}{c\tau + d}\right) = (c\tau + d)^k f(\tau). \tag{A.52}$$

Here, evidently, the transformation matrix is an element of $\Gamma$. We have already introduced the modular $\lambda$-function, which is a weight 0 modular form for $\Gamma(2)$. Note that $\vartheta_2^4, \vartheta_3^4$ and $\vartheta_4^4$ are all weight 2 modular forms for $\Gamma(2)$.

A fundamental domain for $\Gamma \subset \mathrm{PSL}(2,\mathbb{Z})$ is an open subset $\mathcal{F}_\Gamma \subset \mathbb{H}$ such that no two distinct points are equivalent under the action of $\Gamma$, unless they are on the boundary of $\mathcal{F}_\Gamma$. Additionally, any point of the upper half-plane is mapped to the closure of this set $\mathcal{F}_\Gamma$. Given a list of coset representatives $\{\alpha_i\}$, the fundamental domain $\mathcal{F}_\Gamma$ can be obtained as the disjoint union [7,8]:

$$\mathcal{F}_\Gamma = \bigsqcup_{i=1}^{n_\Gamma} \alpha_i \mathcal{F}_0. \tag{A.53}$$

Here, $\mathcal{F}_0$ is the fundamental domain of the modular group, while $n_\Gamma$ is the index of $\Gamma$ in $\mathrm{PSL}(2,\mathbb{Z})$, defined as the number of right-cosets of $\Gamma$ in the modular group. The coset representatives $\alpha_i$ are chosen such that $\mathcal{F}_\Gamma$ has a connected interior.

Congruence subgroups can have two types of special points: *cusps* and *elliptic points*. The cusps are equivalence classes in $\mathbb{Q} \cup \{\infty\}$ under the action of the congruence subgroup, while elliptic points are points with non-trivial stabilizer. The modular group $\mathrm{PSL}(2,\mathbb{Z})$ has a single cusp, with the equivalence class representative typically chosen as $\tau_\infty = i\infty$. Every congruence subgroup $\Gamma$ has at least one cusp, which corresponds to the aforementioned representative. The *width* of this cusp $\tau_\infty$ in $\Gamma$ is the smallest integer $w$ such that $T^w \in \Gamma$. Note that for more general cusps positioned on the real axis at $\widetilde{\tau} = \gamma \tau_\infty$, for some $\gamma \in \mathrm{PSL}(2,\mathbb{Z})$, the width is defined as the width of $\tau_\infty$ for the group $\gamma^{-1}\Gamma\gamma$.

The Coulomb branch is called *modular* if it can be mapped to the upper-half plane by a bi-holomorphim. A key point of this mapping is that cusps correspond to certain singular fibers of the SW geometry [7,8]:

$$\text{cusp of width } w \;\longleftrightarrow\; I_w \text{ or } I_w^* \text{ singular fiber.} \tag{A.54}$$

In such cases, fundamental domains are particularly useful for determining the monodromies around the Coulomb branch singularities. We use this correspondence throughout the text to determine the electromagnetic charges of the light BPS states, directly from the fundamental domains. Consider, for instance, a cusp of width $w$ (corresponding to an undeformable $I_w$ singularity) positioned at $\tau = -\frac{q}{m}$ on the real axis, with $q, m \in \mathbb{Z}$ mutually prime. Then, we associate a light BPS state of magnetic-electric charges $\pm\sqrt{w}\,(m,q)$. Given a coset representative $\alpha$ such that $\alpha(i\infty) = -\frac{q}{m}$, we have:

$$\alpha T^w \alpha^{-1} = \mathbb{M}_{\sqrt{w}(m,q)}, \tag{A.55}$$

where the monodromy generated by the BPS state is given as in (A.5). We refer to [8] for more details on this identification.

# B Charge lattices and defect groups

In this appendix, we expound on the relation between our conventions and the conventions of [3] for the $\mathfrak{su}(2)$ gauge theories. Let $\mathfrak{t} \cong \mathfrak{u}(1)$ denote the one-dimensional Cartan subalgebra. The weight lattices of $SU(2)$ and $SO(3)$ are related as:

$$\Lambda_{\mathrm{w}}^{SO(3)} \subset \Lambda_{\mathrm{w}}^{SU(2)} \equiv \Lambda_{\mathrm{w}}^{\mathfrak{su}(2)} \subset \mathfrak{t}^*, \tag{B.1}$$

and the magnetic weight lattices are related as:

$$\mathfrak{t} \supset \Lambda_{\mathrm{mw}}^{SO(3)} \equiv \Lambda_{\mathrm{mw}}^{\mathfrak{su}(2)} \supset \Lambda_{\mathrm{mw}}^{SU(2)}. \tag{B.2}$$

The non-trivial center of $SU(2)$ is isomorphic to the following quotients:

$$Z(SU(2)) \cong \frac{\Lambda_{\mathrm{w}}^{SU(2)}}{\Lambda_{\mathrm{w}}^{SO(3)}} \cong \frac{\Lambda_{\mathrm{mw}}^{SO(3)}}{\Lambda_{\mathrm{mw}}^{SU(2)}} \cong \mathbb{Z}_2. \tag{B.3}$$

Note also that, for a fixed gauge group $G$, the weight and magnetic weight lattices are dual, $\Lambda_{\mathrm{mw}}^{G} \cong \left(\Lambda_{\mathrm{w}}^{G}\right)^*$. When discussing the $\mathfrak{su}(2)$ gauge theory without committing to a global form of the gauge group, it is natural to denote electromagnetic charges as in [3]:

$$(\lambda_m, \lambda_e) \in \Lambda_{\mathrm{mw}}^{\mathfrak{su}(2)} \oplus \Lambda_{\mathrm{w}}^{\mathfrak{su}(2)}, \qquad \lambda_m, \lambda_e \in \mathbb{Z}. \tag{B.4}$$

In this paper, instead, we consider the $SU(2)$ and $SO(3)_+$ charges:

$$\begin{aligned}
(m, q) &\in \Lambda_{\mathrm{mw}}^{SU(2)} \oplus \Lambda_{\mathrm{w}}^{SU(2)}, & m, q &\in \mathbb{Z}, \\
(m_+, q_+) &\in \Lambda_{\mathrm{mw}}^{SO(3)} \oplus \Lambda_{\mathrm{w}}^{SO(3)}, & m_+, q_+ &\in \mathbb{Z},
\end{aligned} \tag{B.5}$$

which are related to the conventions of [3] by:

$$(m, q) = \left(\frac{\lambda_m}{2}, \lambda_e\right), \qquad (m_+, q_+) = \left(\lambda_m, \frac{\lambda_e}{2}\right). \tag{B.6}$$

This directly gives us the relation (29). We also have:

$$(m_-, q_-) = \left(\lambda_m, \frac{\lambda_e - \lambda_m}{2}\right), \qquad m_-, q_- \in \mathbb{Z}, \tag{B.7}$$

for the $SO(3)_-$ normalisation. Starting from the $\mathfrak{su}(2)$ gauge theory and imposing the condition

$$\left\langle (\lambda_m, \lambda_e), (\lambda'_m, \lambda'_e) \right\rangle = 0 \bmod 2, \tag{B.8}$$

so that the lines are mutually local, there are three consistent spectrum of lines [2, 3]:

$$\begin{aligned}
SU(2) &: & \lambda_m &\in 2\mathbb{Z}, & \lambda_e &\in \mathbb{Z}, \\
SO(3)_+ &: & \lambda_m &\in \mathbb{Z}, & \lambda_e &\in 2\mathbb{Z}, \\
SO(3)_- &: & \lambda_m, \lambda_e &\in \mathbb{Z}, & \lambda_m + \lambda_e &\in 2\mathbb{Z}.
\end{aligned} \tag{B.9}$$

This is equivalent to our condition that $\gamma_L \in \Gamma$ in section 2.1.1, meaning that the charges $m, q$ and $m_\pm, q_\pm$ are all integers.

**Defect group.** Given a 4d $\mathcal{N} = 2$ gauge theory whose BPS particles on its Coulomb branch have electromagnetic charges taking value in some lattice $\Lambda$, the defect group of the theory is the group $\mathbb{D} = \Lambda^*/\Lambda$ that encodes the charges of BPS lines that cannot be screened by dynamical particles – see *e.g.* [14, 65]. In the case at hand, we simply have:

$$\mathbb{D} = \mathbb{Z}_2 \oplus \mathbb{Z}_2, \tag{B.10}$$

corresponding to the elements $(\lambda_m \bmod 2, \lambda_e \bmod 2)$. The absolute theories (B.9) correspond to the three distinct Lagrangian subgroups $\mathbb{Z}_2 \subset \mathbb{D}$ generated by $(0, 1)$, $(1, 0)$ and $(1, 1)$, respectively.

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
