# Peer review of "Reading between the rational sections: Global structures of 4d $\mathcal{N}=2$ KK theories"

_SciPost Physics, doi:SciPost Phys. 16, 137 (2024)_

## Round 3 · Referee Report · Anonymous (Referee 1) · 2024-2-13

Report

This is a sound, original, and well-written paper on a subject of interest to the quantum field theory community, especially those interested in supersymmetric field theories. It addresses, through detailed calculation in a few examples, the general problem of how and to what extent various symmetry properties of supersymmetric QFTs are reflected in their supersymmetry-protected low energy observables --- in this case the geometry of the moduli space of vacua of the theory. This gives evidence for an an interesting and compelling hypothesis relating the 1-form symmetry of the theory to the Mordell-Weil group of the Seiberg-Witten curve describing the Coulomb branch moduli geometry.

---

## Round 3 · Referee Report · Anonymous (Referee 2) · 2024-5-2

Report

This is a very high quality paper meeting the journal expectation "Provide a novel and synergetic link between different research areas.
" -- The paper connects the study of 1-form symmetries of N=2 theories with their associated Seiberg-Witten geometries.

The connection has been explained cleanly, explored in a multitude of scenarios, and potential pitfalls and directions of future research have been sketched clearly. Thus the paper meets all the general acceptance criteria of the journal.

I have no complaints about the scientific content of the paper and recommend the paper for publication wholeheartedly.

Recommendation

Publish (surpasses expectations and criteria for this Journal; among top 10%)

---

## Editorial Decision

published